# FREE LUNCH FOR DOMAIN ADVERSARIAL TRAINING: ENVIRONMENT LABEL SMOOTHING

**Yi-Fan Zhang**[1,2]*, **Xue Wang**[3], **Jian Liang**[1,2],
**Zhang Zhang**[1,2], **Liang Wang**[1,2], **Rong Jin**[3]†, **Tieniu Tan**[1,2]
[1]National Laboratory of Pattern Recognition (NLPR), Institute of Automation
[2]School of Artificial Intelligence, University of Chinese Academy of Sciences (UCAS)
[3] Machine Intelligence Technology, Alibaba Group.

## ABSTRACT

A fundamental challenge for machine learning models is how to generalize learned models for out-of-distribution (OOD) data. Among various approaches, exploiting invariant features by Domain Adversarial Training (DAT) received widespread attention. Despite its success, we observe training instability from DAT, mostly due to over-confident domain discriminator and environment label noise. To address this issue, we proposed **E**nvironment **L**abel **S**moothing (ELS), which encourages the discriminator to output soft probability, which thus reduces the confidence of the discriminator and alleviates the impact of noisy environment labels. We demonstrate, both experimentally and theoretically, that ELS can improve training stability, local convergence, and robustness to noisy environment labels. By incorporating ELS with DAT methods, we are able to yield the state-of-art results on a wide range of domain generalization/adaptation tasks, particularly when the environment labels are highly noisy. The code is avaliable at https://github.com/yfzhang114/Environment-Label-Smoothing.

## 1 INTRODUCTION

Despite being empirically effective on visual recognition benchmarks (Russakovsky et al., 2015), modern neural networks are prone to learning shortcuts that stem from spurious correlations (Geirhos et al., 2020), resulting in poor generalization for out-of-distribution (OOD) data. A popular thread of methods, minimizing domain divergence by Domain Adversarial Training (DAT) (Ganin et al., 2016), has shown better domain transfer performance, suggesting that it is potential to be an effective candidate to extract domain-invariant features. Despite its power for domain adaptation and domain generalization, DAT is known to be difficult to train and converge (Roth et al., 2017; Jenni & Favaro, 2019; Arjovsky & Bottou, 2017; Sønderby et al., 2016).

The main difficulty for stable training is to maintain healthy competition between the encoder and the domain discriminator. Recent work seeks to attain this goal by designing novel optimization methods (Acuna et al., 2022; Rangwani et al., 2022), however, most of them require additional optimization steps and slow the convergence. In this work, we aim to tackle the challenge from a totally different aspect from previous works, *i.e.,* the environment label design.

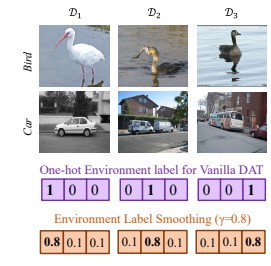

Figure 1: **A motivating example** of ELS with 3 domains on the VLCS dataset.

Two important observations that lead to the training instability of DAT motivate this work: (i) The environment label noise from environment partition (Creager et al., 2021) and training (Thanh-Tung et al., 2019). As shown in Figure 1, different domains of the VLCS benchmark have no significant difference in image style and some images are indistinguishable for which domain they belong. Besides, when the encoder gets better, the generated features from different domains are more similar. However, regardless of their quality, features are still labeled differently. As shown in (Thanh-Tung et al., 2019), *discriminators will*

---

*Work done during an internship at Alibaba Group.
†Work done at Alibaba Group, and now affiliated with Twitter.

*overfit these mislabelled examples and then has poor generalization capability*. (ii) To our best knowledge, DAT methods all assign one-hot environment labels to each data sample for domain discrimination, where the output probabilities will be highly confident. For DAT, *a very confident domain discriminator leads to highly oscillatory gradients* (Arjovsky & Bottou, 2017), which is harmful to training stability. The first observation inspires us to force the training process to be robust with regard to environment-label noise, and the second observation encourages the discriminator to estimate soft probabilities rather than confident classification. To this end, we propose **E**nvironment **L**abel **S**moothing (ELS), which is a simple method to tackle the mentioned obstacles for DAT. Next, we summarize the main methodological, theoretical, and experimental contributions.

**Methodology**: To our best knowledge, this is the first work to smooth environment labels for DAT. The proposed ELS yields three main advantages: (i) it does not require any extra parameters and optimization steps and yields faster convergence speed, better training stability, and more robustness to label noise theoretically and empirically; (ii) despite its efficiency, ELS is also easily to implement. People can easily incorporate ELS with any DAT methods in very few lines of code; (iii) ELS equipped DAT methods attain superior generalization performance compared to their native counterparts;

**Theories**: The benefit of ELS is theoretically verified in the following aspects. (i) *Training stability*. We first connect DAT to Jensen–Shannon/Kullback–Leibler divergence minimization, where ELS is shown able to extend the support of training distributions and relieve both the oscillatory gradients and gradient vanishing phenomenons, which results in stable and well-behaved training. (ii) *Robustness to noisy labels*. We theoretically verify that the negative effect caused by noisy labels can be reduced or even eliminated by ELS with a proper smooth parameter. (iii) *Faster non-asymptotic convergence speed*. We analyze the non-asymptotic convergence properties of DANN. The results indicate that incorporating with ELS can further speed up the convergence process. In addition, we also provide the empirical gap and analyze some commonly used DAT tricks.

**Experiments:** (i) Experiments are carried out on various benchmarks with different backbones, including *image classification, image retrieval, neural language processing, genomics data, graph, and sequential data*. ELS brings consistent improvement when incorporated with different DAT methods and achieves competitive or SOTA performance on various benchmarks, *e.g.,* average accuracy on Rotating MNIST ($52.1\% \to 62.1\%$), worst group accuracy on CivilComments ($61.7\% \to 65.9\%$), test ID accuracy on RxRx1 ($22.9\% \to 26.7\%$), average accuracy on Spurious-Fourier dataset ($11.1\% \to 15.6\%$). (ii) Even if the environment labels are random or partially known, the performance of ELS + DANN will not degrade much and is superior to native DANN. (iii) Abundant analyzes on training dynamics are conducted to verify the benefit of ELS empirically. (iv) We conduct thorough ablations on hyper-parameter for ELS and some useful suggestions about choosing the best smooth parameter considering the dataset information are given.

## 2 METHODOLOGY

For domain generalization tasks, there are $M$ source domains $\{\mathcal{D}_i\}_{i=1}^M$. Let the hypothesis $h$ be the composition of $h = \hat{h} \circ g$, where $g \in \mathcal{G}$ pushes forward the data samples to a representation space $\mathcal{Z}$ and $\hat{h} = (\hat{h}_1(\cdot), \ldots, \hat{h}_M(\cdot)) \in \hat{\mathcal{H}} : \mathcal{Z} \to [0,1]^M; \sum_{i=1}^M \hat{h}_i(\cdot) = 1$ is the domain discriminator with softmax activation function. The classifier is defined as $\hat{h}' \in \hat{\mathcal{H}}' : \mathcal{Z} \to [0,1]^C; \sum_{i=1}^C \hat{h}'_i(\cdot) = 1$, where $C$ is the number of classes. The cost used for the discriminator can be defined as:

$$\max_{\hat{h} \in \hat{\mathcal{H}}} d_{\hat{h},g}(\mathcal{D}_1, \ldots, \mathcal{D}_M) = \max_{\hat{h} \in \mathcal{H}} \mathbb{E}_{\mathbf{x} \in \mathcal{D}_1} \log \hat{h}_1 \circ g(\mathbf{x}) + \cdots + \mathbb{E}_{\mathbf{x} \in \mathcal{D}_M} \log \hat{h}_M \circ g(\mathbf{x}), \quad (1)$$

where $\hat{h}_i \circ g(\mathbf{x})$ is the prediction probability that $\mathbf{x}$ is belonged to $\mathcal{D}_i$. Denote $y$ the class label, then the overall objective of DAT is

$$\min_{\hat{h}',g} \max_{\hat{h}} \frac{1}{M} \sum_{i=1}^M \mathbb{E}_{\mathbf{x} \in \mathcal{D}_i} [\ell(\hat{h}' \circ g(\mathbf{x}), y)] + \lambda d_{\hat{h},g}(\mathcal{D}_1, \ldots, \mathcal{D}_M), \quad (2)$$

where $\ell$ is the cross-entropy loss for classification tasks and MSE for regression tasks, and $\lambda$ is the tradeoff weight. We call the first term empirical risk minimization (ERM) part and the second term

adversarial training (AT) part. Applying ELS, the target in Equ. (1) can be reformulated as

$$\max_{\hat{h} \in \hat{\mathcal{H}}} d_{\hat{h},g,\gamma}(\mathcal{D}_1, \ldots, \mathcal{D}_M) = \max_{\hat{h} \in \hat{\mathcal{H}}} \mathbb{E}_{\mathbf{x} \in \mathcal{D}_1} \left[ \gamma \log \hat{h}_1 \circ g(\mathbf{x}) + \frac{(1-\gamma)}{M-1} \sum_{j=1; j \neq 1}^{M} \log \left( \hat{h}_j \circ g(\mathbf{x}) \right) \right] + \cdots +$$

$$\mathbb{E}_{\mathbf{x} \in \mathcal{D}_M} \left[ \gamma \log \hat{h}_M \circ g(\mathbf{x}) + \frac{(1-\gamma)}{M-1} \sum_{j=1; j \neq M}^{M} \log \left( \hat{h}_j \circ g(\mathbf{x}) \right) \right].$$

$$(3)$$

## 3 THEORETICAL VALIDATION

In this section, we first assume the discriminator is optimized with no constraint, providing a theoretical interpretation of applying ELS. Then how ELS makes the training process more stable is discussed based on the interpretation and some analysis of the gradients. We next theoretically show that with ELS, the effect of label noise can be eliminated. Finally, to mitigate the impact of the **no constraint** assumption, the empirical gap, parameterization gap, and non-asymptotic convergence property are analyzed respectively. All omitted proofs can be found in the Appendix.

### 3.1 DIVERGENCE MINIMIZATION INTERPRETATION

In this subsection, the connection between ELS/one-sided ELS and divergence minimization is studied. The advantages brought by ELS and why GANs prefer one-sided ELS are theoretically claimed. We begin with the two-domain setting, which is used in domain adaptation and generative adversarial networks. Then the result in the multi-domain setting is further developed.

**Proposition 1.** *Given two domain distributions $\mathcal{D}_S, \mathcal{D}_T$ over $X$, and a hypothesis class $\mathcal{H}$. We suppose $\hat{h} \in \hat{\mathcal{H}}$ the optimal discriminator with no constraint, denote the mixed distributions with hyper-parameter $\gamma \in [0.5, 1]$ as $\left\{ \begin{array}{l} \mathcal{D}_{S'} = \gamma \mathcal{D}_S + (1-\gamma) \mathcal{D}_T \\ \mathcal{D}_{T'} = \gamma \mathcal{D}_T + (1-\gamma) \mathcal{D}_S \end{array} \right.$ . Then minimizing domain divergence by adversarial training with **ELS** is equal to minimizing $2D_{JS}(\mathcal{D}_{S'}\|\mathcal{D}_{T'}) - 2\log 2$, where $D_{JS}$ is the Jensen-Shanon (JS) divergence.*

Compared to Proposition 2 in (Acuna et al., 2021) that adversarial training in DANN is equal to minimize $2D_{JS}(\mathcal{D}_S\|\mathcal{D}_T) - 2\log 2$. The only difference here is the mixed distributions $\mathcal{D}_{S'}, \mathcal{D}_{T'}$, which allows more flexible control on divergence minimization. For example, when $\gamma = 1$, $\mathcal{D}_{S'} = \mathcal{D}_S, \mathcal{D}_{T'} = \mathcal{D}_T$ which is the same as the original adversarial training; when $\gamma = 0.5$, $\mathcal{D}_{S'} = \mathcal{D}_{T'} = 0.5(\mathcal{D}_S + \mathcal{D}_T)$ and $D_{JS}(\mathcal{D}_{S'}\|\mathcal{D}_{T'}) = 0$, which means that this term will not supply gradients during training and the training process will convergence like ERM. In other words, $\gamma$ controls the tradeoff between algorithm convergence and adversarial divergence minimization. One main argue that adjusting the tradeoff weight $\lambda$ can also balance AT and ERM, however, $\lambda$ can only adjust the gradient contribution of AT part, *i.e.,* $2\lambda \nabla D_{JS}(\mathcal{D}_S, \mathcal{D}_T)$ and cannot affect the training dynamic of $D_{JS}(\mathcal{D}_S, \mathcal{D}_T)$. For example, when $\mathcal{D}_S, \mathcal{D}_T$ have disjoint support, $\nabla D_{JS}(\mathcal{D}_S, \mathcal{D}_T)$ is always zero no matter what $\lambda$ is given. On the contrary, the proposed technique smooths the optimization distribution $\mathcal{D}_S, \mathcal{D}_T$ of AT, making the whole training process more stable, but controlling $\lambda$ cannot do. In the experimental section, we show that in some benchmarks, the model cannot converge even if the tradeoff weight is small enough, however, when ELS is applied, DANN+ELS attains superior results and without the need for small tradeoff weights or small learning rate.

As shown in (Goodfellow, 2016), GANs always use a technique called *one-sided label smoothing*, which is a simple modification of the label smoothing technique and only replaces the target for real examples with a value slightly less than one, such as 0.9. Here we connect one-sided label smoothing to JS divergence and seek the difference between native and one-sided label smoothing techniques. See Appendix A.2 for proof and analysis. We further extend the above theoretical analysis to multi-domain settings, *e.g.,* domain generalization, and multi-source GANs (Trung Le et al., 2019) (See Proposition 3 in Appendix A.3 for detailed proof and analysis.). We find that with ELS, a flexible control on algorithm convergence and divergence minimization tradeoffs can be attained.

### 3.2 TRAINING STABILITY

**Noise injection for extending distribution supports.** The main source of training instability of GANs is the real and the generated distributions have disjoint supports or lie on low dimensional

manifolds (Arjovsky & Bottou, 2017; Roth et al., 2017). Adding noise from an arbitrary distribution to the data is shown to be able to extend the support of both distributions (Jenni & Favaro, 2019; Arjovsky & Bottou, 2017; Sønderby et al., 2016) and will protect the discriminator against measure 0 adversarial examples (Jenni & Favaro, 2019), which result in stable and well-behaved training. Environment label smoothing can be viewed as a kind of noise injection, *e.g.,* in Proposition 1, $\mathcal{D}_{S'} = \mathcal{D}_T + \gamma(\mathcal{D}_S - \mathcal{D}_T)$ where the noise is $\gamma(\mathcal{D}_S - \mathcal{D}_T)$ and the two distributions will be more likely to have joint supports.

**ELS relieves the gradient vanishing phenomenon.** As shown in Section 3.1, the adversarial target is approximating KL or JS divergence, and when the discriminator is not optimal, a such approximation is inaccurate. We show that in vanilla DANN, as the discriminator gets better, the gradient passed from discriminator to the encoder vanishes (Proposition 4 and Proposition 5). Namely, *either the approximation is inaccurate, or the gradient vanishes*, which will make adversarial training extremely hard (Arjovsky & Bottou, 2017). Incorporating ELS is shown able to relieve the gradient vanishing phenomenon when the discriminator is close to the optimal one and stabilizes the training process.

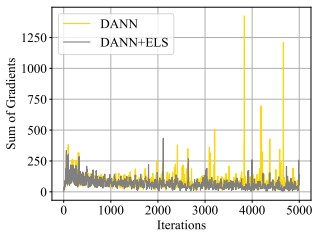

**ELS serves as a data-driven regularization and stabilizes the oscillatory gradients.** Gradients of the encoder with respect to adversarial loss remain highly oscillatory in native DANN, which is an important reason for the instability of adversarial training (Mescheder et al., 2018). Figure 2 shows the gradient dynamics throughout the training process, where the PACS dataset is used as an example. With ELS, the gradient brought by the adversarial loss is smoother and more stable. The benefit is theoretically supported in Section A.6, where applying ELS is shown similar to adding a regularization term on discriminator parameters, which stabilizes the supplied gradients compared to the vanilla adversarial loss.

Figure 2: The sum of gradients provided to the encoder by the adversarial loss.

### 3.3 ELS MEETS NOISY LABELS

To analyze the benefits of ELS when noisy labels exist, we adopt the symmetric noise model (Kim et al., 2019). Specifically, given two environments with a high-dimensional feature $x$ and environment label $y \in \{0, 1\}$, assume that noisy labels $\tilde{y}$ are generated by random noise transition with noise rate $e = P(\tilde{y} = 1|y = 0) = P(\tilde{y} = 0|y = 1)$. Denote $f := \hat{h} \circ g$, $\ell$ the cross-entropy loss and $\tilde{y}^\gamma$ the smoothed noisy label, then minimizing the smoothed loss with noisy labels can be converted to

$$
\begin{aligned}
\min_f \mathbb{E}_{(x,\tilde{y})\sim\tilde{\mathcal{D}}}[\ell(f(x), \tilde{y}^\gamma)] &= \min_f \mathbb{E}_{(x,\tilde{y})\sim\tilde{\mathcal{D}}}\left[\gamma\ell(f(x), \tilde{y}) + (1-\gamma)\ell(f(x), 1-\tilde{y})\right] \\
&= \min_f \mathbb{E}_{(x,y)\sim\mathcal{D}}[\ell(f(x), y^{\gamma^*})] + (\gamma^* - \gamma - e + 2\gamma e)\mathbb{E}_{(x,y)\sim\mathcal{D}}[\ell(f(x), 1-y) - \ell(f(x), y)]
\end{aligned}
\tag{4}
$$

where $\gamma^*$ is the optimal smooth parameter that makes the classifier return the best performance on unseen clean data (Wei et al., 2022). The first term in Equ. (4) is the risk under the clean label. The influence of both noisy labels and ELS are reflected in the last term of the Equ. (4). $\mathbb{E}_{(x,y)\sim\mathcal{D}}[\ell(f(x), 1-y) - \ell(f(x), y)]$ is the opposite of the optimization process as we expect. Without label smoothing, the weight will be $\gamma^* - 1 + e$ and a high noisy rate $e$ will let this harmful term contributes more to our optimization. On the contrary, by choosing a smooth parameter $\gamma = \frac{\gamma^* - e}{1 - 2e}$, the second term will be removed. For example, if $e = 0$, the best smooth parameter is just $\gamma^*$.

### 3.4 EMPIRICAL GAP AND PARAMETERIZATION GAP

Propositions in Section 3.1 and Section 3.2 are based on two unrealistic assumptions. (i) Infinite data samples, and (ii) the discriminator is optimized without a constraint, namely, the discriminator is optimized over infinite-dimensional space. In practice, only empirical distributions with finite samples are observed and the discriminator is always constrained to smaller classes such as neural networks (Goodfellow et al., 2014) or reproducing kernel Hilbert spaces (RKHS) (Li et al., 2017a). Besides, as shown in (Arora et al., 2017; Schäfer et al., 2019), JS divergence has a large empirical gap, *e.g., let $\mathcal{D}_\mu, \mathcal{D}_\nu$ be uniform Gaussian distributions $\mathcal{N}(0, \frac{1}{d}I)$, and $\hat{\mathcal{D}}_\mu, \hat{\mathcal{D}}_\nu$ be empirical versions of $\mathcal{D}_\mu, \mathcal{D}_\nu$ with $n$ examples. Then we have $|d_{JS}(\mathcal{D}_\mu, \mathcal{D}_\nu) - d_{JS}(\hat{\mathcal{D}}_\mu, \hat{\mathcal{D}}_\nu)| = \log 2$ with high probability.* Namely, the empirical divergence cannot reflect the true distribution divergence.

A natural question arises: "Given finite samples to multi-domain AT over finite-dimensional parameterized space, whether the expectation over the empirical distribution converges to the expectation over the true distribution?". In this subsection, we seek to answer this question by analyzing the *empirical gap and parameterization gap*, which is $|d_{\hat{h},g}(\mathcal{D}_1,\ldots,\mathcal{D}_M) - d_{\hat{h},g}(\hat{\mathcal{D}}_1,\ldots,\hat{\mathcal{D}}_M)|$, where $\hat{D}_i$ is the empirical distribution of $\mathcal{D}_i$ and $\hat{h}$ is constrained. We first show that, let $\mathcal{H}$ be a hypothesis class of VC dimension $d$, then for any $\delta \in (0,1)$, with probability at least $1-\delta$, the gap is less than $4\sqrt{(d\log(2n^*) + \log 2/\delta)/n^*}$, where $n^* = \min(n_1,\ldots,n_M)$ and $n_i$ is the number of samples in $\mathcal{D}_i$ (Appendix A.8). The above analysis is based on $\mathcal{H}$ divergence and the VC dimension; we further analyze the gap when the discriminator is constrained to the Lipschitz continuous and build a connection between the gap and the model parameters. Specifically, suppose that each $\hat{h}_i$ is $L$-Lipschitz with respect to the parameters and use $p$ to denote the number of parameters of $\hat{h}_i$. Then given a universal constant $c$ such that when $n^* \geq cpM \log(Lp/\epsilon)/\epsilon$, we have with probability at least $1-\exp(-p)$, the gap is less than $\epsilon$ (Appendix A.9). Although the analysis cannot support the benefits of ELS, as far as we know, it is the first attempt to study the empirical and parameterization gap of multi-domain AT.

## 3.5 NON-ASYMPTOTIC CONVERGENCE

As mentioned in Section 3.4, the analysis in Section 3.1 and Section 3.2 assume the optimal discriminator can be obtained, which implies that both the hypothesis set has infinite modeling capacity and the training process can converge to the optimal result. If the objective of AT is convex-concave, then many works can support the global convergence behaviors (Nowozin et al., 2016; Yadav et al., 2017). However, the convex-concave assumption is too unrealistic to hold true (Nie & Patel, 2020; Nagarajan & Kolter, 2017), namely, the updates of DAT are no longer guaranteed to converge. In this section, we focus on the local convergence behaviors of DAT of points near the equilibrium. Specifically, we focus on the non-asymptotic convergence, which is shown able to more precisely reveal the convergence of the dynamic system than the asymptotic analysis (Nie & Patel, 2020).

We build a toy example to help us understand the convergence of DAT. Denote $\eta$ the learning rate, $\gamma$ the parameter for ELS, and $c$ a constant. We conclude our theoretical results (which are detailed in Appendix A.10): (1) Simultaneous Gradient Descent (GD) DANN, which trains the discriminator and encoder simultaneously, has no guarantee of the non-asymptotic convergence. (2) If we train the discriminator $n_d$ times once we train the encoder $n_e$ times, the resulting alternating Gradient Descent (GD) DANN could converge with a sublinear convergence rate only when the $\eta \leq \frac{4}{\sqrt{n_d n_e} c}$. Such results support the importance of alternating GD training, which is commonly used during DANN implementation (Gulrajani & Lopez-Paz, 2021). (3) Incorporate ELS into alternating GD speeds up the convergence rate by a factor $\frac{1}{2\gamma-1}$, that is, when $\eta \leq \frac{4}{\sqrt{n_d n_e} c} \frac{1}{2\gamma-1}$, the model could converge.

**Remark.** In the above analysis, we made some assumptions *e.g.,* in Section 3.5, we assume the algorithms are initialized in a neighborhood of a unique equilibrium point, and in Section 3.4 we assume that the NN is L-Lipschitz. These assumptions may not hold in practice, and they are computationally hard to verify. To this end, we empirically support our theoretical results, namely, verifying the benefits to convergence, training stability, and generalization results in the next section.

## 4 EXPERIMENTS

To demonstrate the effectiveness of our ELS, in this section, we select a broad range of tasks (in Table 1), which are *image classification*, *image retrieval*, *neural language processing*, *genomics*, *graph*, and *sequential prediction tasks*. Our target is to include benchmarks with (i) various numbers of domains (from 3 to $120,084$); (ii) various numbers of classes (from 2 to $18,530$); (iii) various dataset sizes (from $3,200$ to $448,000$); (iv) various dimensionalities and backbones (Transformer, ResNet, MobileNet, GIN, RNN). See Appendix C for full details of all experimental settings, including dataset details, hyper-parameters, implementation details, and model structures. We conduct all the experiments on a machine with i7-8700K, 32G RAM, and four GTX2080ti. All experiments are repeated 3 times with different seeds and the full experimental results can be found in the appendix.

### 4.1 NUMERICAL RESULTS ON DIFFERENT SETTINGS AND BENCHMARKS

**Domain Generalization and Domain Adaptation on Image Classification Tasks.** We first incorporate ELS into SDAT, which is a variant of the DAT method and achieves the state-of-the-art

Table 1: **A summary on evaluation benchmarks.** Wg. acc. denotes worst group accuracy, 10 %/ acc. denotes 10th percentile accuracy. GIN (Xu et al., 2018) denotes Graph Isomorphism Networks, and CRNN (Gagnon-Audet et al., 2022) denotes convolutional recurrent neural networks.

| Task | Dataset | Domains | Classes | Metric | Backbone | # Data Examples |
|---|---|---|---|---|---|---|
| Images Classification | Rotated MNIST | 6 rotated angles | 10 | Avg. acc. | MNIST ConvNet | 70,000 |
| | PACS | 4 image styles | 7 | Avg. acc. | ResNet50 | 9,991 |
| | VLCS | 4 image styles | 5 | Avg. acc. | ResNet50 | 10,729 |
| | Office-31 | 3 image styles | 31 | Avg. acc. | ResNet50/ResNet18 | 4,110 |
| | Office-Home | 4 image styles | 65 | Avg. acc. | ResNet50/ViT | 15,500 |
| | Rotating MNIST | 8 rotated angles | 10 | Avg. acc. | EncoderSTN | 60,000 |
| Image Retrieval | MS | 5 locations | 18,530 | mAP, Rank $m$ | MobileNet×1.4 | 121,738 |
| Neural Language Processing | CivilComments | 8 demographic groups | 2 | Avg/Wg acc. | DistillBERT | 448,000 |
| | Amazon | 7676 reviewers | 5 | 10 %/Avg/Wg acc. | DistillBERT | 100,124 |
| Genomics and Graph | RxRx1 | 51 experimental batch | 1139 | Wg/Avg/Test ID acc. | ResNet-50 | 125,510 |
| | OGB-MolPCBA | 120,084 molecular scaffold | 128 | Avg. acc. | GIN | 437,929 |
| Sequential Prediction | Spurious-Fourier | 3 spurious correlations | 2 | Avg. acc. | LSTM | 12,000 |
| | HHAR | 5 smart devices | 6 | Avg. acc. | Deep ConvNets | 13,674 |

Table 2: **The domain adaptation accuracies (%) on Office-31**. ↑ denotes improvement of a method with ELS compared to that wo/ ELS.

| | A - W | D - W | W - D | A - D | D - A | W - A | Avg |
|---|---|---|---|---|---|---|---|
| | | | | **ResNet18** | | | |
| ERM (Vapnik, 1999) | 72.2 | 97.7 | 100.0 | 72.3 | 61.0 | 59.9 | 77.2 |
| DANN (Ganin et al., 2016) | 84.1 | 98.1 | 99.8 | 81.3 | 60.8 | 63.5 | 81.3 |
| DANN+ELS | 85.5 | 99.1 | 100.0 | 82.7 | 62.1 | 64.5 | 82.4 |
| ↑ | 1.4 | 1.0 | 0.2 | 1.4 | 1.3 | 1.1 | 1.1 |
| SDAT (Rangwani et al., 2022) | 87.8 | 98.7 | 100.0 | 82.5 | 73.0 | 72.7 | 85.8 |
| SDAT+ELS | **88.9** | **99.3** | **100.0** | **83.9** | **74.1** | **73.9** | **86.7** |
| ↑ | 1.1 | 0.5 | 0.0 | 1.4 | 1.1 | 1.2 | 0.9 |
| | | | | **ResNet50** | | | |
| ERM (Vapnik, 1999) | 75.8 | 95.5 | 99.0 | 79.3 | 63.6 | 63.8 | 79.5 |
| ADDA (Tzeng et al., 2017) | 94.6 | 97.5 | 99.7 | 90.0 | 69.6 | 72.5 | 87.3 |
| CDAN (Long et al., 2018) | 93.8 | 98.5 | 100.0 | 89.9 | 73.4 | 70.4 | 87.7 |
| MCC (Jin et al., 2020) | 94.1 | 98.4 | 99.8 | **95.6** | 75.5 | 74.2 | 89.6 |
| DANN (Ganin et al., 2016) | 91.3 | 97.2 | 100.0 | 84.1 | 72.9 | 73.6 | 86.5 |
| DANN+ELS | 92.2 | 98.5 | 100.0 | 85.9 | 74.3 | 75.3 | 87.7 |
| ↑ | 0.9 | 1.3 | 0.0 | 1.8 | 1.4 | 1.7 | 1.2 |
| SDAT (Rangwani et al., 2022) | 92.7 | 98.9 | 100.0 | 93.0 | 78.5 | 75.7 | 89.8 |
| SDAT+ELS | **93.6** | **99.0** | **100.0** | 93.4 | **78.7** | **77.5** | **90.4** |
| ↑ | 0.9 | 0.1 | 0.0 | 0.4 | 0.2 | 1.8 | 0.6 |

performance on the Office-Home dataset. Table 2 and Table 4 show that with the simple smoothing trick, the performance of SDAT is consistently improved, and on many of the domain pairs, the improvement is greater than $1\%$. Besides, the ELS can also bring consistent improvement both with ResNet-18, ResNet-50, and ViT backbones. The average domain generalization results on other benchmarks are shown in Table 3. We observe consistent improvements achieved by DANN+ELS compared to DANN and the average accuracy on VLCS achieved by DANN+ELS ($81.5\%$) clearly outperforms all other methods. See Appendix D.1 for *Multi-Source Domain Generalization* performance, *DG performance on Rotated MNIST* and on *Image Retrieval* benchmarks.

**Domain Generalization with Partial Environment labels.** One of the main advantages brought by ELS is the robustness to environment label noise. As shown in Figure 3(a), when all environment labels are known (GT), DANN+ELS is slightly better than DANN. When partial environment labels are known, for example, $30\%$ means the environment labels of $30\%$ training data are known and others are annotated differently than the ground truth annotations, DANN+ELS outperform DANN by a large margin (more than $5\%$ accuracy when only $20\%$ correct environment labels are given). Besides, we further assume the total number of environments is also unknown and the environment number is generated randomly. M=2 in Figure 3(a) means we partition all the training data randomly into two domains, which are used for training then. With random environment partitions, DANN+ELS consistently beats DANN by a large margin, which verifies that the smoothness of the discrimination loss brings significant robustness to environment label noise for DAT.

**Continuously Indexed Domain Adaptation.** We compare DANN+ELS with state-of-the-art continuously indexed domain adaptation methods. Table 5 compares the accuracy of various methods. DANN shows an inferior performance to CIDA. However, with ELS, DANN+ELS boosts the generalization performance by a large margin and beats the SOTA method CIDA (Wang et al., 2020). We also

Table 3: The domain generalization accuracies (%) on VLCS, and PACS. ↑ denotes improvement of DANN+ELS compared to DANN.

| Algorithm | PACS | | | | | VLCS | | | | |
|---|---|---|---|---|---|---|---|---|---|---|
| | A | C | P | S | Avg | C | L | S | V | Avg |
| ERM (VAPNIK, 1999) | 87.8 ± 0.4 | 82.8 ± 0.5 | 97.6 ± 0.4 | 80.4 ± 0.6 | 87.2 | 97.7 ± 0.3 | 65.2 ± 0.4 | 73.2 ± 0.7 | 75.2 ± 0.4 | 77.8 |
| IRM (ARJOVSKY ET AL., 2019) | 85.7 ± 1.0 | 79.3 ± 1.1 | 97.6 ± 0.4 | 75.9 ± 1.0 | 84.6 | 97.6 ± 0.5 | 64.7 ± 1.1 | 69.7 ± 0.5 | 76.6 ± 0.7 | 77.2 |
| DANN (GANIN ET AL., 2016) | 85.4 ± 1.2 | 83.1 ± 0.8 | 96.3 ± 0.4 | 79.6 ± 0.8 | 86.1 | 98.6 ± 0.8 | 73.2 ± 1.1 | 72.8 ± 0.8 | 78.8 ± 1.2 | 80.8 |
| ARM (Zhang et al., 2021b) | 85.0 ± 1.2 | 81.4 ± 0.2 | 95.9 ± 0.3 | 80.9 ± 0.5 | 85.8 | 97.6 ± 0.6 | 66.5 ± 0.3 | 72.7 ± 0.6 | 74.4 ± 0.7 | 77.8 |
| Fisher (Rame et al., 2021) | —— | —— | —— | —— | 86.9 | —— | —— | —— | —— | 76.2 |
| DDG (Zhang et al., 2021a) | **88.9 ± 0.6** | **85.0 ± 1.9** | 97.2 ± 1.2 | **84.3 ± 0.7** | **88.9** | **99.1 ± 0.6** | 66.5 ± 0.3 | 73.3 ± 0.6 | **80.9 ± 0.6** | 80.0 |
| DANN+ELS | 87.8 ± 0.8 | 83.8 ± 1.6 | 97.1 ± 0.4 | 81.4 ± 1.3 | 87.5 | **99.1 ± 0.3** | 73.2 ± 1.1 | **73.8 ± 0.9** | 79.9 ± 0.9 | **81.5** |
| ↑ | 2.4 | 0.7 | 0.8 | 1.8 | 1.4 | 0.5 | 0 | 1 | 1.1 | 0.7 |

Table 4: **Accuracy (%) on Office-Home for unsupervised DA** (with ResNet-50 and ViT backbone). SDAT+ELS outperforms other SOTA DA techniques and improves SDAT consistently.

| Method | Backbone | A-C | A-P | A-R | C-A | C-P | C-R | P-A | P-C | P-R | R-A | R-C | R-P | Avg |
|---|---|---|---|---|---|---|---|---|---|---|---|---|---|---|
| ResNet-50 (He et al., 2016) | ResNet-50 | 34.9 | 50.0 | 58.0 | 37.4 | 41.9 | 46.2 | 38.5 | 31.2 | 60.4 | 53.9 | 41.2 | 59.9 | 46.1 |
| DANN (Ganin et al., 2016) | | 45.6 | 59.3 | 70.1 | 47.0 | 58.5 | 60.9 | 46.1 | 43.7 | 68.5 | 63.2 | 51.8 | 76.8 | 57.6 |
| CDAN (Long et al., 2018) | | 49.0 | 69.3 | 74.5 | 54.4 | 66.0 | 68.4 | 55.6 | 48.3 | 75.9 | 68.4 | 55.4 | 80.5 | 63.8 |
| MMD (Zhang et al., 2019) | | 54.9 | 73.7 | 77.8 | 60.0 | 71.4 | 71.8 | 61.2 | 53.6 | 78.1 | 72.5 | 60.2 | 82.3 | 68.1 |
| f-DAL (Acuna et al., 2021) | | 56.7 | 77.0 | 81.1 | 63.1 | 72.2 | 75.9 | 64.5 | 54.4 | 81.0 | 72.3 | 58.4 | 83.7 | 70.0 |
| SRDC (Tang et al., 2020) | | 52.3 | 76.3 | 81.0 | **69.5** | 76.2 | **78.0** | **68.7** | 53.8 | 81.7 | **76.3** | 57.1 | 85.0 | 71.3 |
| SDAT (Rangwani et al., 2022) | | 57.8 | 77.4 | 82.2 | 66.5 | 76.6 | 76.2 | 63.3 | 57.0 | **82.2** | 75.3 | 62.6 | 85.2 | 71.8 |
| SDAT+ELS | | **58.2** | **79.7** | **82.5** | 67.5 | **77.2** | 77.2 | 64.6 | **57.9** | **82.2** | 75.4 | **63.1** | **85.5** | **72.6** |
| ↑ | | 0.4 | 2.3 | 0.3 | 1.0 | 0.6 | 1.0 | 1.3 | 0.9 | 0.0 | 0.1 | 0.5 | 0.3 | 0.8 |
| TVT (Yang et al., 2021) | ViT | **74.9** | 86.6 | 89.5 | 82.8 | 87.9 | 88.3 | 79.8 | 71.9 | 90.1 | 85.5 | 74.6 | 90.6 | 83.6 |
| CDAN (Long et al., 2018) | | 62.6 | 82.9 | 87.2 | 79.2 | 84.9 | 87.1 | 77.9 | 63.3 | 88.7 | 83.1 | 63.5 | 90.8 | 79.3 |
| SDAT (Rangwani et al., 2022) | | 70.8 | 87.0 | 90.5 | 85.2 | 87.3 | 89.7 | 84.1 | 70.7 | 90.6 | 88.3 | 75.5 | 92.1 | 84.3 |
| SDAT+ELS | | 72.1 | **87.3** | **90.6** | **85.2** | **88.1** | **89.7** | **84.1** | 70.7 | **90.8** | **88.4** | 76.5 | **92.1** | **84.6** |
| ↑ | | 1.3 | 0.3 | 0.1 | 0.0 | 0.8 | 0.0 | 0.0 | 0.0 | 0.2 | 0.1 | 1.0 | 0.0 | 0.3 |

visualize the classification results on Circle Dataset (See Appendix C.1.1 for dataset details). Figure 5 shows that the representative DA method (ADDA) performs poorly when asked to align domains with continuous indices. However, the proposed DANN+ELS can get a near-optimal decision boundary.

**Generalization results on other structural datasets and Sequential Datasets.** Table 6 shows the generalization results on NLP datasets, and Table 7, 14 show the results on genomics datasets. DANN+ELS bring huge performance improvement on most of the evaluation metrics, *e.g.,* 4.17% test worst-group accuracy on CivilComments, 3.79% test ID accuracy on RxRx1, and 3.13% test accuracy on OGB-MolPCBA. Generalization results on sequential prediction tasks are shown in Table 15 and Table 18, where DANN works poorly but DANN+ELS brings consistent improvement and beats all baselines on the Spurious-Fourier dataset.

### 4.2 INTERPRETATION AND ANALYSIS

**To choose the best $\gamma$.** Figure 3(b) visualizes the best $\gamma$ values in our experiments. For datasets like PACS and VLCS, where each domain will be set as a target domain respectively and has one best $\gamma$, we calculate the mean and standard deviation of all these $\gamma$ values. Our main observation is that, as the number of domains increases, the optimal $\gamma$ will also decrease, which is intuitive because more domains mean that the discriminator is more likely to overfit and thus needs a lower $\gamma$ to solve the problem. An interesting thing is that in Figure 3(b), PACS and VLCS both have 4 domains,

Table 5: **Rotating MNIST accuracy (%) at the source domain and each target domain.** $X°$ denotes the domain whose images are Rotating by $[X°, X° + 45°]$.

| Algorithm | Rotating MNIST | | | | | | | | |
|---|---|---|---|---|---|---|---|---|---|
| | $0°$(Source) | $45°$ | $90°$ | $135°$ | $180°$ | $225°$ | $270°$ | $315°$ | Average |
| ERM (Vapnik, 1999) | 99.2 | 79.7 | 26.8 | 31.6 | 35.1 | 37.0 | 28.6 | 76.2 | 45.0 |
| ADDA (Tzeng et al., 2017) | 97.6 | 70.7 | 22.2 | 32.6 | 38.2 | 31.5 | 20.9 | 65.8 | 40.3 |
| DANN (Ganin et al., 2016) | 98.4 | **81.4** | 38.9 | 35.4 | 40.0 | 43.4 | 48.8 | 77.3 | 52.1 |
| CIDA (Wang et al., 2020) | **99.5** | 80.0 | 33.2 | **49.3** | **50.2** | **51.7** | **54.6** | **81.0** | 57.1 |
| DANN+ELS | 98.4 | **81.4** | 55.0 | 39.9 | 43.7 | 45.9 | 53.7 | 78.7 | **62.1** |
| ↑ | 0.0 | 0.0 | 16.1 | 4.5 | 3.7 | 2.5 | 4.9 | 1.4 | 10.0 |

Table 6: **Domain generalization performance on neural language datasets.** The backbone is *DistillBERT-base-uncased* and all results are reported over 3 random seed runs.

| Algorithm | Val Avg Acc | Test Avg Acc | Val 10% Acc | Test 10% Acc | Val Worst-group acc | Test Worst-group acc |
|---|---|---|---|---|---|---|
| | | | **Amazon-Wilds** | | | |
| ERM (Vapnik, 1999) | **72.7 ± 0.1** | **71.9 ± 0.1** | **55.2 ± 0.7** | 53.8 ± 0.8 | 20.3 ± 0.1 | 4.2 ± 0.2 |
| Group DRO (Sagawa et al., 2019) | 70.7 ± 0.6 | 70.0 ± 0.6 | 54.7 ± 0.0 | 53.3 ± 0.0 | **54.2 ± 0.3** | 6.3 ± 0.2 |
| CORAL (Sun & Saenko, 2016) | 72.0 ± 0.3 | 71.1 ± 0.3 | 54.7 ± 0.0 | 52.9 ± 0.8 | 30.0 ± 0.2 | 6.1 ± 0.1 |
| IRM (Arjovsky et al., 2019) | 71.5 ± 0.3 | 70.5 ± 0.3 | 54.2 ± 0.8 | 52.4 ± 0.8 | 32.2 ± 0.8 | 5.3 ± 0.2 |
| Reweight | 69.1 ± 0.5 | 68.6 ± 0.6 | 52.1 ± 0.2 | 52.0 ± 0.0 | 34.9 ± 1.2 | 9.1 ± 0.4 |
| DANN (Ganin et al., 2016) | 72.1 ± 0.2 | 71.3 ± 0.1 | 54.6 ± 0.0 | 52.9 ± 0.6 | 4.4 ± 1.3 | 8.0 ± 0.0 |
| DANN+ELS | 72.3 ± 0.1 | 71.5 ± 0.1 | 54.7 ± 0.1 | **53.8 ± 0.0** | 4.9 ± 0.6 | **9.4 ± 0.0** |
| ↑ | 0.2 | 0.2 | 0.1 | 0.9 | 0.5 | 1.4 |

| Algorithm | Val Avg Acc | Val Worst-Group Acc | Test Avg Acc | Test Worst-Group Acc |
|---|---|---|---|---|
| | | **CivilComments-Wilds** | | |
| Group DRO (Sagawa et al., 2019) | 90.4 ± 0.4 | 65.0 ± 3.8 | 90.2 ± 0.3 | **69.1 ± 1.8** |
| Reweighted | 90.0 ± 0.7 | 63.7 ± 2.7 | 89.8 ± 0.8 | 66.6 ± 1.6 |
| IRM (Arjovsky et al., 2019) | 89.0 ± 0.7 | 65.9 ± 2.8 | 88.8 ± 0.7 | 66.3 ± 2.1 |
| ERM (Vapnik, 1999) | **92.3 ± 0.2** | 50.5 ± 1.9 | **92.2 ± 0.1** | 56.0 ± 3.6 |
| DANN (Ganin et al., 2016) | 87.0 ± 0.3 | 64.0 ± 2.0 | 87.0 ± 0.3 | 61.7 ± 2.2 |
| DANN+ELS | 88.5 ± 0.4 | **65.9 ± 1.1** | 88.4 ± 0.4 | 66.0 ± 2.2 |
| ↑ | 1.4 | 1.9 | 1.4 | 4.3 |

Table 7: Domain generalization performance on genomics dataset, RxRx1.

| Algorithm | Val Acc | Test ID Acc | Test Acc | Val Worst-Group Acc | Test ID Worst-Group Acc | Test Worst-Group Acc |
|---|---|---|---|---|---|---|
| | | | **RxRx1-Wilds** | | | |
| ERM (Vapnik, 1999) | **19.4 ± 0.2** | **35.9 ± 0.4** | **29.9 ± 0.4** | — | — | — |
| Group DRO (Sagawa et al., 2019) | 15.2 ± 0.1 | 28.1 ± 0.3 | 23.0 ± 0.3 | — | — | — |
| IRM (Arjovsky et al., 2019) | 5.6 ± 0.4 | 9.9 ± 1.4 | 8.2 ± 1.1 | 0.8 ± 0.2 | 1.9 ± 0.4 | 1.5 ± 0.2 |
| DANN (Ganin et al., 2016) | 12.7 ± 0.2 | 22.9 ± 0.1 | 19.2 ± 0.1 | **1.0 ± 0.1** | 4.6 ± 0.4 | 3.6 ± 0.0 |
| DANN+ELS | 14.1 ± 0.1 | 26.7 ± 0.1 | 21.2 ± 0.2 | **1.1 ± 0.1** | **7.2 ± 0.3** | **4.2 ± 0.1** |
| ↑ | 1.4 | 3.8 | 2 | 0.1 | 2.6 | 0.6 |

but VLCS needs a higher $\gamma$. Figure 6 shows that images from different domains in PACS are of great visual difference and can be easily discriminated. In contrast, domains in VLCS do not show significant visual differences, and it is hard to discriminate which domain one image belongs to. The discrimination difficulty caused by this inter-domain distinction is another important factor affecting the selection of $\gamma$.

**Annealing $\gamma$.** To achieve better generalization performance and avoid troublesome parametric searches, we propose to gradually decrease $\gamma$ as training progresses, specifically, $\gamma = 1.0 - \frac{M-1}{M}\frac{t}{T}$, where $t, T$ are the current training step and the total training steps. Figure 3(c) shows that annealing $\gamma$ achieves a comparable or even better generalization performance than fine-grained searched $\gamma$.

**Empirical Verification of our theoretical results.** We use the PACS dataset as an example to empirically support our theoretical results, namely verifying the benefits to convergence, training stability, and generalization results. In Figure 4, 'A' is set as the target domain and other domains

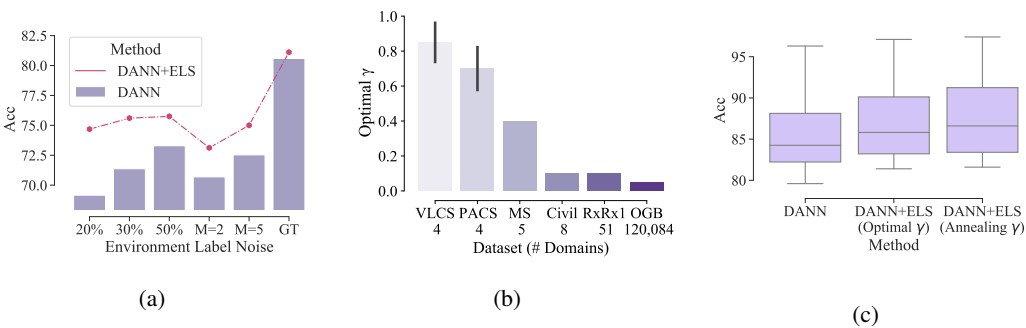

(a)       (b)       (c)

Figure 3: (a) Generalization performance of DANN+ELS compared to DANN with partial correct environment label on the PACS dataset ($P$ as target domain). (b) The best $\gamma$ for each dataset. Civil is the CivilComments dataset and OGB is the OGB-MolPCBA dataset. (c) Average generalization accuracy on the PACS dataset with different smoothing policies.

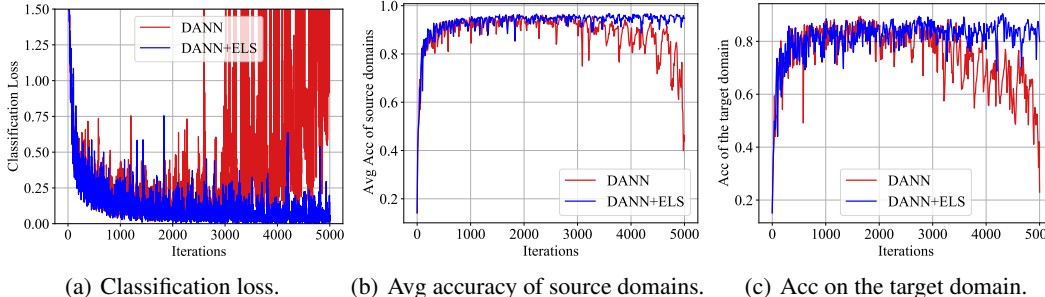

| (a) Classification loss. | (b) Avg accuracy of source domains. | (c) Acc on the target domain. |

Figure 4: **Training statistics on PACS datasets.** Alternating GD with $n_d = 5, n_e = 1$ is used. All other parameters setting are the same and only on the default hyperparameters and without the fine-grained parametric search.

as sources. Considering ELS, we can see that in all the experimental results, DANN+ELS with appropriate $\gamma$ attains high training stability, faster and stable convergence, and better performance compared to DANN. In comparison, the training dynamics of native DANN is highly oscillatory, especially in the middle and late stages of training.

## 5 RELATED WORKS

**Label Smoothing and Analysis** is a technique from the 1980s, and independently re-discovered by (Szegedy et al., 2016). Recently, label smoothing is shown to reduce the vulnerability of neural networks (Warde-Farley & Goodfellow, 2016) and reduce the risk of adversarial examples in GANs (Salimans et al., 2016). Several works seek to theoretically or empirically study the effect of label smoothing. (Chen et al., 2020) focus on studying the minimizer of the training error and finding the optimal smoothing parameter. (Xu et al., 2020) analyzes the convergence behaviors of stochastic gradient descent with label smoothing. However, as far as we know, no study focuses on the effect of label smoothing on the convergence speed and training stability of DAT.

**Domain Adversarial Training** (Ganin et al., 2016) using a domain discriminator to distinguish the source and target domains and the gradients of the discriminator to the encoder are reversed by the Gradient Reversal layer (GRL), which achieves the goal of learning domain invariant features. (Schoenauer-Sebag et al., 2019; Zhao et al., 2018) extend generalization bounds in DANN (Ganin et al., 2016) to multi-source domains and propose multisource domain adversarial networks. (Hu et al., 2021) incorporates the prototypical features into DAT to achieve semantic domain alignment. (Acuna et al., 2022) interprets the DAT framework through the lens of game theory and proposes to replace gradient descent with high-order ODE solvers. (Rangwani et al., 2022) finds that enforcing the smoothness of the classifier leads to better generalization on the target domain and presents Smooth Domain Adversarial Training (SDAT). The proposed method is orthogonal to existing DAT methods and yields excellent optimization properties theoretically and empirically.

For space limit, the related works about domain adaptation, domain generalization, and adversarial Training in GANs are in the appendix.

## 6 CONCLUSION

In this work, we propose a simple approach, *i.e.,* ELS, to optimize the training process of DAT methods from an environment label design perspective, which is orthogonal to most existing DAT methods. Incorporating ELS into DAT methods is empirically and theoretically shown to be capable of improving robustness to noisy environment labels, converge faster, attain more stable training and better generalization performance. As far as we know, our work takes a first step towards utilizing and understanding label smoothing for environmental labels. Although ELS is designed for DAT methods, reducing the effect of environment label noise and a soft environment partition may benefit all DG/DA methods, which is a promising future direction.

## 7 ACKNOWLEDGEMENT

This work was partially funded by the National Natural Science Foundation of China (Grant No. 62276256, 62076078), the Beijing Nova Program under Grant Z211100002121108, and the National Natural Science Foundation of China (62236010, 61721004, and U1803261)

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

# Appendix

CONTENTS

# A  PROOFS OF THEORETICAL STATEMENTS

The commonly used notations and their corresponding descriptions are concluded in Table 8.

## A.1  CONNECT ENVIRONMENT LABEL SMOOTHING TO JS DIVERGENCE MINIMIZATION

To complete the proofs, we begin by introducing some necessary definitions and assumptions.

**Definition 1.** *($\mathcal{H}$-divergence (Ben-David et al., 2006)). Given two domain distributions $\mathcal{D}_S, \mathcal{D}_T$ over $X$, and a hypothesis class $\mathcal{H}$, the $\mathcal{H}$-divergence between $\mathcal{D}_S, \mathcal{D}_T$ is*

$$d_{\mathcal{H}}(\mathcal{D}_S, \mathcal{D}_T) = 2 \sup_{h \in \mathcal{H}} |\mathbb{E}_{\mathbf{x} \sim \mathcal{D}_S}[h(\mathbf{x}) = 1] - \mathbb{E}_{\mathbf{x} \sim \mathcal{D}_T}[h(\mathbf{x}) = 1]| \tag{5}$$

**Definition 2.** *(Empirical $\mathcal{H}$-divergence (Ben-David et al., 2006).) For an symmetric hypothesis class $\mathcal{H}$, one can compute the empirical $\mathcal{H}$-divergence between two empirical distributions $\hat{\mathcal{D}}_S$ and $\hat{\mathcal{D}}_T$ by computing*

$$\hat{d}_{\mathcal{H}}(\hat{\mathcal{D}}_S, \hat{\mathcal{D}}_T) = 2 \left( 1 - \min_{h \in \mathcal{H}} \left[ \frac{1}{m} \sum_{i=1}^{m} I[h(\mathbf{x}_i) = 0] + \frac{1}{n} \sum_{i=1}^{n} I[h(\mathbf{x}_i) = 1] \right] \right), \tag{6}$$

*where $m, n$ is the number of data samples of $\hat{\mathcal{D}}_S$ and $\hat{\mathcal{D}}_T$ respectively and $I[a]$ is the indicator function which is 1 if predicate $a$ is true, and 0 otherwise.*

Vanilla DANN estimating the "min" part of Equ. (6) by a domain discriminator, that models the probability that a given input is from the source domain or the target domain. Specially, let the hypothesis $h$ be the composition of $h = \hat{h} \circ g$, where $\hat{h} \in \hat{\mathcal{H}}$ is a additional hypothesis and $g \in \mathcal{G}$ pushes forward the data samples to a representation space $\mathcal{Z}$. DANN (Ben-David et al., 2006) seeks to approximate the $\mathcal{H}$-divergence of Equ. (6) by

$$\max_{\hat{h} \in \hat{\mathcal{H}}} d_{\hat{h}, g}(\mathcal{D}_S, \mathcal{D}_T) = \max_{\hat{h} \in \hat{\mathcal{H}}} \mathbb{E}_{\mathbf{x}_s \sim \mathcal{D}_S} \log \hat{h} \circ g(\mathbf{x}_s) + \mathbb{E}_{\mathbf{x}_t \sim \mathcal{D}_T} \log \left( 1 - \hat{h} \circ g(\mathbf{x}_t) \right), \tag{7}$$

Table 8: Notations.

| Symbol | Description |
|---|---|
| $\mathcal{D}_S, \mathcal{D}_T, \mathcal{D}_i$ | Distributions for source domain, target domain, and domain $i$. |
| $\hat{\mathcal{D}}_S, \hat{\mathcal{D}}_T, \hat{\mathcal{D}}_i$ | Empirical distributions for source domain, target domain, and domain $i$. |
| $p_s, p_t, p_i$ | Density functions for source domain, target domain, and domain $i$. |
| $\mathbf{x}_s, \mathbf{x}_t, \mathbf{x}_i$ | Data samples from source domain, target domain, and domain $i$. |
| $\mathcal{D}_S^z, \mathcal{D}_T^z, \mathcal{D}_i^z$ | Feature distributions of $\mathcal{D}_S, \mathcal{D}_T, \mathcal{D}_i$ respectively, which is also termed $g \circ \mathcal{D}_S, g \circ \mathcal{D}_T, g \circ \mathcal{D}_i$. |
| $p_s^z, p_t^z, p_i^z$ | Density functions for $\mathcal{D}_S^z, \mathcal{D}_T^z, \mathcal{D}_i^z$ respectively. |
| $\mathbf{z}_s, \mathbf{z}_t, \mathbf{z}_i$ | Data samples from $\mathcal{D}_S^z, \mathcal{D}_T^z, \mathcal{D}_i^z$. |
| $\mathcal{H}, \hat{\mathcal{H}}, \mathcal{G}$ | Support sets for hypothesis, discriminator, and feature encoder. |
| $h, \hat{h}, \hat{h}^*, g$ | Hypothesis, discriminator, the optimal discriminator, and feature encoder. |
| $M, n_i$ | Number of training distributions, number of data samples in $\mathcal{D}_i$. |
| $\gamma$ | Hyper-parameter for the environment label smoothing. |
| $d_{\mathcal{H}}, \hat{d}_{\mathcal{H}}$ | $\mathcal{H}$-divergence and Empirical $\mathcal{H}$-divergence. |

where the sigmoid activate function is ignored for simplicity, $\hat{h} \circ g(\mathbf{x})$ is the prediction probability that $\mathbf{x}$ is belonged to $\mathcal{D}_S$ and $1 - \hat{h} \circ g(\mathbf{x})$ is the prediction probability that $\mathbf{x}$ is belonged to $\mathcal{D}_T$. Applying environment label smoothing, the target can be reformulated to

$$
\max_{\hat{h} \in \hat{\mathcal{H}}} d_{\hat{h},g,\gamma}(\mathcal{D}_S, \mathcal{D}_T) = \max_{\hat{h} \in \hat{\mathcal{H}}} \mathbb{E}_{\mathbf{x}_s \sim \mathcal{D}_S} \left[ \gamma \log \hat{h} \circ g(\mathbf{x}_s) + (1-\gamma) \log \left(1 - \hat{h} \circ g(\mathbf{x}_s)\right) \right] +
$$
$$
\mathbb{E}_{\mathbf{x}_t \sim \mathcal{D}_T} \left[ (1-\gamma) \log \hat{h} \circ g(\mathbf{x}_t) + \gamma \log \left(1 - \hat{h} \circ g(\mathbf{x}_t)\right) \right]
$$

(8)

When $\gamma \in \{0, 1\}$, Equ. (8) is equal to Equ. (7) and no environment label smoothing is applied. Then we prove the proposition 1

**Proposition 1.** *Suppose $\hat{h}$ the optimal domain classifier with no constraint and mixed distributions* $\begin{cases} \mathcal{D}_{S'} = \gamma \mathcal{D}_S + (1-\gamma)\mathcal{D}_T \\ \mathcal{D}_{T'} = \gamma \mathcal{D}_T + (1-\gamma)\mathcal{D}_S \end{cases}$ *with hyper-parameter $\gamma$, then $\max_{\hat{h} \in \hat{\mathcal{H}}} d_{\hat{h},g,\gamma}(\mathcal{D}_S, \mathcal{D}_T) = 2D_{JS}(\mathcal{D}_{S'} \| \mathcal{D}_{T'}) - 2\log 2$, where $D_{JS}$ is the Jensen-Shanon (JS) divergence.*

*Proof.* Denote the injected source/target density as $p_s^z := g \circ p_s, p_t^z := g \circ p_t$, where $p_s, p_t$ is the density of $\mathcal{D}_S, \mathcal{D}_T$ respectively. We can rewrite Equ. (8) as:

$$
d_{\hat{h},g,\gamma}(\mathcal{D}_S, \mathcal{D}_T) = \int_{\mathcal{Z}} p_s^z(\mathbf{z}) \log \left[ \gamma \log \hat{h}(\mathbf{z}) + (1-\gamma) \log \left(1 - \hat{h}(\mathbf{z})\right) \right] +
$$
$$
p_t^z(\mathbf{z}) \left[ (1-\gamma) \log \hat{h}(\mathbf{z}) + \gamma \log \left(1 - \hat{h}(\mathbf{z})\right) \right]
$$

(9)

We first take derivatives and find the optimal $\hat{h}^*$:

$$
\frac{\partial d_{\hat{h},g,\gamma}(\mathcal{D}_S, \mathcal{D}_T)}{\partial \hat{h}(\mathbf{z})} = p_s^z(\mathbf{z}) \left[ \gamma \frac{1}{\hat{h}(\mathbf{z})} + (1-\gamma) \frac{-1}{1-\hat{h}(\mathbf{z})} \right] + p_t^z(\mathbf{z}) \left[ (1-\gamma) \log \frac{1}{\hat{h}(\mathbf{z})} + \gamma \frac{-1}{1-\hat{h}(\mathbf{z})} \right] = 0
$$
$$
\Rightarrow p_s^z(\mathbf{z}) \left[ \gamma(1-\hat{h}(\mathbf{z})) - (1-\gamma)\hat{h}(\mathbf{z}) \right] + p_t^z(\mathbf{z}) \left[ (1-\gamma)(1-\hat{h}(\mathbf{z})) - \gamma\hat{h}(\mathbf{z}) \right] = 0
$$
$$
\Rightarrow p_s^z(\mathbf{z}) \left[ \gamma - \hat{h}(\mathbf{z}) \right] + p_t^z(\mathbf{z}) \left[ 1 - \gamma - \hat{h}(\mathbf{z}) \right] = 0
$$
$$
\Rightarrow \hat{h}^*(\mathbf{z}) = \frac{p_t^z(\mathbf{z}) + \gamma(p_s^z(\mathbf{z}) - p_t^z(\mathbf{z}))}{p_s^z(\mathbf{z}) + p_t^z(\mathbf{z})}
$$

(10)

For simplicity, we use $p_s, p_t$ denote $p_s^z(\mathbf{z}), p_t^z(\mathbf{z})$ respectively and ignore the $\int_{\mathcal{Z}}$. Plugging Equ. (10) into Equ. (8) we can get

$$
\max_{\hat{h} \in \hat{\mathcal{H}}} d_{\hat{h},g,\gamma}(\mathcal{D}_S, \mathcal{D}_T) = \int_{\mathcal{Z}} p_s \left[ \gamma \log \left[ \frac{p_t + \gamma(p_s - p_t)}{p_s + p_t} \right] + (1-\gamma) \log \left[ \frac{p_s + \gamma(p_t - p_s)}{p_s + p_t} \right] \right]
$$
$$
+ p_t \left[ (1-\gamma) \log \left[ \frac{p_t + \gamma(p_s - p_t)}{p_s + p_t} \right] + \gamma \log \left[ \frac{p_s + \gamma(p_t - p_s)}{p_s + p_t} \right] \right] d_{\mathbf{z}}
$$
$$
= \int_{\mathcal{Z}} \underbrace{p_s \log \frac{p_s + \gamma(p_t - p_s)}{p_s + p_t} + p_t \log \frac{p_t + \gamma(p_s - p_t)}{p_s + p_t}}_{①} +
$$
$$
\underbrace{p_s \gamma \log \frac{p_t + \gamma(p_s - p_t)}{p_s + \gamma(p_t - p_s)} + p_t \gamma \frac{p_s + \gamma(p_t - p_s)}{p_t + \gamma(p_s - p_t)}}_{②} d_{\mathbf{z}}
$$

(11)

Let $\begin{cases} p_{s'} = p_s + (1-\gamma)p_t \\ p_{t'} = p_t + (1-\gamma)p_s \end{cases}$ two distribution densities that are the convex combinations of $p_s, p_t$, we

have $\begin{cases} p_s = \frac{\gamma p_{s'} + (\gamma-1)p_{t'}}{2\gamma-1} \\ p_t = \frac{\gamma p_{t'} + (\gamma-1)p_{s'}}{2\gamma-1} \end{cases}$ , and $p_{s'} + p_{t'} = p_s + p_t$. Then ① in Equ. (11) can be rearranged to

$$
\begin{aligned}
& \frac{\gamma}{2\gamma-1}\left(p_{s'}\log\frac{p_{t'}}{p_{s'}+p_{t'}} + p_{t'}\log\frac{p_{s'}}{p_{s'}+p_{t'}}\right) + \frac{\gamma-1}{2\gamma-1}\left(p_{t'}\log\frac{p_{t'}}{p_{s'}+p_{t'}} + p_{s'}\log\frac{p_{s'}}{p_{s'}+p_{t'}}\right) \\
&= \frac{\gamma}{2\gamma-1}\left(p_{s'}\log\frac{p_{t'}}{p_{s'}+p_{t'}} + p_{s'}\log\frac{p_{s'}}{p_{t'}} - p_{s'}\log\frac{p_{s'}}{p_{t'}} + p_{t'}\log\frac{p_{s'}}{p_{s'}+p_{t'}}\right) + \frac{\gamma-1}{2\gamma-1}\left(p_{t'}\log\frac{p_{t'}}{p_{s'}+p_{t'}} + p_{s'}\log\frac{p_{s'}}{p_{s'}+p_{t'}}\right) \\
&= \left(p_{t'}\log\frac{p_{t'}}{p_{s'}+p_{t'}} + p_{s'}\log\frac{p_{s'}}{p_{s'}+p_{t'}}\right) - \frac{\gamma}{2\gamma-1}\left(p_{s'}\log\frac{p_{s'}}{p_{t'}} + p_{t'}\log\frac{p_{t'}}{p_{s'}}\right) \\
&= 2\frac{1}{2}\left(p_{t'}\log\frac{2p_{t'}}{p_{s'}+p_{t'}} + p_{s'}\log\frac{2p_{s'}}{p_{s'}+p_{t'}} - 2\log 2\right) - \frac{\gamma}{2\gamma-1}\left(p_{s'}\log\frac{p_{s'}}{p_{t'}} + p_{t'}\log\frac{p_{t'}}{p_{s'}}\right) \\
&= 2D_{JS}(\mathcal{D}_{S'}\|\mathcal{D}_{T'}) - 2\log 2 - \frac{\gamma}{2\gamma-1}(p_{s'}-p_{t'})\log\frac{p_{s'}}{p_{t'}}
\end{aligned}
$$

$$(12)$$

② in Equ. (11) can be rearranged to

$$
\begin{aligned}
& \gamma\left(p_s\log\frac{p_{s'}}{p_{t'}} + p_t\log\frac{p_{t'}}{p_{s'}}\right) \\
&= \gamma\log\frac{p_{s'}}{p_{t'}}\left(\frac{\gamma p_{s'} + (\gamma-1)p_{t'}}{2\gamma-1} - \frac{\gamma p_{t'} + (\gamma-1)p_{s'}}{2\gamma-1}\right) \\
&= \frac{\gamma}{2\gamma-1}(p_{s'}-p_{t'})\log\frac{p_{s'}}{p_{t'}}
\end{aligned}
$$

$$(13)$$

By plugging the rearranged ① and ② into Equ. (11), we get

$$
\max_{\hat{h}\in\hat{\mathcal{H}}} d_{\hat{h},g,\gamma}(\mathcal{D}_S, \mathcal{D}_T) = 2D_{JS}(\mathcal{D}_{S'}\|\mathcal{D}_{T'}) - 2\log 2 \tag{14}
$$

□

## A.2 CONNECT ONE-SIDED ENVIRONMENT LABEL SMOOTHING TO JS DIVERGENCE MINIMIZATION

**Proposition 2.** *Given two domain distributions $\mathcal{D}_S, \mathcal{D}_T$ over $X$, where $\mathcal{D}_S$ is the read data distribution and $\mathcal{D}_T$ is the generated data distribution. The cost used for the discriminator is:*

$$
\max_{h\in\mathcal{H}} d_h(\mathcal{D}_S, \mathcal{D}_T) = \max_{h\in\mathcal{H}} \mathbb{E}_{\mathbf{x}_s\sim\mathcal{D}_S}\log h(\mathbf{x}_s) + \mathbb{E}_{\mathbf{x}_t\sim\mathcal{D}_T}\log\left(1 - h(\mathbf{x}_t)\right), \tag{15}
$$

*where $h \in \mathcal{H} : \mathcal{X} \to [0,1]$. Suppose $h \in \mathcal{H}$ the optimal discriminator with no constraint and mixed distributions $\begin{cases} \mathcal{D}_{S'} = \gamma\mathcal{D}_S \\ \mathcal{D}_{T'} = \mathcal{D}_T + (1-\gamma)\mathcal{D}_S \end{cases}$ with hyper-parameter $\gamma$. Then to minimize domain divergence by adversarial training with **one-sided environment label smoothing** is equal to minimize $2D_{JS}(\mathcal{D}_{S'}\|\mathcal{D}_{T'}) - 2\log 2$, where $D_{JS}$ is the Jensen-Shanon (JS) divergence.*

*Proof.* Applying *one-sided environment label smoothing*, the target can be reformulated to

$$
\begin{aligned}
\max_{h\in\mathcal{H}} d_{h,\gamma}(\mathcal{D}_S, \mathcal{D}_T) &= \max_{h\in\mathcal{H}} \mathbb{E}_{\mathbf{x}_s\sim\mathcal{D}_S}\left[\gamma\log h(\mathbf{x}_s) + (1-\gamma)\log\left(1-h(\mathbf{x}_s)\right)\right] + \mathbb{E}_{\mathbf{x}_t\sim\mathcal{D}_T}\left[\log\left(1-h(\mathbf{x}_t)\right)\right] \\
&= \max_{h\in\mathcal{H}} \int_{\mathcal{X}} p_s(\mathbf{x})\log\left[\gamma\log h(\mathbf{x}) + (1-\gamma)\log(1-h(\mathbf{x}))\right] + p_t(\mathbf{x})\log(1-h(\mathbf{x}))
\end{aligned}
$$

$$(16)$$

where $\gamma$ is a value slightly less than one, $p_s(\mathbf{x}), p_t(\mathbf{x})$ is the density of $\mathcal{D}_S, \mathcal{D}_T$ respectively. By taking derivatives and finding the optimal $h$ we can get $h^* = \frac{\gamma p_s(\mathbf{x})}{p_s(\mathbf{x})+p_t(\mathbf{x})}$. Plugging the optimal $h^*$

into the original target we can get:

$$
= \int_{\mathcal{X}} p_s(\mathbf{x}) \left[ \gamma \log \frac{\gamma p_s(\mathbf{x})}{p_s(\mathbf{x}) + p_t(\mathbf{x})} + (1 - \gamma) \log \frac{p_t(\mathbf{x}) + (1 - \gamma) p_s(\mathbf{x})}{p_s(\mathbf{x}) + p_t(\mathbf{x})} \right] + p_t(\mathbf{x}) \log \frac{p_t(\mathbf{x}) + (1 - \gamma) p_s(\mathbf{x})}{p_s(\mathbf{x}) + p_t(\mathbf{x})} d_{\mathbf{x}}
$$

$$
= \int_{\mathcal{X}} p_s(\mathbf{x}) \gamma \log \frac{\gamma p_s(\mathbf{x})}{p_s(\mathbf{x}) + p_t(\mathbf{x})} + [p_s(1 - \gamma) + p_t(\mathbf{x})] \log \frac{p_t(\mathbf{x}) + (1 - \gamma) p_s(\mathbf{x})}{p_s(\mathbf{x}) + p_t(\mathbf{x})} d_{\mathbf{x}}
$$

$$
= \int_{\mathcal{X}} p_{s'}(\mathbf{x}) \log \frac{p_{s'}(\mathbf{x})}{p_{s'}(\mathbf{x}) + p_{t'}(\mathbf{x})} + p_{t'}(\mathbf{x}) \log \frac{p_{t'}(\mathbf{x})}{p_{s'}(\mathbf{x}) + p_{t'}(\mathbf{x})} d_{\mathbf{x}}
$$

$$
= 2 D_{JS}(\mathcal{D}_{S'} \| \mathcal{D}_{T'}) - 2 \log 2,
$$

(17)

where $\begin{cases} \mathcal{D}_{S'} = \gamma \mathcal{D}_S \\ \mathcal{D}_{T'} = \mathcal{D}_T + (1 - \gamma) \mathcal{D}_S \end{cases}$ are two mixed distributions and $\begin{cases} p_{s'} = \gamma p_s \\ p_{t'} = p_t + (1 - \gamma) p_s \end{cases}$ are their densities. $\square$

Our result supplies an explanation to "*why GANs only use one-sided label smoothing rather than native label smoothing*". That is, if the density of real data in a region is near zero $p_s(\mathbf{x}) \to 0$, native environment label smoothing will be dominated by only the generated sample densities because $\begin{cases} p_{s'} = p_t + \gamma(p_s - p_t) \approx (1 - \gamma) p_t \\ p_{t'} = p_s + \gamma(p_t - p_s) \approx \gamma p_t \end{cases}$ . Namely, the discriminator will not align the distribution between generated samples and real samples, but enforce the generator to produce samples that follow the fake mode $\mathcal{D}_T$. In contrast, one-sided label smoothing reserves the real distribution density as far as possible, that is, $p_{s'} = \gamma p_s, p_{t'} \approx \gamma p_t$, which avoids divergence minimization between fake mode to fake mode and relieves model collapse.

### A.3  CONNECT MULTI-DOMAIN ADVERSARIAL TRAINING TO KL DIVERGENCE MINIMIZATION

**Proposition 3.** *Given domain distributions $\{\mathcal{D}_i\}_{i=1}^M$ over $X$, and a hypothesis class $\mathcal{H}$. Suppose $\hat{h} \in \hat{\mathcal{H}}$ the optimal discriminator with no constraint and mixed distributions $\mathcal{D}_{Mix} = \sum_{i=1}^M \mathcal{D}_i$, and $\{\mathcal{D}_{i'} = \gamma \mathcal{D}_i + \frac{1-\gamma}{M-1} \sum_{j=1;j \neq i}^M \mathcal{D}\}_{i=1}^M$ with hyper-parameter $\gamma \in [0.5, 1]$. Then to minimize domain divergence by adversarial training w/wo **environment label smoothing** is equal to minimize $\sum_{i=1}^M D_{KL}(\mathcal{D}_i \| \mathcal{D}_{Mix})$, and $\sum_{i=1}^M D_{KL}(\mathcal{D}_{i'} \| \mathcal{D}_{Mix})$ respectively, where $D_{KL}$ is the Kullback–Leibler (KL) divergence.*

*Proof.* We restate corresponding notations and definitions as follows. Given $M$ domains $\{\mathcal{D}_i\}_{i=1}^M$. Let the hypothesis $h$ be the composition of $h = \hat{h} \circ g$, where $g \in \mathcal{G}$ pushes forward the data samples to a representation space $\mathcal{Z}$ and the domain discriminator with softmax activation function is defined as $\hat{h} = (\hat{h}_1(\cdot), \dots, \hat{h}_M(\cdot)) \in \hat{\mathcal{H}} : \mathcal{Z} \to [0, 1]^M; \sum_{i=1}^M \hat{h}_i(\cdot) = 1$. Denote $g \circ \mathcal{D}_i$ the feature distribution of $\mathcal{D}_i$ which is encoded by encoder $g$. The cost used for the discriminator can be defined as:

$$
\max_{\hat{h} \in \hat{\mathcal{H}}} d_{\hat{h}, g}(\mathcal{D}_1, \dots, \mathcal{D}_M) = \max_{\hat{h} \in \mathcal{H}} \mathbb{E}_{\mathbf{z} \sim g \circ \mathcal{D}_1} \log \hat{h}_1(\mathbf{z}) + \dots + \mathbb{E}_{\mathbf{z} \sim g \circ \mathcal{D}_M} \log \hat{h}_M(\mathbf{z}), \text{s.t.} \sum_{i=1}^M \hat{h}_i(\mathbf{z}) = 1
$$

(18)

Denote $p_i^z(\mathbf{z})$ the density of feature distribution $g \circ \mathcal{D}_i$. For simplicity, we ignore $\int_{\mathcal{Z}}$. Applying lagrange multiplier and taking the first derivative with respect to each $\hat{h}_i$, we can get

$$
\begin{cases} \frac{\partial d_{\hat{h}, g}}{\partial \hat{h}_1} = p_1^z(\mathbf{z}) \frac{1}{\hat{h}_1(z)} - \lambda = 0 \\ \vdots \\ \frac{\partial d_{\hat{h}, g}}{\partial \hat{h}_M} = p_M^z(\mathbf{z}) \frac{1}{\hat{h}_M(z)} - \lambda = 0 \end{cases} \Rightarrow \begin{cases} \hat{h}_1(\mathbf{z}) = \frac{p_1^z(\mathbf{z})}{\lambda} \\ \vdots \\ \hat{h}_M(\mathbf{z}) = \frac{p_M^z(\mathbf{z})}{\lambda} \end{cases} \Rightarrow ① \begin{cases} \hat{h}_1^*(\mathbf{z}) = \frac{p_1^z(\mathbf{z})}{p_1^z(\mathbf{z}) + \dots + p_M^z(\mathbf{z})} \\ \vdots \\ \hat{h}_M^*(\mathbf{z}) = \frac{p_M^z(\mathbf{z})}{p_1^z(\mathbf{z}) + \dots + p_M^z(\mathbf{z})} \end{cases}
$$

(19)

where $\lambda$ is the lagrange variable and ① is because the constraint $\sum_{i=1}^M \hat{h}_i(\mathbf{z}) = 1$. Denote $\mathcal{D}_{Mix} = \sum_{i=1}^M \mathcal{D}_i$ is a mixed distribution and $p_{Mix} = \sum_{i=1}^M p_i$ is the density. Then we have

$$\max_{\hat{h}\in\hat{\mathcal{H}}} d_{\hat{h},g}(\mathcal{D}_1,\ldots,\mathcal{D}_M) = \int_{\mathcal{Z}} p_1^z(\mathbf{z})\log\frac{p_1^z(\mathbf{z})}{p_{Mix}^z(\mathbf{z})} + p_2^z(\mathbf{z})\log\frac{p_2^z(\mathbf{z})}{p_{Mix}^z(\mathbf{z})} + \cdots + p_M^z(\mathbf{z})\log\frac{p_M^z(\mathbf{z})}{p_{Mix}^z(\mathbf{z})}d_{\mathbf{z}}$$

$$= \sum_{i=1}^{M} D_{KL}(\mathcal{D}_i\|\mathcal{D}_{Mix}),$$

(20)

where $D_{KL}$ is the KL divergence. With **environment label smoothing**, the target is

$$\max_{\hat{h}\in\hat{\mathcal{H}}} d_{\hat{h},g,\gamma}(\mathcal{D}_1,\ldots,\mathcal{D}_M) = \max_{\hat{h}\in\hat{\mathcal{H}}} \mathbb{E}_{\mathbf{z}\sim g\circ\mathcal{D}_1}\left[\gamma\log\hat{h}_1(\mathbf{z}) + \frac{(1-\gamma)}{M-1}\sum_{j=1;j\neq1}^{M}\log\left(\hat{h}_j(\mathbf{z})\right)\right] + \cdots +$$

$$\mathbb{E}_{\mathbf{z}\sim g\circ\mathcal{D}_M}\left[\gamma\log\hat{h}_M(\mathbf{z}) + \frac{(1-\gamma)}{M-1}\sum_{j=1;j\neq M}^{M}\log\left(\hat{h}_j(\mathbf{z})\right)\right], \text{s.t.} \sum_{i=1}^{M}\hat{h}_i(\mathbf{z})=1$$

(21)

Take the same operation as Equ. (19) we can get

$$\begin{cases}\frac{\partial d_{\hat{h},g,\gamma}}{\partial\hat{h}_1} = \gamma p_1^z(\mathbf{z})\frac{1}{\hat{h}_1(z)} + \frac{1-\gamma}{M-1}\sum_{j=1;j\neq1}^{M}p_j^z(\mathbf{z})\frac{1}{\hat{h}_1(z)} - \lambda = 0 \\ \vdots \\ \frac{\partial d_{\hat{h},g,\gamma}}{\partial\hat{h}_M} = \gamma p_M^z(\mathbf{z})\frac{1}{\hat{h}_M(z)} + \frac{1-\gamma}{M-1}\sum_{j=1;j\neq M}^{M}p_j^z(\mathbf{z})\frac{1}{\hat{h}_M(z)} - \lambda = 0\end{cases} \Rightarrow \begin{cases}\hat{h}_1^*(\mathbf{z}) = \frac{\gamma p_1^z(\mathbf{z})+\frac{1-\gamma}{M-1}\sum_{j=1;j\neq1}^{M}p_j^z(\mathbf{z})}{p_1^z(\mathbf{z})+\cdots+p_M^z(\mathbf{z})} \\ \vdots \\ \hat{h}_M^*(\mathbf{z}) = \frac{\gamma p_M^z(\mathbf{z})+\frac{1-\gamma}{M-1}\sum_{j=1;j\neq M}^{M}p_j^z(\mathbf{z})}{p_1^z(\mathbf{z})+\cdots+p_M^z(\mathbf{z})}\end{cases}$$

(22)

Denote $\{\mathcal{D}_{i'} = \gamma\mathcal{D}_i + \frac{1-\gamma}{M-1}\sum_{j=1;j\neq i}^{M}\mathcal{D}\}_{i=1}^{M}$ a set of mixed distributions and $\{p_{i'}(\mathbf{z}) = \gamma p_i^z(\mathbf{z}) + \frac{1-\gamma}{M-1}\sum_{j=1;j\neq i}^{M}p_j^z(\mathbf{z})\}_{i=1}^{M}$ the corresponding densities. Plugging Equ. (22) to the target we can get

$$\sum_{i=1}^{M}\left[\int_{\mathcal{Z}}\gamma p_i^z(z)\log\frac{\gamma p_i^z(\mathbf{z})+\frac{1-\gamma}{M-1}\sum_{j=1;j\neq i}^{M}p_j^z(\mathbf{z})}{p_i^z(\mathbf{z})+\cdots+p_M^z(\mathbf{z})} + \frac{(1-\gamma)}{M-1}\sum_{k=1;k\neq i}^{M}p_i^z(\mathbf{z})\log\frac{\gamma p_k^z(\mathbf{z})+\frac{1-\gamma}{M-1}\sum_{j=1;j\neq i}^{M}p_j^z(\mathbf{z})}{p_i^z(\mathbf{z})+\cdots+p_M^z(\mathbf{z})}d_{\mathbf{z}}\right]$$

$$= \sum_{i=1}^{M}\left[\int_{\mathcal{Z}}\gamma p_i^z(z)\log\frac{\gamma p_i^z(\mathbf{z})+\frac{1-\gamma}{M-1}\sum_{j=1;j\neq i}^{M}p_j^z(\mathbf{z})}{p_i^z(\mathbf{z})+\cdots+p_M^z(\mathbf{z})} + \frac{(1-\gamma)}{M-1}\sum_{k=1;k\neq i}^{M}p_k^z(\mathbf{z})\log\frac{\gamma p_i^z(\mathbf{z})+\frac{1-\gamma}{M-1}\sum_{j=1;j\neq i}^{M}p_j^z(\mathbf{z})}{p_i^z(\mathbf{z})+\cdots+p_M^z(\mathbf{z})}d_{\mathbf{z}}\right]$$

$$= \sum_{i=1}^{M}\left[\int_{\mathcal{Z}}\left(\gamma p_i^z(z)+\frac{(1-\gamma)}{M-1}\sum_{k=1;k\neq i}^{M}p_k^z(\mathbf{z})\right)\log\frac{\gamma p_i^z(\mathbf{z})+\frac{1-\gamma}{M-1}\sum_{j=1;j\neq i}^{M}p_j^z(\mathbf{z})}{p_i^z(\mathbf{z})+\cdots+p_M^z(\mathbf{z})}d_{\mathbf{z}}\right]$$

$$= \sum_{i=1}^{M}D_{KL}(\mathcal{D}_{i'}\|\mathcal{D}_{Mix})$$

(23)

$\square$

## A.4 TRAINING STABILITY BROUGHT BY ENVIRONMENT LABEL SMOOTHING

Let $\mathcal{D}_S, \mathcal{D}_T$ two distributions and $\mathcal{D}_S^z, \mathcal{D}_T^z$ their induced distributions projected by encoder $g: \mathcal{X} \to \mathcal{Z}$ over feature space. We first show that if $\mathcal{D}_S^z, \mathcal{D}_T^z$ are disjoint or lie in low dimensional manifolds, there is always a perfect discriminator between them.

**Theorem 1.** *(Theorem 2.1. in (Arjovsky & Bottou, 2017).) If two distribution $\mathcal{D}_S^z, \mathcal{D}_T^z$ have support contained on two disjoint compact subsets $\mathcal{M}$ and $\mathcal{P}$ respectively, then there is a smooth optimal discriminator $\hat{h}^*: \mathcal{Z} \to [0,1]$ that has accuracy 1 and $\nabla_{\mathbf{z}}\hat{h}^*(\mathbf{z}) = 0$ for all $\mathbf{z} \sim \mathcal{M}\cup\mathcal{P}$.*

**Theorem 2.** *(Theorem 2.2. in (Arjovsky & Bottou, 2017).) Assume two distribution $\mathcal{D}_S^z, \mathcal{D}_T^z$ have support contained in two closed manifolds $\mathcal{M}$ and $\mathcal{P}$ that don't perfectly align and don't have full dimension. Both $\mathcal{D}_S^z, \mathcal{D}_T^z$ are assumed to be continuous in their respective manifolds. Then, there is a smooth optimal discriminator $\hat{h}^*: \mathcal{Z} \to [0,1]$ that has accuracy 1, and for almost all $\mathbf{z} \sim \mathcal{M}\cup\mathcal{P}$, $\hat{h}^*$ is smooth in a neighbourhood of $\mathbf{z}$ and $\nabla_{\mathbf{z}}\hat{h}^*(\mathbf{z}) = 0$.*

Namely, if the two distributions have supports that are disjoint or lie on low dimensional manifolds, the optimal discriminator will be accurate on all samples and its gradient will be zero almost everywhere. Then we can study the gradients we pass to the generator through a discriminator.

**Proposition 4.** *Denote $g(\theta; \cdot) : \mathcal{X} \to \mathcal{Z}$ a differentiable function that induces distributions $\mathcal{D}_S^z, \mathcal{D}_T^z$ with parameter $\theta$, and $\hat{h}$ a differentiable discriminator. If Theorem 1 or 2 holds, given a $\epsilon$-optimal discriminator $\hat{h}$, that is $\sup_{\mathbf{z} \in \mathcal{Z}} \parallel \nabla_{\mathbf{z}} \hat{h}(\mathbf{z}) \parallel_2 + |\hat{h}(\mathbf{z}) - \hat{h}^*(\mathbf{z})| < \epsilon^1$, assume the Jacobian matrix of $g(\theta; \mathbf{x})$ given $\mathbf{x}$ is bounded by $\sup_{\mathbf{x} \in \mathcal{X}} [\parallel J_\theta(g(\theta; \mathbf{x})) \parallel_2] \leq C$, then we have*

$$\lim_{\epsilon \to 0} \parallel \nabla_\theta d_{\hat{h}, g}(\mathcal{D}_S, \mathcal{D}_T) \parallel_2 = 0 \tag{24}$$

$$\lim_{\epsilon \to 0} \parallel \nabla_\theta d_{\hat{h}, g, \gamma}(\mathcal{D}_S, \mathcal{D}_T) \parallel_2 < 2(1 - \gamma)C \tag{25}$$

*Proof.* Theorem 1 or 2 show that in Equ. (8), $\hat{h}^*$ is locally one on the support of $\mathcal{D}_S^z$ and zero on the support of $\mathcal{D}_T^z$. Then, using Jensen's inequality, triangle inequality, and the chain rule on these supports, the gradients we pass to the generator through a discriminator given $\mathbf{x}_s \sim \mathcal{D}_S$ is

$$\parallel \nabla_\theta \mathbb{E}_{\mathbf{x}_s \sim \mathcal{D}_S} \left[ \gamma \log \hat{h} \circ g(\theta; \mathbf{x}_s) + (1 - \gamma) \log \left( 1 - \hat{h} \circ g(\theta; \mathbf{x}_s) \right) \right] \parallel_2$$

$$\leq \mathbb{E}_{\mathbf{x}_s \sim \mathcal{D}_S} \left[ \parallel \nabla_\theta \gamma \log \hat{h} \circ g(\theta; \mathbf{x}_s) \parallel_2 \right] + \mathbb{E}_{\mathbf{x}_s \sim \mathcal{D}_S} \left[ \parallel \nabla_\theta (1 - \gamma) \log \left( 1 - \hat{h} \circ g(\theta; \mathbf{x}_s) \right) \parallel_2 \right]$$

$$\leq \mathbb{E}_{\mathbf{x}_s \sim \mathcal{D}_S} \left[ \gamma \frac{\parallel \nabla_\theta \hat{h} \circ g(\theta; \mathbf{x}_s) \parallel_2}{|\hat{h} \circ g(\theta; \mathbf{x}_s)|} \right] + \mathbb{E}_{\mathbf{x}_s \sim \mathcal{D}_S} \left[ (1 - \gamma) \frac{\parallel \nabla_\theta \hat{h} \circ g(\theta; \mathbf{x}_s) \parallel_2}{|1 - \hat{h} \circ g(\theta; \mathbf{x}_s)|} \right]$$

$$\leq \mathbb{E}_{\mathbf{x}_s \sim \mathcal{D}_S} \left[ \gamma \frac{\parallel \nabla_{\mathbf{z}} \hat{h}(\mathbf{z}) \parallel_2 \parallel J_\theta(g(\theta; \mathbf{x}_s)) \parallel_2}{|\hat{h} \circ g(\theta; \mathbf{x}_s)|} \right] + \mathbb{E}_{\mathbf{x}_s \sim \mathcal{D}_S} \left[ (1 - \gamma) \frac{\parallel \nabla_{\mathbf{z}} \hat{h}(\mathbf{z}) \parallel_2 \parallel J_\theta(g(\theta; \mathbf{x}_s)) \parallel_2}{|1 - \hat{h} \circ g(\theta; \mathbf{x}_s)|} \right]$$

$$< \gamma \mathbb{E}_{\mathbf{x}_s \sim \mathcal{D}_S} \left[ \frac{\epsilon \parallel J_\theta(g(\theta; \mathbf{x}_s)) \parallel_2}{|\hat{h}^* \circ g(\theta; \mathbf{x}_s) - \epsilon|} \right] + (1 - \gamma) \mathbb{E}_{\mathbf{x}_s \sim \mathcal{D}_S} \left[ \frac{\epsilon \parallel J_\theta(g(\theta; \mathbf{x}_s)) \parallel_2}{|1 - \hat{h}^* \circ g(\theta; \mathbf{x}_s) + \epsilon|} \right]$$

$$\leq \gamma \frac{\epsilon C}{1 - \epsilon} + (1 - \gamma)C, \tag{26}$$

where the fifth line is because we have $\hat{h}(z) \approx \hat{h}^*(z) - \epsilon$ when $\epsilon$ is small enough and $\parallel \nabla_{\mathbf{z}} \hat{h}(\mathbf{z}) \parallel_2 < \epsilon$. Similarly we can get the gradients given $\mathbf{x}_t \sim \mathcal{D}_T$ is

$$\parallel \nabla_\theta \mathbb{E}_{\mathbf{x}_t \sim \mathcal{D}_T} \left[ (1 - \gamma) \log \hat{h} \circ g(\mathbf{x}_t) + \gamma \log \left( 1 - \hat{h} \circ g(\mathbf{x}_t) \right) \right] \parallel_2$$

$$< (1 - \gamma) \mathbb{E}_{\mathbf{x}_t \sim \mathcal{D}_T} \left[ \frac{\epsilon \parallel J_\theta(g(\theta; \mathbf{x}_t)) \parallel_2}{|\hat{h}^* \circ g(\theta; \mathbf{x}_t) + \epsilon|} \right] + \gamma \mathbb{E}_{\mathbf{x}_t \sim \mathcal{D}_T} \left[ \frac{\epsilon \parallel J_\theta(g(\theta; \mathbf{x}_t)) \parallel_2}{|1 - \hat{h}^* \circ g(\theta; \mathbf{x}_t) - \epsilon|} \right] \tag{27}$$

$$\leq (1 - \gamma)C + \gamma \frac{\epsilon C}{1 - \epsilon}$$

Here $\hat{h}(z) \approx \hat{h}^*(z) + \epsilon$ because $\hat{h}^*$ is locally zero on the support of $\mathcal{D}_T^z$. Then we have

$$\lim_{\epsilon \to 0} \parallel \nabla_\theta d_{\hat{h}, g, \gamma}(\mathcal{D}_S, \mathcal{D}_T) \parallel_2$$

$$\leq \lim_{\epsilon \to 0} \parallel \nabla_\theta \mathbb{E}_{\mathbf{x}_s \sim \mathcal{D}_S} \left[ \gamma \log \hat{h} \circ g(\theta; \mathbf{x}_s) + (1 - \gamma) \log \left( 1 - \hat{h} \circ g(\theta; \mathbf{x}_s) \right) \right] \parallel_2$$

$$+ \parallel \nabla_\theta \mathbb{E}_{\mathbf{x}_t \sim \mathcal{D}_T} \left[ (1 - \gamma) \log \hat{h} \circ g(\mathbf{x}_t) + \gamma \log \left( 1 - \hat{h} \circ g(\mathbf{x}_t) \right) \right] \parallel_2 \tag{28}$$

$$< \lim_{\epsilon \to 0} \underbrace{\gamma \frac{\epsilon C}{1 - \epsilon} + \gamma \frac{\epsilon C}{1 - \epsilon}}_{①} + \underbrace{(1 - \gamma)C + (1 - \gamma)C}_{②}$$

$$= 2(1 - \gamma)C,$$

where ① is equal to the gradient of native DANN in Equ. (7) times $\gamma$, namely

$$\lim_{\epsilon \to 0} \parallel \nabla_\theta d_{\hat{h}, g}(\mathcal{D}_S, \mathcal{D}_T) \parallel_2 = 0, \tag{29}$$

---

[1]The constraint on $\parallel \nabla_{\mathbf{z}} \hat{h}(\mathbf{z}) \parallel_2$ is because the optimal discriminator has zero gradients almost everywhere, and $|\hat{h}(\mathbf{z}) - \hat{h}^*(\mathbf{z})|$ is a constraint on the prediction accuracy.

which shows that as our discriminator gets better, the gradient of the encoder vanishes. With environment label smoothing, we have

$$\lim_{\epsilon \to 0} \| \nabla_\theta d_{\hat{h},g,\gamma}(\mathcal{D}_S, \mathcal{D}_T) \|_2 = 2(1-\gamma)C, \tag{30}$$

which alleviates the problem of gradients vanishing. $\qquad\square$

## A.5 TRAINING STABILITY ANALYSIS OF MULTI-DOMAIN SETTINGS

Let $\{\mathcal{D}_i\}_{i=1}^M$ a set of data distributions and $\{\mathcal{D}_i^z\}_{i=1}^M$ their induced distributions projected by encoder $g : \mathcal{X} \to \mathcal{Z}$ over feature space. Recall that the domain discriminator with softmax activation function is defined as $\hat{h} = (\hat{h}_1, \ldots, \hat{h}_M) \in \hat{\mathcal{H}} : \mathcal{Z} \to [0,1]^M$, where $\hat{h}_i(\mathbf{z})$ denotes the probability that $\mathbf{z}$ belongs to $\mathcal{D}_i^z$. To verify the existence of each optimal discriminator $\hat{h}_i^*$, we can easily replace $\mathcal{D}_s^z, \mathcal{D}_t^z$ in Theorem 1 and Theorem 2 by $\mathcal{D}_i^z, \sum_{j=1;j\neq i}^M \mathcal{D}_j^z$ respectively. Namely, if distribution $\mathcal{D}_i^z$ and $\sum_{j=1;j\neq i}^M \mathcal{D}_j^z$ have supports that are disjoint or lie on low dimensional manifolds, $\hat{h}_i^*$ can perfectly discriminate samples within and beyond $\mathcal{D}_i^z$ and its gradient will be zero almost everywhere.

**Proposition 5.** *Denote $g(\theta; \cdot) : \mathcal{X} \to \mathcal{Z}$ a differentiable function that induces distributions $\{\mathcal{D}_i^z\}_{i=1}^M$ with parameter $\theta$, and $\{\hat{h}_i\}_{i=1}^M$ corresponding differentiable discriminators. If optimal discriminators for induced distributions exist, given any $\epsilon$-optimal discriminator $\hat{h}_i$, we have $\sup_{\mathbf{z} \in \mathcal{Z}} \| \nabla_{\mathbf{z}} \hat{h}_i(\mathbf{z}) \|_2 + |\hat{h}_i(\mathbf{z}) - \hat{h}_i^*(\mathbf{z})| < \epsilon$, assume the Jacobian matrix of $g(\theta; \mathbf{x})$ given $\mathbf{x}$ is bounded by $\sup_{\mathbf{x} \in \mathcal{X}} [\| J_\theta(g(\theta; \mathbf{x})) \|_2] \leq C$, then we have*

$$\lim_{\epsilon \to 0} \| \nabla_\theta d_{\hat{h},g}(\mathcal{D}_1, \ldots, \mathcal{D}_M) \|_2 = 0 \tag{31}$$

$$\lim_{\epsilon \to 0} \| \nabla_\theta d_{\hat{h},g,\gamma}(\mathcal{D}_1, \ldots, \mathcal{D}_M) \|_2 < M(1-\gamma)C \tag{32}$$

*Proof.* Following the proof in Proposition 4, we have

$$
\begin{aligned}
&\lim_{\epsilon \to 0} \| \nabla_\theta \mathbb{E}_{\mathbf{x} \in \mathcal{D}_i} \left[ \gamma \log \hat{h}_i \circ g(\mathbf{x}) + \frac{(1-\gamma)}{M-1} \sum_{j=1;j\neq i}^M \log\left(\hat{h}_j \circ g(\mathbf{x})\right) \right] \|_2 \\
&\leq \lim_{\epsilon \to 0} \mathbb{E}_{\mathbf{x} \sim \mathcal{D}_i} \left[ \gamma \frac{\| \nabla_\theta \hat{h}_i \circ g(\theta; \mathbf{x}) \|_2}{|\hat{h}_i \circ g(\theta; \mathbf{x})|} \right] + \frac{(1-\gamma)}{M-1} \sum_{j=1;j\neq i}^M \mathbb{E}_{\mathbf{x} \sim \mathcal{D}_j} \left[ \gamma \frac{\| \nabla_\theta \hat{h}_j \circ g(\theta; \mathbf{x}) \|_2}{|\hat{h}_j \circ g(\theta; \mathbf{x})|} \right] \\
&< \lim_{\epsilon \to 0} \gamma \mathbb{E}_{\mathbf{x} \sim \mathcal{D}_i} \left[ \frac{\epsilon \| J_\theta(g(\theta; \mathbf{x})) \|_2}{|\hat{h}_i^* \circ g(\theta; \mathbf{x}) - \epsilon|} \right] + \frac{(1-\gamma)}{M-1} \sum_{j=1;j\neq i}^M \gamma \mathbb{E}_{\mathbf{x} \sim \mathcal{D}_j} \left[ \frac{\epsilon \| J_\theta(g(\theta; \mathbf{x})) \|_2}{|\hat{h}_j^* \circ g(\theta; \mathbf{x}) + \epsilon|} \right] \\
&\leq \lim_{\epsilon \to 0} \left[ \gamma \frac{\epsilon C}{1-\epsilon} + (1-\gamma)C \right] \\
&= (1-\gamma)C
\end{aligned}
\tag{33}
$$

where the second line is because for $\mathbf{z} \sim \mathcal{D}_i^z$, $\hat{h}_i^*(\mathbf{z})$ is locally one and other optimal discriminators $\hat{h}_j^*(\mathbf{z})|j \neq i, j \in [M]$ are all locally zero, thus we have $\hat{h}_i(\mathbf{z}) \approx \hat{h}_i^*(\mathbf{z}) - \epsilon$, and $\hat{h}_j(\mathbf{z}) \approx \hat{h}_j^*(\mathbf{z}) + \epsilon$. $\lim_{\epsilon \to 0} \frac{\epsilon C}{1-\epsilon} = 0$ is the gradient that passed to the generator by native multi-domain DANN (Equ. (1)). Environment label smoothing leads to another term, that is $(1-\gamma)C$ and avoid gradients vanishing. Consider all distributions, we have

$$
\begin{aligned}
&\lim_{\epsilon \to 0} \| \nabla_\theta d_{\hat{h},g,\gamma}(\mathcal{D}_1, \ldots, \mathcal{D}_M) \|_2 \\
&\leq \lim_{\epsilon \to 0} \| \nabla_\theta \mathbb{E}_{\mathbf{x} \in \mathcal{D}_1} \left[ \gamma \log \hat{h}_1 \circ g(\mathbf{x}) + \frac{(1-\gamma)}{M-1} \sum_{j=2}^M \log\left(\hat{h}_j \circ g(\mathbf{x})\right) \right] \|_2 \\
&\quad + \cdots + \lim_{\epsilon \to 0} \| \nabla_\theta \mathbb{E}_{\mathbf{x} \in \mathcal{D}_M} \left[ \gamma \log \hat{h}_M \circ g(\mathbf{x}) + \frac{(1-\gamma)}{M-1} \sum_{j=1}^{M-1} \log\left(\hat{h}_j \circ g(\mathbf{x})\right) \right] \|_2 \\
&= M(1-\gamma)C,
\end{aligned}
\tag{34}
$$

$\square$

## A.6 ELS STABILIZE THE OSCILLATORY GRADIENT

For the clarity of our proof, the notations here is a little different compared to other sections. Let $ec(i)$ be the cross-entropy loss for class $i$, we denote $g$ is the encoder and $\{w_i\}_{i=1}^M$ is the classification parameter for all domains, then the adversarial loss function for a given sample $x$ with domain index $i$ here is

$$
\begin{aligned}
F(x,i) &= (1-\gamma)ec(i) + \frac{\gamma}{M}\sum_{j\neq i}ec(j) \\
&= ec(i) + \frac{\gamma}{M-1}\sum_j\left(ec(j) - ec(i)\right) \\
&= ec(i) + \frac{\gamma}{M-1}\sum_j\left(-\log\left(\frac{\exp(w_j^\top g(x))}{\sum_k\exp(w_k^\top g(x))}\right) + \log\left(\frac{\exp(w_i^\top g(x))}{\sum_k\exp(w_k^\top g(x))}\right)\right) \\
&= ec(i) + \frac{\gamma}{M-1}\sum_j\left(-w_j^\top g(x) + \log\left(\sum_k\exp(w_k^\top g(x))\right) + w_i^\top g(x) - \log\left(\sum_k\exp(w_k^\top g(x))\right)\right) \\
&= ec(i) + \frac{\gamma}{M-1}\sum_j\left((w_i - w_j)^\top g(x)\right) \\
&= -w_i^\top g(x) + \log\left(\sum_k\exp(w_k^\top g(x))\right) + \frac{\gamma}{M-1}\sum_j\left((w_i - w_j)^\top g(x)\right)
\end{aligned}
$$
(35)

We compute the gradient:

$$
\frac{\partial F(x,i)}{\partial w_i} = -g(x) + \frac{\exp(w_i^\top g(x))}{\sum_k\exp(w_k^\top g(x))}g(x) + \frac{\gamma}{M-1}g(x) = \left(-1 + p(i) + \frac{\gamma}{M-1}\right)g(x), \quad (36)
$$

where $p(i)$ denotes $\frac{\exp(w_i^\top g(x))}{\sum_k\exp(w_k^\top g(x))}$. When $\gamma$ is small (e.g., $\gamma \lesssim M(1-p(i))$), the gradient will be further pullback towards 0. Similarly, for $w_j$ and $g(x)$, we have

$$
\frac{\partial F(x,i)}{\partial w_j} = \frac{\exp(w_j^\top g(x))}{\sum_k\exp(w_k^\top g(x))}g(x) - \frac{\gamma}{M-1}g(x) = \left(p(j) - \frac{\gamma}{M-1}\right)g(x)
$$

$$
\frac{\partial F(x,i)}{\partial g(x)} = -w_i + \sum_j\frac{\exp(w_j^\top g(x))}{\sum_k\exp(w_k^\top g(x))}w_j + \frac{\gamma}{M-1}\sum_j(w_i - w_j) = -(1 - \frac{\gamma}{M-1})w_i + \sum_j\left(p(j) - \frac{\gamma}{M-1}\right)w_j,
$$
(37)

then with proper choice of $\gamma$ (e.g., $\gamma \lesssim \min_j Mp(j)$), the gradient w.r.t $w_j$ and $g(x)$ will also shrink towards zero.

## A.7 ENVIRONMENT LABEL SMOOTHING MEETS NOISY LABELS

In this subsection, we focus on binary classification settings and adopt the symmetric noise model (Kim et al., 2019). Some of our proofs follow (Wei et al., 2022) but different results and analyses are given. The symmetric noise model is widely accepted in the literature on learning with noisy labels and generates the noisy labels by randomly flipping the clean label to the other possible classes. Specifically, given two environment with high-dimensional feature $x$ environment label $y \in \{0,1\}$, denote noisy labels $\tilde{y}$ is generated by a noise transition matrix $T$, where $T_{ij}$ denotes denotes the probability of flipping the clean label $y = i$ to the noisy label $\tilde{y} = j$, i.e., $T_{ij} = P(\tilde{y} = j|y = i)$. Let $e = P(\tilde{y} = 1|y = 0) = P(\tilde{y} = 0|y = 1)$ denote the noisy rate, the binary symmetric transition matrix becomes:

$$
T = \begin{pmatrix} 1-e & e \\ e & 1-e \end{pmatrix}, \tag{38}
$$

Suppose $(x,y)$ are drawn from a joint distribution $\mathcal{D}$, but during training, only samples with noisy labels are accessible from $(x,\tilde{y}) \sim \tilde{\mathcal{D}}$. Denote $f := \hat{h} \circ g$ and $\ell$ the cross-entropy loss, minimizing the smoothed loss with noisy labels can then be converted to

$$
\min_f \mathbb{E}_{(x,\tilde{y})\sim\tilde{\mathcal{D}}}[\ell(f(x),\tilde{y}^\gamma)] = \min_f \mathbb{E}_{(x,\tilde{y})\sim\tilde{\mathcal{D}}}\left[\gamma\ell(f(x),\tilde{y}) + (1-\gamma)\ell(f(x),1-\tilde{y})\right] \tag{39}
$$

Let $c_1 = \gamma, c_2 = 1 - \gamma$, according to the law of total probability, we have Equ. (39) is equal to

$$
\begin{aligned}
\min_f \mathbb{E}_{x.y=0}[P(\tilde{y} = 0|y = 0)(c_1\ell(f(x),0) + c_2\ell(f(x),1) \\
+ P(\tilde{y} = 1|y = 0)(c_1\ell(f(x),1) + c_2\ell(f(x),0)] \\
+ \mathbb{E}_{x.y=1}[P(\tilde{y} = 0|y = 1)(c_1\ell(f(x),0) + c_2\ell(f(x),1) \\
+ P(\tilde{y} = 1|y = 1)(c_1\ell(f(x),1) + c_2\ell(f(x),0)]
\end{aligned}
\tag{40}
$$

recall that $e = P(\tilde{y} = 1|y = 0) = P(\tilde{y} = 0|y = 1)$, the above equation is equal to

$$
\begin{aligned}
\min_f \mathbb{E}_{x.y=0}&\left[(1-e)(c_1\ell(f(x),0) + c_2\ell(f(x),1) + e(c_1\ell(f(x),1) + c_2\ell(f(x),0)\right] \\
&+ \mathbb{E}_{x.y=1}\left[e(c_1\ell(f(x),0) + c_2\ell(f(x),1) + (1-e)(c_1\ell(f(x),1) + c_2\ell(f(x),0)\right] \\
= \min_f \mathbb{E}_{x.y=0}&\left[[(1-e)c_1 + ec_2]\ell(f(x),0) + [(1-e)c_2 + ec_1]\ell(f(x),1)\right] \\
&+ \mathbb{E}_{x.y=1}\left[[ec_2 + (1-e)c_1]\ell(f(x),1) + [ec_1 + (1-e c_2)]\ell(f(x),0)\right] \\
= \min_f \mathbb{E}_{x.y=0}&\left[[(1-e)c_1 + ec_2]\ell(f(x),0) + [(1-e)c_2 + ec_1]\ell(f(x),1)\right] \\
&+ \mathbb{E}_{x.y=1}\left[[(1-e)c_1 + ec_2]\ell(f(x),1) + [(1-e)c_2 + ec_1]\ell(f(x),0)\right] \\
&+ \mathbb{E}_{x.y=1}\left[[(e-e)(c_2-c_1)]\ell(f(x),1) - [(e-e)(c_2-c_1)]\ell(f(x),0)\right] \\
= \min_f \mathbb{E}_{(x,y)\sim\mathcal{D}}&\left[[(1-e)c_1 + ec_2]\ell(f(x),y) + [(1-e)c_2 + ec_1]\ell(f(x),1-y)\right] \\
= \min_f \mathbb{E}_{(x,y)\sim\mathcal{D}}&\left[(c_1+c_2)\ell(f(x),y)\right] \\
&+ [(1-e)c_2 + ec_1]\mathbb{E}_{(x,y)\sim\mathcal{D}}\left[\ell(f(x),1-y) - \ell(f(x),y)\right] \\
= \min_f \mathbb{E}_{(x,y)\sim\mathcal{D}}&[\ell(f(x),y)] + (1-\gamma-e+2\gamma e)\mathbb{E}_{(x,y)\sim\mathcal{D}}[\ell(f(x),1-y) - \ell(f(x),y)]
\end{aligned}
\tag{41}
$$

Assume $\gamma^*$ is the optimal smooth parameter that makes the corresponding classifier return the best performance on unseen clean data distribution (Wei et al., 2022). Then the above equation can be converted to

$$
\begin{aligned}
= \min_f \mathbb{E}_{(x,y)\sim\mathcal{D}}&[\ell(f(x),y^{\gamma^*})] \\
&+ (\gamma^* - \gamma - e + 2\gamma e))\mathbb{E}_{(x,y)\sim\mathcal{D}}[\ell(f(x),1-y) - \ell(f(x),y)],
\end{aligned}
\tag{42}
$$

namely minimizing the smoothed loss with noisy labels is equal to optimizing two terms,

$$
\min_f \underbrace{\mathbb{E}_{(x,y)\sim\mathcal{D}}[\ell(f(x),y^{\gamma^*})]}_{①\ \text{Risk under clean label}} + \underbrace{(\gamma^* - \gamma - e + 2\gamma e))\mathbb{E}_{(x,y)\sim\mathcal{D}}[\ell(f(x),1-y) - \ell(f(x),y)]}_{②\ \text{Reverse optimization}}
\tag{43}
$$

where ① is the risk under the clean label. The influence of both noisy labels and ELS are reflected in the last term of Equ. (43). Considering the reverse optimization term ②, which is the opposite of the optimization process as we expect. Without label smoothing, the weight of ② will be $\gamma^* - 1 + e$ and a high noisy rate $e$ will let this harmful term contributes more to our optimization. In contrast, by choosing the smooth parameter $\gamma = \frac{\gamma^* - e}{1 - 2e}$, ② will be removed. For example, if the noisy rate is zero, the best smooth parameter is just $\gamma^*$.

## A.8 Empirical Gap Analysis Adopted from Vapnik-Chervonenkis framework

**Theorem 3.** *(Lemma 1 in (Ben-David et al., 2010)) Given Definition 1 and Definition 2, let $\mathcal{H}$ be a hypothesis class of VC dimension $d$. If empirical distributions $\hat{\mathcal{D}}_S$ and $\hat{\mathcal{D}}_T$ all have at least $n$ samples, then for any $\delta \in (0,1)$, with probability at least $1 - \delta$,*

$$
d_{\mathcal{H}}(\mathcal{D}_S, \mathcal{D}_T) \leq \hat{d}_{\mathcal{H}}(\hat{\mathcal{D}}_S, \hat{\mathcal{D}}_T) + 4\sqrt{\frac{d\log(2n) + \log\frac{2}{\delta}}{n}}
\tag{44}
$$

Denote convex hull $\Lambda$ the set of mixture distributions, $\Lambda = \{\bar{\mathcal{D}}_{Mix} : \bar{\mathcal{D}}_{Mix} = \sum_{i=1}^M \pi_i \mathcal{D}_i, \pi_i \in \Delta\}$, where $\Delta$ is standard $M-1$-simplex. The convex hull assumption is commonly used in domain generalization setting (Zhang et al., 2021a; Albuquerque et al., 2019; Zhang et al., 2022a), while none of them focus on the empirical gap. Note that $d_{\mathcal{H}}(\bar{\mathcal{D}}_{Mix}, \mathcal{D}_T)$ in domain generalization setting is intractable for the unseen target domain $\mathcal{D}_T$ is unavailable during training. We thus need to convert $d_{\mathcal{H}}(\bar{\mathcal{D}}_{Mix}, \mathcal{D}_T)$ to a tractable objective. Let $\bar{\mathcal{D}}^*_{Mix} = \sum_{i=1}^M \pi_i^* \mathcal{D}_i, (\pi_0^*, \ldots, \pi_M^*) \in \Delta$, where $\pi_0^*, \ldots, \pi_M^* = \arg\min_{\pi_0, \ldots, \pi_M} d_{\mathcal{H}}(\bar{\mathcal{D}}_{Mix}, \mathcal{D}_T)$, and $\bar{\mathcal{D}}^*_{Mix}$ is the element within $\Lambda$ which is closest to the unseen target domain. Then we have

$$
\begin{aligned}
d_{\mathcal{H}}(\bar{\mathcal{D}}_{Mix}, \mathcal{D}_T) &= 2 \sup_{h \in \mathcal{H}} \left| \mathbb{E}_{\mathbf{x} \sim \bar{\mathcal{D}}_{Mix}}[h(\mathbf{x}) = 1] - \mathbb{E}_{\mathbf{x} \sim \mathcal{D}_T}[h(\mathbf{x}) = 1] \right| \\
&= 2 \sup_{h \in \mathcal{H}} \big| \, \mathbb{E}_{\mathbf{x} \sim \bar{\mathcal{D}}_{Mix}}[h(\mathbf{x}) = 1] - \mathbb{E}_{\mathbf{x} \sim \bar{\mathcal{D}}^*_{Mix}}[h(\mathbf{x}) = 1] \\
&\quad + \mathbb{E}_{\mathbf{x} \sim \bar{\mathcal{D}}^*_{Mix}}[h(\mathbf{x}) = 1] - \mathbb{E}_{\mathbf{x} \sim \mathcal{D}_T}[h(\mathbf{x}) = 1] \, \big| \\
&\leq d_{\mathcal{H}}(\bar{\mathcal{D}}^*_{Mix}, \mathcal{D}_T) + d_{\mathcal{H}}(\bar{\mathcal{D}}_{Mix}, \bar{\mathcal{D}}^*_{Mix})
\end{aligned}
\tag{45}
$$

The explanation follows (Zhang et al., 2021a) that the first term corresponds to "To what extent can the convex combination of the source domain approximate the target domain". The minimization of the first term requires diverse data or strong data augmentation, such that the unseen distribution lies within the convex combination of source domains. We dismiss this term in the following because it includes $\mathcal{D}_T$ and cannot be optimized. Follows Lemma 1 in (Albuquerque et al., 2019), the second term can be bounded by,

$$
d_{\mathcal{H}}(\bar{\mathcal{D}}_{Mix}, \bar{\mathcal{D}}^*_{Mix}) \leq \sum_{i=1}^M \sum_{j=1}^M \pi_i \pi_j^* d_{\mathcal{H}}(\mathcal{D}_i, \mathcal{D}_j) \leq \max_{i,j \in [M]} d_{\mathcal{H}}(\mathcal{D}_i, \mathcal{D}_j),
\tag{46}
$$

namely the second term can be bounded by the combination of pairwise $\mathcal{H}$-divergence between source domains. The cost (Equ. (1)) used for the multi-domain adversarial training can be seen as an approximation of such a target. Until now, we can bound the empirical gap with the help of Theorem 3

$$
\begin{aligned}
\sum_{i=1}^M \sum_{j=1}^M \pi_i \pi_j^* d_{\mathcal{H}}(\mathcal{D}_i, \mathcal{D}_j) &\leq \sum_{i=1}^M \sum_{j=1}^M \pi_i \pi_j^* \left[ \hat{d}_{\mathcal{H}}(\hat{\mathcal{D}}_i, \hat{\mathcal{D}}_j) + 4 \sqrt{\frac{d \log(2 \min(n_i, n_j)) + \log \frac{2}{\delta}}{\min(n_i, n_j)}} \right] \\
\left| \sum_{i=1}^M \sum_{j=1}^M \pi_i \pi_j^* d_{\mathcal{H}}(\mathcal{D}_i, \mathcal{D}_j) - \sum_{i=1}^M \sum_{j=1}^M \pi_i \pi_j^* \hat{d}_{\mathcal{H}}(\hat{\mathcal{D}}_i, \hat{\mathcal{D}}_j) \right| &\leq 4 \sqrt{\frac{d \log(2n^\star) + \log \frac{2}{\delta}}{n^\star}}
\end{aligned}
\tag{47}
$$

where $n_i$ is the number of samples in $\mathcal{D}_i$ and $n^\star = \min(n_1, \ldots, n_M)$.

## A.9 EMPIRICAL GAP ANALYSIS ADOPTED FROM NEURAL NET DISTANCE

**Proposition 6.** *(Adapted from Theorem A.2 in (Arora et al., 2017)) Let $\{\mathcal{D}_i\}_{i=1}^M$ a set of distributions and $\{\hat{\mathcal{D}}_i\}_{i=1}^M$ be empirical versions with at least $n^\star$ samples each. We assume that the set of discriminators with softmax activation function $\hat{h}(\theta; \cdot) = (\hat{h}_1(\theta_1, \cdot), \ldots, \hat{h}_M(\theta_M, \cdot)) \in \hat{\mathcal{H}} : \mathcal{Z} \to [0,1]^M; \sum_{i=1}^M \hat{h}_i(\theta_i; \cdot) = 1^2$ are L-Lipschitz with respect to the parameters $\theta$ and use $p$ denote the number of parameter $\theta_i$. There is a universal constant $c$ such that when $n^\star \geq \frac{cpM \log(Lp/\epsilon)}{\epsilon}$, we have with probability at least $1 - \exp(-p)$ over the randomness of $\{\hat{\mathcal{D}}_i\}_{i=1}^M$,*

$$
\mid d_{\hat{h}, g}(\mathcal{D}_1, \ldots, \mathcal{D}_M) - d_{\hat{h}, g}(\hat{\mathcal{D}}_1, \ldots, \hat{\mathcal{D}}_M) \mid \leq \epsilon
\tag{48}
$$

*Proof.* For simplicity, we ignore the parameter $\theta_i$ when using $h_i(\cdot)$. According to the following triangle inequality, below we focus on the term $\mid \mathbb{E}_{\mathbf{z} \sim g \circ \mathcal{D}_1} \log \hat{h}_1(\mathbf{z}) - \mathbb{E}_{\mathbf{z} \sim g \circ \hat{\mathcal{D}}_1} \log \hat{h}_1(\mathbf{z}) \mid$ and other

---

[2]There might be some confusion here because in Section A.4 we use $\theta$ as the parameters of encoder $h$. The usage is just for simplicity but does not mean that $h, g$ have the same parameters.

terms have the same results.

$$| d_{\hat{h},g}(\mathcal{D}_1, \ldots, \mathcal{D}_M) - d_{\hat{h},g}(\hat{\mathcal{D}}_1, \ldots, \hat{\mathcal{D}}_M) |$$

$$= \left| \mathbb{E}_{\mathbf{z} \sim g \circ \mathcal{D}_1} \log \hat{h}_1(\mathbf{z}) + \cdots + \mathbb{E}_{\mathbf{z} \sim g \circ \mathcal{D}_M} \log \hat{h}_M(\mathbf{z}) - \mathbb{E}_{\mathbf{z} \sim g \circ \hat{\mathcal{D}}_1} \log \hat{h}_1(\mathbf{z}) - \cdots - \mathbb{E}_{\mathbf{z} \sim g \circ \hat{\mathcal{D}}_M} \log \hat{h}_M(\mathbf{z}) \right|$$

$$\leq | \mathbb{E}_{\mathbf{z} \sim g \circ \mathcal{D}_1} \log \hat{h}_1(\mathbf{z}) - \mathbb{E}_{\mathbf{z} \sim g \circ \hat{\mathcal{D}}_1} \log \hat{h}_1(\mathbf{z}) | + \cdots + | \mathbb{E}_{\mathbf{z} \sim g \circ \mathcal{D}_M} \log \hat{h}_M(\mathbf{z}) - \mathbb{E}_{\mathbf{z} \sim g \circ \hat{\mathcal{D}}_M} \log \hat{h}_M(\mathbf{z}) |$$

$$\tag{49}$$

Let $\Phi$ be a finite set such that every $\theta_1 \in \Theta$ is within distance $\frac{\epsilon}{4LM}$ of a $\theta_1 \in \Phi$, which is also termed a $\frac{\epsilon}{4LM}$-net. Standard construction given a $\Phi$ satisfying $\log |\Phi| \leq O(p \log(Lp/\epsilon))$, namely there aren't too many distinct discriminators in $\Phi$. By Chernoff bound, we have

$$\Pr\left[ \left| \mathbb{E}_{\mathbf{z} \sim g \circ \mathcal{D}_1} \log \hat{h}_1(\mathbf{z}) - \mathbb{E}_{\mathbf{z} \sim g \circ \hat{\mathcal{D}}_1} \log \hat{h}_1(\mathbf{z}) \right| \geq \frac{\epsilon}{2M} \right] \leq 2 \exp(-\frac{n^* \epsilon}{2M}) \tag{50}$$

Therefore, when $n^* \geq \frac{cpM \log(Lp/\epsilon)}{\epsilon}$ for large enough constant $c$, we can union bound over all $\theta_1 \in \Phi$. With probability at least $1 - \exp(-p)$, for all $\theta_1 \in \Phi$, we have $\left| \mathbb{E}_{\mathbf{z} \sim g \circ \mathcal{D}_1} \log \hat{h}_1(\mathbf{z}) - \mathbb{E}_{\mathbf{z} \sim g \circ \hat{\mathcal{D}}_1} \log \hat{h}_1(\mathbf{z}) \right| \leq \frac{\epsilon}{2M}$. Then for every $\theta_1 \in \Theta$, we can find a $\theta_1' \in \Phi$ such that $\|\theta_1 - \theta_1'\| \leq \epsilon/4LM$. Therefore

$$\left| \mathbb{E}_{\mathbf{z} \sim g \circ \mathcal{D}_1} \log \hat{h}_1(\theta_1; \mathbf{z}) - \mathbb{E}_{\mathbf{z} \sim g \circ \hat{\mathcal{D}}_1} \log \hat{h}_1(\theta_1; \mathbf{z}) \right|$$

$$\leq \left| \mathbb{E}_{\mathbf{z} \sim g \circ \mathcal{D}_1} \log \hat{h}_1(\theta_1'; \mathbf{z}) - \mathbb{E}_{\mathbf{z} \sim g \circ \hat{\mathcal{D}}_1} \log \hat{h}_1(\theta_1'; \mathbf{z}) \right|$$

$$+ \left| \mathbb{E}_{\mathbf{z} \sim g \circ \mathcal{D}_1} \log \hat{h}_1(\theta_1'; \mathbf{z}) - \mathbb{E}_{\mathbf{z} \sim g \circ \mathcal{D}_1} \log \hat{h}_1(\theta_1; \mathbf{z}) \right| \tag{51}$$

$$+ \left| \mathbb{E}_{\mathbf{z} \sim g \circ \hat{\mathcal{D}}_1} \log \hat{h}_1(\theta_1'; \mathbf{z}) - \mathbb{E}_{\mathbf{z} \sim g \circ \hat{\mathcal{D}}_1} \log \hat{h}_1(\theta_1; \mathbf{z}) \right|$$

$$\leq \frac{\epsilon}{2M} + \frac{\epsilon}{4M} + \frac{\epsilon}{4M} = \frac{\epsilon}{M}$$

Namely we have

$$| d_{\hat{h},g}(\mathcal{D}_1, \ldots, \mathcal{D}_M) - d_{\hat{h},g}(\hat{\mathcal{D}}_1, \ldots, \hat{\mathcal{D}}_M) | \leq M \times \frac{\epsilon}{M} = \epsilon \tag{52}$$

The result verifies that for the multi-domain adversarial training, the expectation over the empirical distribution converges to the expectation over the true distribution for all discriminators given enough data samples. □

## A.10 CONVERGENCE THEORY

In this subsection, we first provide some preliminaries before domain adversarial training convergence analysis. We then show simultaneous gradient descent DANN is not stable near the equilibrium but alternating gradient descent DANN could converge with a sublinear convergence rate, which support the importance of training encoder and discriminator separately. Finally, when incorporated with environment label smoothing, alternating gradient descent DANN is shown able to attain a faster convergence speed.

### A.10.1 PRELIMINARIES

The **asymptotic convergence analysis** is defined as applying the "ordinary differential equation (ODE) method" to analyze the convergence properties of dynamic systems. Given a discrete-time system characterized by the gradient descent:

$$F_\eta(\theta^t) := \theta^{t+1} = \theta^t + \eta h(\theta^t), \tag{53}$$

where $h(\cdot) : \mathbb{R} \to \mathbb{R}$ is the gradient and $\eta$ is the learning rate. The important technique for analyzing asymptotic convergence analysis is *Hurwitz condition* (Khalil., 1996): if the Jacobian of the dynamic system $A \triangleq h'(\theta)_{|\theta=\theta^*}$ at a stationary point $\theta^*$ is Hurwitz, namely the real part of every eigenvalue of $A$ is positive then the continuous gradient dynamics are asymptotically stable.

Given the same discrete-time system and Jacobian $A$, to ensure the **non-asymptotic convergence**, we need to provide an appropriate range of $\eta$ by solving $|1 + \lambda_i(A)| < 1, \forall \lambda_i \in Sp(A)$, where $Sp(A)$ is the spectrum of $A$. Namely, we can get constraint of the learning rate, which thus is able to evaluate the minimum number of iterations for an $\epsilon$-error solution and could more precisely reveal the convergence performance of the dynamic system than the asymptotic analysis (Nie & Patel, 2020).

**Theorem 4.** *(Proposition 4.4.1 in (Bertsekas, 1999).) Let $F : \Omega \to \Omega$ be a continuously differential function on an open subset $\Omega$ in $\mathbb{R}$ and let $\theta \in \Omega$ be so that*

*1. $F_\eta(\theta^*) = \theta^*$, and*

*2. the absolute values of the eigenvalues of the Jacobian $|\lambda_i| < 1, \forall \lambda_i \in Sp(F'_\eta(\theta^*))$.*

*Then there is an open neighborhood $U$ of $\theta^*$ so that for all $\theta^0 \in U$, the iterates $\theta^{k+1} = F_\eta(\theta^k)$ is locally converge to $\theta^*$. The rate of convergence is at least linear. More precisely, the error $\| \theta^k - \theta^* \|$ is in $\mathcal{O}(|\lambda_{max}|^k)$ for $k \to \infty$ where $\lambda_{max}$ is the eigenvalue of $F'_\eta(\theta^*)$ with the largest absolute value. When $|\lambda_i| > 1$, $F$ will not converge and when $|\lambda_i| = 1$, $F$ is either converge with a sublinear convergence rate or cannot converge.*

Finding fixed points of $F_\eta(\theta) = \theta + \eta h(\theta)$ is equivalent to finding solutions to the nonlinear equation $h(\theta) = 0$ and the Jacobian is given by:

$$F'_\eta(\theta) = I + \eta h'(\theta), \tag{54}$$

where both $F'_\eta(\theta), h'(\theta)$ are not symmetric and can therefore have complex eigenvalues. The following Theorem shows when a fixed point of $F$ satisfies the conditions of Theorem 4.

**Theorem 5.** *(Lemma 4 in (Mescheder et al., 2017).) Assume $A \triangleq h'(\theta)_{|\theta=\theta^*}$ only has eigenvalues with negative real-part and let $\eta > 0$, then the eigenvalues of the matrix $I + \eta A$ lie in the unit ball if and only if*

$$\eta < \frac{2a}{a^2 + b^2} = \frac{1}{|a|} \frac{2}{1 + (\frac{b}{a})^2}; \forall \lambda = -a + bi \in Sp(A) \tag{55}$$

Namely, both the maximum value of $a$ and $b/a$ determine the maximum possible learning rate. Although (Acuna et al., 2021) shows domain adversarial training is indeed a three-player game among classifier, feature encoder, and domain discriminator, it also indicates that the **complex eigenvalues with a large imaginary component are originated from encoder-discriminator adversarial training**. Hence here we only focus on the two-player zero-sum game between the feature encoder, and domain discriminator. One interesting thing is that, from non-asymptotic convergence analysis, we can get a result (Theorem 5) that is very similar to that from the Hurwitz condition (Corollary 1 in (Acuna et al., 2021): $\eta < \frac{-2a}{b^2-a^2}; \forall \lambda = a + bi \in Sp(A)$ and $|a| < |b|$).

### A.10.2 A SIMPLE ADVERSARIAL TRAINING EXAMPLE

According to Ali Rahimi's test of times award speech at NIPS 17, *simple experiments, simple theorems are the building blocks that help us understand more complicated systems*. Along this line, we propose this toy example to understand the convergence of domain adversarial training. Denote $\mathcal{D}_S = x_s, \mathcal{D}_t = x_t$ two Dirac distribution where both $x_1$ and $x_2$ are float number. In this setting, both the encoder and discriminator have exactly one parameter, which is $\theta_e, \theta_d$ respectively[3]. The DANN training objective in Equ. (7) is given by

$$d_\theta = f(\theta_d \theta_e x_s) + f(-\theta_d \theta_e x_t), \tag{56}$$

where $f(t) = \log\left(1/(1 + \exp(-t))\right)$ and **the unique equilibrium point of the training objective in Equ. (56) is given by $\theta_e^* = \theta_d^* = 0$**. We then recall the update operators of simultaneous and alternating Gradient Descent, for the former, we have

$$F_\eta(\theta) = \begin{pmatrix} \theta_e - \eta \nabla_{\theta_e} d_\theta \\ \theta_d + \eta \nabla_{\theta_d} d_\theta \end{pmatrix} \tag{57}$$

For the latter, we have $F_\eta = F_{\eta,2}(\theta) \circ F_{\eta,1}(\theta)$, and $F_{\eta,1}, F_{\eta,2}$ are defined as

$$F_{\eta,1}(\theta) = \begin{pmatrix} \theta_e - \eta \nabla_{\theta_e} d_\theta \\ \theta_d \end{pmatrix}, F_{\eta,2}(\theta) = \begin{pmatrix} \theta_e \\ \theta_d + \eta \nabla_{\theta_d} d_\theta \end{pmatrix}, \tag{58}$$

If we update the discriminator $n_d$ times after we update the encoder $n_e$ times, then the update operator will be $F_\eta = F_{\eta,1}^{n_e}(\theta) \circ F_{\eta,1}^{n_d}(\theta)$. To understand convergence of simultaneous and alternating gradient descent, we have to understand when the Jacobian of the corresponding update operator has only eigenvalues with absolute value smaller than 1.

---

[3]One may argue that neural networks are non-linear, but *Theorem 4.5 from (Khalil., 1996)* shows that one can "linearize" any non-linear system near equilibrium and analyze the stability of the linearized system to comment on the local stability of the original system.

### A.10.3 SIMULTANEOUS GRADIENT DESCENT DANN

**Proposition 7.** *The unique equilibrium point of the training objective in Equ. (56) is given by* $\theta_e^* = \theta_d^* = 0$. *Moreover, the Jacobian of* $F_\eta(\theta) = \begin{pmatrix} \theta_e - \eta\nabla_{\theta_e} d_\theta \\ \theta_d + \eta\nabla_{\theta_d} d_\theta \end{pmatrix}$ *at the equilibrium point has the two eigenvalues*

$$\lambda_{1/2} = 1 \pm \frac{\eta}{2}|x_s - x_t|i, \tag{59}$$

*namely* $F_\eta(\theta)$ *will never satisfies the second conditions of Theorem 4 whatever $\eta$ is, which shows that this continuous system is generally not linearly convergent to the equilibrium point.*

*Proof.* The Jacobian of $F_\eta(\theta) = \begin{pmatrix} \theta_e - \eta\nabla_{\theta_e} d_\theta \\ \theta_d + \eta\nabla_{\theta_d} d_\theta \end{pmatrix}$ is

$$
\begin{aligned}
\nabla_\theta F_\eta(\theta) &= \nabla_\theta \begin{pmatrix} \theta_e - \eta\left(\theta_d x_s f'(\theta_d\theta_e x_s) - \theta_d x_t f'(\theta_d\theta_e x_t)\right) \\ \theta_d + \eta\left(\theta_e x_s f'(\theta_d\theta_e x_s) - \theta_e x_t f'(\theta_d\theta_e x_t)\right) \end{pmatrix} \\
&= \begin{pmatrix} 1 & -\eta\left(x_s f'(\theta_d\theta_e x_s) - x_t f'(\theta_d\theta_e x_t)\right) \\ \eta\left(x_s f'(\theta_d\theta_e x_s) - x_t f'(\theta_d\theta_e x_t)\right) & 1 \end{pmatrix} \\
&= \begin{pmatrix} 1 & -\frac{\eta}{2}\left(x_s - x_t\right) \\ \frac{\eta}{2}\left(x_s - x_t\right) & 1 \end{pmatrix},
\end{aligned}
\tag{60}
$$

The derivation result of $\nabla_{\theta_e}\theta_e - \eta\left(\theta_d x_s f'(\theta_d\theta_e x_s) - \theta_d x_t f'(\theta_d\theta_e x_t)\right)$ should have been

$$1 - \eta\left(\theta_d^2 x_s^2 f''(\theta_d\theta_e x_s) - \theta_d^2 x_t^2 f''(\theta_d\theta_e x_t)\right) \tag{61}$$

Since the equilibrium point $(\theta_e^*, \theta_d^*) = (0,0)$, for points near the equilibrium, we ignore high-order infinitesimal terms *e.g.*, $\theta_e^2, \theta_d^2, \theta_e\theta_d$. We can thus obtain the derivation of the second line. The eigenvalues of the second-order matrix $A = \begin{pmatrix} a & b \\ c & d \end{pmatrix}$ are $\lambda = \frac{a+d\pm\sqrt{(a+d)^2-4(ad-bc)}}{2}$, and then the eigenvalues of $\nabla_\theta F_\eta(\theta)$ is $1 \pm \frac{\eta}{2}|x_s - x_t|i$. Obviously $|\lambda| > 1$ and the proposition is completed. □

### A.10.4 ALTERNATING GRADIENT DESCENT DANN

**Proposition 8.** *The unique equilibrium point of the training objective in Equ. (56) is given by* $\theta_e^* = \theta_d^* = 0$. *If we update the discriminator $n_d$ times after we update the encoder $n_e$ times. Moreover, the Jacobian of* $F_\eta = F_{\eta,2}(\theta) \circ F_{\eta,1}(\theta)$ *(Equ. (58)) has eigenvalues*

$$\lambda_{1/2} = 1 - \frac{\alpha^2}{2} \pm \sqrt{\left(1 - \frac{\alpha^2}{2}\right)^2 - 1}, \tag{62}$$

*where* $\alpha = \frac{1}{2}\sqrt{n_d n_e}\eta|x_s - x_t|$. $|\lambda_{1/2}| = 1$ *for* $\eta \leq \frac{4}{\sqrt{n_e n_d}|x_s-x_t|}$ *and* $|\lambda_{1/2}| > 1$ *otherwise. Such result indicates that although alternating gradient descent does not converge linearly to the Nash-equilibrium, it could converge with a sublinear convergence rate.*

*Proof.* The Jacobians of alternating gradient descent DANN operators (Equ. (58)) near the equilibrium are given by:

$$\nabla_\theta F_{\eta,1}(\theta) = \begin{pmatrix} 1 & -\eta\left(x_s f'(\theta_d\theta_e x_s) - x_t f'(\theta_d\theta_e x_t)\right) \\ 0 & 1 \end{pmatrix} = \begin{pmatrix} 1 & -\frac{\eta}{2}\left(x_s - x_t\right) \\ 0 & 1 \end{pmatrix}, \tag{63}$$

similarly we can get $\nabla_\theta F_{\eta,2}(\theta) = \begin{pmatrix} 1 & 0 \\ \frac{\eta}{2}\left(x_s - x_t\right) & 1 \end{pmatrix}$. As a result, the Jacobian of the combined update operator $\nabla_\theta F_\eta(\theta)$ is

$$\nabla_\theta F_\eta(\theta) = \nabla_\theta F_{\eta,2}^{n_e}(\theta)\nabla_\theta F_{\eta,1}^{n_d}(\theta) = \begin{pmatrix} 1 & -\frac{\eta n_e}{2}\left(x_s - x_t\right) \\ \frac{\eta n_d}{2}\left(x_s - x_t\right) & -\frac{\eta n_d n_e}{4}\left(x_s - x_t\right)^2 + 1 \end{pmatrix}. \tag{64}$$

An easy calculation shows that the eigenvalues of this matrix are

$$\lambda_{1/2} = 1 - \frac{n_e n_d}{8}\eta^2(x_s - x_t)^2 \pm \sqrt{\left(1 - \frac{n_e n_d}{8}\eta^2(x_s - x_t)^2\right)^2 - 1} \tag{65}$$

Let $\alpha = \frac{1}{2}\sqrt{n_d n_e}\eta|x_s - x_t|$, we can get $\lambda_{1/2} = 1 - \frac{\alpha^2}{2} \pm \sqrt{\left(1 - \frac{\alpha^2}{2}\right)^2 - 1}$. If $\left(1 - \frac{\alpha^2}{2}\right)^2 > 1$, namely $\alpha > 2$, then $|\lambda_{1/2}| = \sqrt{2\left(1 - \frac{\alpha^2}{2}\right)^2 - 1}$. To satisfy $|\lambda| < 1$, we have $\left(1 - \frac{\alpha^2}{2}\right)^2 < 1$, which conflicts with the assumption. That is $\alpha \leq 2$, and in this case $|\lambda_{1/2}| = 1$. $\qquad\square$

### A.10.5  Alternating gradient descent DANN+ELS

Incorporate environment label smoothing to Equ. (56), the target is revised into:

$$d_{\theta,\gamma} = \gamma f(\theta_d \theta_e x_s) + (1-\gamma) f(-\theta_d \theta_e x_s) + \gamma f(-\theta_d \theta_e x_t) + (1-\gamma) f(\theta_d \theta_e x_t), \qquad (66)$$

**Proposition 9.** *The unique equilibrium point of the training objective in Equ. (66) is given by* $\theta_e^* = \theta_d^* = 0$. *If we update the discriminator $n_d$ times after we update the encoder $n_e$ times. Moreover, the Jacobian of $F_\eta = F_{\eta,2}(\theta) \circ F_{\eta,1}(\theta)$ (Equ. (58)) has eigenvalues*

$$\lambda_{1/2} = 1 - \frac{\alpha^2}{2} \pm \sqrt{\left(1 - \frac{\alpha^2}{2}\right)^2 - 1}, \qquad (67)$$

*where* $\alpha = \frac{2\gamma-1}{2}\sqrt{n_d n_e}\eta|x_s - x_t|$. $|\lambda_{1/2}| = 1$ *for* $\eta \leq \frac{4}{\sqrt{n_d n_e}|x_s - x_t|}\frac{1}{2\gamma-1}$ *and* $|\lambda_{1/2}| > 1$ *otherwise. Such result indicates that alternating gradient descent DANN+ELS could converge faster than alternating gradient descent DANN.*

*Proof.* The operator for alternating gradient descent DANN+ELS is $F_\eta = F_{\eta,2}(\theta) \circ F_{\eta,1}(\theta)$, and $F_{\eta,1}, F_{\eta,2}$ near the equilibrium are given by:

$$F_{\eta,1}(\theta) = \begin{pmatrix} \theta_e - \eta\nabla_{\theta_e}d_{\theta,\gamma} \\ \theta_d \end{pmatrix} = \begin{pmatrix} \theta_e - \eta\left(\gamma\theta_d x_s f'(0) - (1-\gamma)\theta_d x_s f'(0) - \gamma\theta_d x_t f'(0) + (1-\gamma)\theta_d x_t f'(0)\right) \\ \theta_d \end{pmatrix}$$
$$F_{\eta,2}(\theta) = \begin{pmatrix} \theta_e \\ \theta_d + \eta\nabla_{\theta_d}d_{\theta,\gamma} \end{pmatrix} = \begin{pmatrix} \theta_e \\ \theta_d + \eta\left(\gamma\theta_e x_s f'(0) - (1-\gamma)\theta_e x_s f'(0) - \gamma\theta_e x_t f'(0) + (1-\gamma)\theta_e x_t f'(0)\right) \end{pmatrix},$$
$$(68)$$

The Jacobians of alternating gradient descent DANN+ELS operators near the equilibrium are given by:

$$\nabla_\theta F_{\eta,1}(\theta) = \begin{pmatrix} 1 & -\frac{\eta(2\gamma-1)}{2}(x_s - x_t) \\ 0 & 1 \end{pmatrix}, \nabla_\theta F_{\eta,2}(\theta) = \begin{pmatrix} 1 & 0 \\ \frac{\eta(2\gamma-1)}{2}(x_s - x_t) & 1 \end{pmatrix}, \qquad (69)$$

As a result, the Jacobian of the combined update operator $\nabla_\theta F_\eta(\theta)$ is

$$\nabla_\theta F_\eta(\theta) = \nabla_\theta F_{\eta,2}^{n_e}(\theta)\nabla_\theta F_{\eta,1}^{n_d}(\theta) = \begin{pmatrix} 1 & -\frac{\eta n_e(2\gamma-1)}{2}(x_s - x_t) \\ \frac{\eta n_d(2\gamma-1)}{2}(x_s - x_t) & -\frac{\eta n_d n_e(2\gamma-1)^2}{4}(x_s - x_t)^2 + 1 \end{pmatrix}. \qquad (70)$$

An easy calculation shows that the eigenvalues of this matrix are

$$\lambda_{1/2} = 1 - \frac{n_e n_d}{8}\eta^2(2\gamma-1)^2(x_s - x_t)^2 \pm \sqrt{\left(1 - \frac{n_e n_d}{8}\eta^2(2\gamma-1)^2(x_s - x_t)^2\right)^2 - 1} \qquad (71)$$

Similarly to the proof of Proposition 8, let $\alpha = \frac{2\gamma-1}{2}\sqrt{n_d n_e}\eta|x_s - x_t|$, we can get $\lambda_{1/2} = 1 - \frac{\alpha^2}{2} \pm \sqrt{\left(1 - \frac{\alpha^2}{2}\right)^2 - 1}$. Only when $\alpha \leq 2$, $\lambda_{1/2}$ are on the unit circle, namely $\eta \leq \frac{4}{\sqrt{n_d n_e}|x_s - x_t|}\frac{1}{2\gamma-1}$. Compared to the result in Proposition 8, which is $\eta \leq \frac{4}{\sqrt{n_d n_e}|x_s - x_t|}$, the additional $\frac{1}{2\gamma-1} > 1$ enables us to choose more large learning rate and could converge to an small error solution by fewer iterations. $\qquad\square$

## B  Extended Related Works

**Domain adaptation and domain generalization** (Muandet et al., 2013; Sagawa et al., 2019; Li et al., 2018a; Blanchard et al., 2021; Li et al., 2018b; Zhang et al., 2021a; 2022c) aims to learn a model

that can extrapolate well in unseen environments. Representative methods like AT method (Ganin et al., 2016) proposed the idea of learning domain-invariant representations as an adversarial game. This approach led to a plethora of methods including state-of-the-art approaches (Zhang et al., 2019; Acuna et al., 2021; 2022). In this paper, we propose a simple but effective trick, ELS, which benefits the generalization performance of methods by using soft environment labels.

**Adversarial Training in GANs** is well studied and many theoretical results of GANs motivate the analysis in this paper. *e.g.,* divergence minimization interpretation (Goodfellow et al., 2014; Nguyen et al., 2017), generalization of the discriminator (Arora et al., 2017; Thanh-Tung et al., 2019), training stability (Thanh-Tung et al., 2019; Schäfer et al., 2019; Arjovsky & Bottou, 2017; Arjovsky et al., 2017), nash equilibrium (Farnia & Ozdaglar, 2020; Nagarajan & Kolter, 2017), and gradient descent in GAN optimization (Nagarajan & Kolter, 2017; Gidel et al., 2018; Chen et al., 2018). Multi-domain image generation is also related to this work, generalization to the JSD metric has been explored to address this challenge (Gan et al., 2017; Pu et al., 2018; Trung Le et al., 2019). However, most of them have to build $\frac{M(M-1)}{2}$ pairwise critics, which is expensive when $M$ is large. $\chi^2$ GAN (Tao et al., 2018) firstly attempts to tackle the challenge and only needs $M-1$ critics.

## C  ADDITIONAL EXPERIMENTAL SETUPS

### C.1  DATASET DETAILS AND EXPERIMENTAL SETTINGS

In this subsection, we introduce all the used datasets and the hyper-parameters for reproducing the experimental results in this work. We have uploaded the codes for all experiments in the supplementary materials to make sure that all the results are reproducible. All the main hyper-parameters for reproducing the experimental results in this work are shown in Table 9.

#### C.1.1  IMAGES CLASSIFICATION DATASETS

**Experimental settings.** *For DG and multi-source DG tasks*, all the baselines are implemented using the codebase of Domainbed (Gulrajani & Lopez-Paz, 2021) and we use as encoders ConvNet for RotatedMNIST (detailed in Appdendix D.1 in (Gulrajani & Lopez-Paz, 2021)) and ResNet-50 for the remaining datasets. The model selection that we use is test-domain validation, one of the three selection methods in (Gulrajani & Lopez-Paz, 2021). That is, we choose the model maximizing the accuracy on a validation set that follows the same distribution of the test domain. *For DA tasks*, all baselines implementation and hyper-parameters follows (Wang & Hou). For *Continuously Indexed Domain Adaptation tasks*, all baselines are implemented using PyTorch with the same architecture as (Wang et al., 2020). Note that although our theoretical analysis on non-asymptotic convergence is based on alternating Gradient Descent, current DA methods mainly build on Gradient Reverse Layer. For a fair comparison, in our experiments considering domain adaptation benchmarks, we also use GRL as default and let the analysis in future work.

**Rotated MNIST** (Ghifary et al., 2015) consists of 70,000 digits in MNIST with different rotated angles where domain is determined by the degrees $d \in \{0, 15, 30, 45, 60, 75\}$.

**PACS** (Li et al., 2017b) includes 9,991 images with 7 classes $y \in \{$ dog, elephant, giraffe, guitar, horse, house, person $\}$ from 4 domains $d \in \{$art, cartoons, photos, sketches$\}$.

**VLCS** (Torralba & Efros, 2011) is composed of 10,729 images, 5 classes $y \in \{$ bird, car, chair, dog, person $\}$ from domains $d \in \{$Caltech101, LabelMe, SUN09, VOC2007$\}$.

**Office-31** (Saenko et al., 2010) contains contains 4,110 images, 31 object categories in three domains: $d \in \{$ Amazon, DSLR, and Webcam$\}$.

**Office-Home** (Venkateswara et al., 2017): consists of 15,500 images from 65 classes and 4 domains: $d \in \{$ Art (Ar), Clipart (Cl), Product (Pr) and Real World (Rw) $\}$.

**Rotating MNIST** (Wang et al., 2020) is adapted from regular MNIST digits with mild rotation to significantly Rotating MNIST digits. In our experiments, $[0°, 45°)$ is set as the source domain and others are unlabeled target domains. The chosen baselines include Adversarial Discriminative Domain Adaptation (ADDA (Tzeng et al., 2017)), and CIDA (Wang et al., 2020). ADDA merges data with different domain indices into one source and one target domain. DANN divides the continuous

domain spectrum into several separate domains and performs adaptation between multiple source and target domains. For Rotating MNIST, the seven target domains contain images rotating by $d \in \{[0°, 45°), [45°, 90°), [90°, 135°), \ldots, [315°, 360°)\}$ degrees, respectively.

**Circle Dataset** (Wang et al., 2020) includes 30 domains indexed from 1 to 30 and Figure 5(a) shows the 30 domains in different colors (from right to left is $1, \ldots, 30$ respectively). Each domain contains data on a circle and the task is binary classification. Figure 5(b) shows positive samples as red dots and negative samples as blue crosses. In our experiments, We use domains 1 to 6 as source domains and the rest as target domains.

### C.1.2   IMAGE RETRIEVAL DATASETS

**Experimental settings.** Following previous generalizable person ReID methods, we use Mo-bileNetV2 (Sandler et al., 2018) with a multiplier of $1.4$ as the backbone network, which is pretrained on ImageNet (Deng et al., 2009). Images are resized to $256 \times 128$ and the training batch size $N$ is set to 80. The SGD optimizer is used to train all the components with a learning rate of $0.01$, a momentum of $0.9$ and a weight decay of $5 \times 10^{-4}$. The learning rate is warmed up in the first 10 epochs and decayed to its $0.1\times$ and $0.01\times$ at 40 and 70 epochs.

We evaluate the proposed method by Person re-identification (ReID) tasks, which aims to find the correspondences between person images from the same identity across multiple camera views. The training datasets include CUHK02 (Li & Wang, 2013), CUHK03 (Li et al., 2014), Market1501 (Zheng et al., 2015), DukeMTMC-ReID (Zheng et al., 2017), and CUHK-SYSU PersonSearch (Xiao et al., 2016). The unseen test domains are VIPeR (Gray et al., 2007), PRID (Hirzer et al., 2011), QMUL GRID (Liu et al., 2012), and i-LIDS (Wei-Shi et al., 2009). Details of the training datasets are summarized in Table 10 and the test datasets are summarized in Table 11. All the assets (*i.e.,* datasets and the codes for baselines) we use include a MIT license containing a copyright notice and this permission notice shall be included in all copies or substantial portions of the software.

**GRID** (Liu et al., 2012) contains 250 probe images and 250 true match images of the probes in the gallery. Besides, there are a total of 775 additional images that do not belong to any of the probes. We randomly take out 125 probe images. The remaining 125 probe images and $1025(775 + 250)$ images in the gallery are used for testing.

**i-LIDS** (Wei-Shi et al., 2009) has two versions, images and sequences. The former is used in our experiments. It involves 300 different pedestrian pairs observed across two disjoint camera views 1 and 2 in public open space. We randomly select 60 pedestrian pairs, two images per pair are randomly selected as probe image and gallery image respectively.

**PRID2011** (Hirzer et al., 2011) has single-shot and multi-shot versions. We use the former in our experiments. The single-shot version has two camera views $A$ and $B$, which capture 385 and 749 pedestrians respectively. Only 200 pedestrians appear in both views. During the evaluation, 100 randomly identities presented in both views are selected, the remaining 100 identities in view $A$ constitute probe set and the remaining 649 identities in view $B$ constitute gallery set.

**VIPeR** (Gray et al., 2007) contains 632 pedestrian image pairs. Each pair contains two images of the same individual seen from different camera views 1 and 2. Each image pair was taken from an arbitrary viewpoint under varying illumination conditions. To compare to other methods, we randomly select half of these identities from camera view 1 as probe images and their matched images in view 2 as gallery images.

We follow the single-shot setting. The average rank-k (R-k) accuracy and mean Average Precision ($m$AP) over 10 random splits are reported based on the evaluation protocol

### C.1.3   NEURAL LANGUAGE DATASETS

**CivilComments-Wilds** (Koh et al., 2021) contains $448,000$ comments on online articles taken from the Civil Comments platform. The input is a text comment and the task is to predicate whether the comment was rated as toxic, *e.g.,* , the comment *Maybe you should learn to write a coherent sentence so we can understand WTF your point is* is rated as toxic and *I applaud your father. He was a good man! We need more like him.* is not. Domain in CivilComments-Wilds dataset is an 8-dimensional

binary vector where each component corresponds to whether the comment mentions one of the 8 demographic identities {male, female, LGBTQ, Christian, Muslim, other religions, Black, White}.

**Amazon-Wilds** (Koh et al., 2021) contains $539,520$ reviews from disjoint sets of users. The input is the review text and the task is to predict the corresponding 1-to-5 star rating from reviews of Amazon products. Domain $d$ identifies the user who wrote the review and the training set has $3,920$ domains. The 10-th percentile of per-user accuracies metric is used for evaluation, which is standard to measure model performance on devices and users at various percentiles in an effort to encourage good performance across many devices.

### C.1.4 Genomics and Graph datasets

**RxRx1-wilds** (Koh et al., 2021) comprises images of cells that have been genetically perturbed by siRNA, which comprises $125,510$ images of cells obtained by fluorescent microscopy. The output $y$ indicates which of the $1,139$ genetic treatments (including no treatment) the cells received, and $d$ specifies 51 batches in which the imaging experiment was run.

**OGB-MolPCBA** (Koh et al., 2021) is a multi-label classification dataset, which comprises $437,929$ molecules with $120,084$ different structural scaffolds. The input is a molecular graph, the label is a 128-dimensional binary vector where each component corresponds to a biochemical assay result, and the domain $d$ specifies the scaffold (*i.e.,* a cluster of molecules with similar structure). The training and test sets contain molecules with disjoint scaffolds; The training set has molecules from over $40,000$ scaffolds. We evaluate models by averaging the Average Precision (AP) across each of the 128 assays.

### C.1.5 Sequential data

**Spurious-Fourier** (Gagnon-Audet et al., 2022) is a binary classification dataset ($y \in$ {low-frequency peak (L) and high-frequency peak (H).}), which is composed of one-dimensional signal. Domains $d \in \{10\%, 80\%, 90\%\}$ contain signal-label pairs, where the label is a noisy function of the low- and high-frequencies such that low-frequency peaks bear a varying correlation of $d$ with the label and high-frequency peaks bear an invariant correlation of $75\%$ with the label.

**HHAR** (Gagnon-Audet et al., 2022) is a 6 activities classification dataset ($y \in$ {Stand, Sit, Walk, Bike, Stairs up, and Stairs Down }), which is composed of recordings of 3-axis accelerometer and 3-axis gyroscope data. Specifically, the input $x$ is recordings of 500 time-steps of a 6-dimensional signal sampled at 100Hz. Domain $d$ consist of five smart device models: $d \in$ {Nexus 4, Galaxy S3, Galaxy S3 Mini, LG Watch, and Samsung Galaxy Gears}.

### C.2 Backbone Structures

Most of the backbones are ResNet-50/ResNet-18 and we follow the same setting as the reference works. Here we briefly introduce some special backbones used in our experiments,*i.e.,* ConvNet for Rotated MNIST, EncoderSTN for Rotating MNIST, DistillBERT for Neural Language datasets, and GIN for OGB-MoIPCBA.

**MNIST ConvNet.** is detailed in Table. 12.

**DistillBERT.** We use the implementation from (Wolf et al., 2019) and finetune a BERT-base-uncased models for neural language datasets. **EncoderSTN** use a four-layer convolutional neural network for the encoder and a three-layer MLP to make the prediction. The domain discriminator is a four-layer MLP. The encoder is incorporated with a Spacial Transfer Network (STN) (Jaderberg et al., 2015), which takes the image and the domain index as input and outputs a set of rotation parameters which are then applied to rotate the given image.

**Graph Isomorphism Networks (GIN) (Xu et al., 2018)** combined with virtual nodes is used for OGB-MoIPCBA dataset, as this is currently the model with the highest performance in the Open Graph Benchmark.

**Deep ConvNets (Schirrmeister et al., 2017)** for HHAR combines temporal and spatial convolution,which fits this data well and we use the implementation in the BrainDecode Schirrmeister (Schirrmeister et al., 2017) Toolbox.

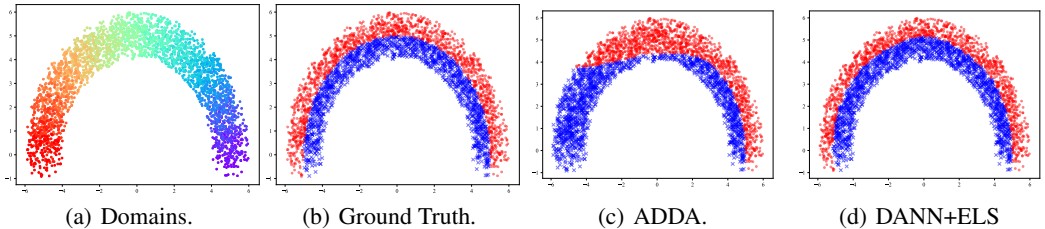

|     (a) Domains.     |     (b) Ground Truth.     |     (c) ADDA.     |     (d) DANN+ELS     |

Figure 5: **Results on the *Circle* dataset with 30 domains.** (a) shows domain index by color, (b) shows label index by color, where red dots and blue crosses are positive and negative data sample. Source domains contain the first 6 domains and others are target domains.

# D    ADDITIONAL EXPERIMENTAL RESULTS

## D.1    ADDITIONAL NUMERICAL RESULTS

**Multi-Source Domain Generalization.** IRM (Arjovsky et al., 2019) introduces specific conditions for an upper bound on the number of training environments required such that an invariant optimal model can be obtained, which stresses the importance of the number of training environments. In this paper, we reduce the training environments on the Rotated MNIST from five to three. As shown in Table 17, as the number of training environment decreases, the performance of IRM fall sharply (*e.g.,* the averaged accuracy from $97.5\%$ to $91.8\%$), and the performance on the most challenging domains $d = \{0, 5\}$ decline the most ($94.9\% \rightarrow 80.9\%$ and $95.2\% \rightarrow 91.1\%$). In contrast, both ERM and DANN+ELS retain high generalization performances and DANN+ELS outperforms ERM in most domains.

**Image Retrieval.** We compare the proposed DANN+ELS with methods on a typical DG-ReID setting. As shown in Table 16, we implement DANN with various hyper-parameters while DANN always fails to converge on ReID benchmarks. As illustrated in Appendix Figure 8, we compare the training statistics with the baseline, where DANN is highly unstable and attains inferior results. However, equipped with ELS and following the same hyper-parameter as DANN, DANN+ELS attains well-training stability and achieves either comparable or better performance when compared with recent state-of-the-art DG-ReID methods. See Appendix D.2 for t-sne visualization and comparison.

Table 14: Domain generalization performance on the OGB-MolPCBA dataset.     Table 15: Domain generalization performance on the Spurious-Fourier dataset.

| OGB-MolPCBA | | |
|---|---|---|
| **Algorithm** | **Val Avg Acc** | **Test Avg Acc** |
| ERM (Vapnik, 1999) | **27.8 ± 0.1** | **27.2 ± 0.3** |
| Group DRO Sagawa et al. (2019) | 23.1 ± 0.6 | 22.4 ± 0.6 |
| CORAL Sun & Saenko (2016) | 18.4 ± 0.2 | 17.9 ± 0.5 |
| IRM (Arjovsky et al., 2019) | 15.8 ± 0.2 | 15.6 ± 0.3 |
| DANN (Ganin et al., 2016) | 15.0 ± 0.6 | 14.1 ± 0.5 |
| DANN+ELS | 18.0 ± 0.3 | 17.2 ± 0.3 |
| ↑ | 3.0 | 3.1 |

| Spurious-Fourier dataset | | |
|---|---|---|
| **Algorithm** | **Train validation** | **Test validation** |
| ERM (Vapnik, 1999) | 9.7 ± 0.3 | 9.3 ± 0.1 |
| IRM (Arjovsky et al., 2019) | 9.3 ± 0.1 | 57.6 ± 0.8 |
| SD Pezeshki et al. (2021) | 10.2 ± 0.1 | 9.2 ± 0.0 |
| VREx Krueger et al. (2021) | 9.7 ± 0.2 | **65.3 ± 4.8** |
| DANN (Ganin et al., 2016) | 9.7 ± 0.1 | 11.1 ± 1.5 |
| DANN+ELS | **10.7 ± 0.6** | 15.6 ± 2.8 |
| ↑ | 1.0 | 4.5 |

## D.2    ADDITIONAL ANALYSIS AND INTERPRETATION

**T-sne visualization.** We compare the proposed DANN+ELS with MetaBIN and ERM through $t$-SNE visualization. We observe a distinct division of different domains in Figure 7(a) and Figure 7(d), which indicates that a domain-specific feature space is learned by the ERM. MetaBIN perform better than ERM and the proposed DANN+ELS can learn more domain-invariant representations while keeping discriminative capability for ReID tasks.

## D.3    ABLATION STUDIES

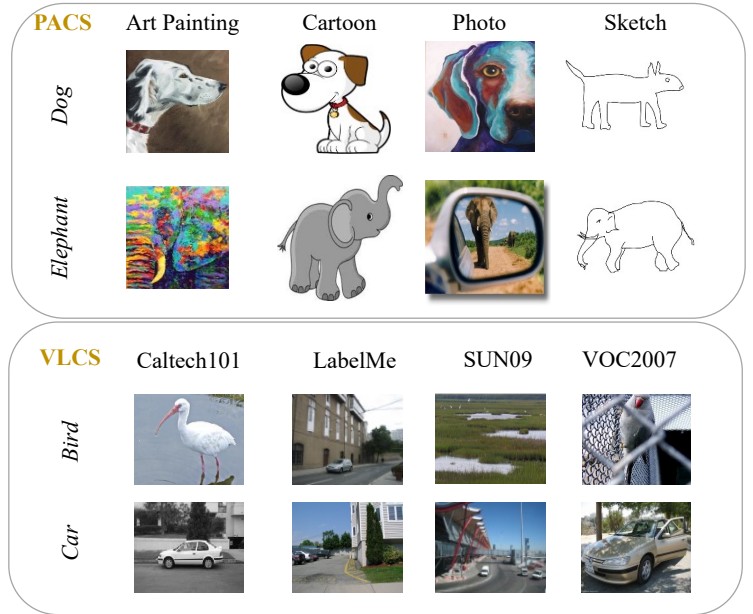

Figure 6: Data examples from the PACS and the VLCS datasets.

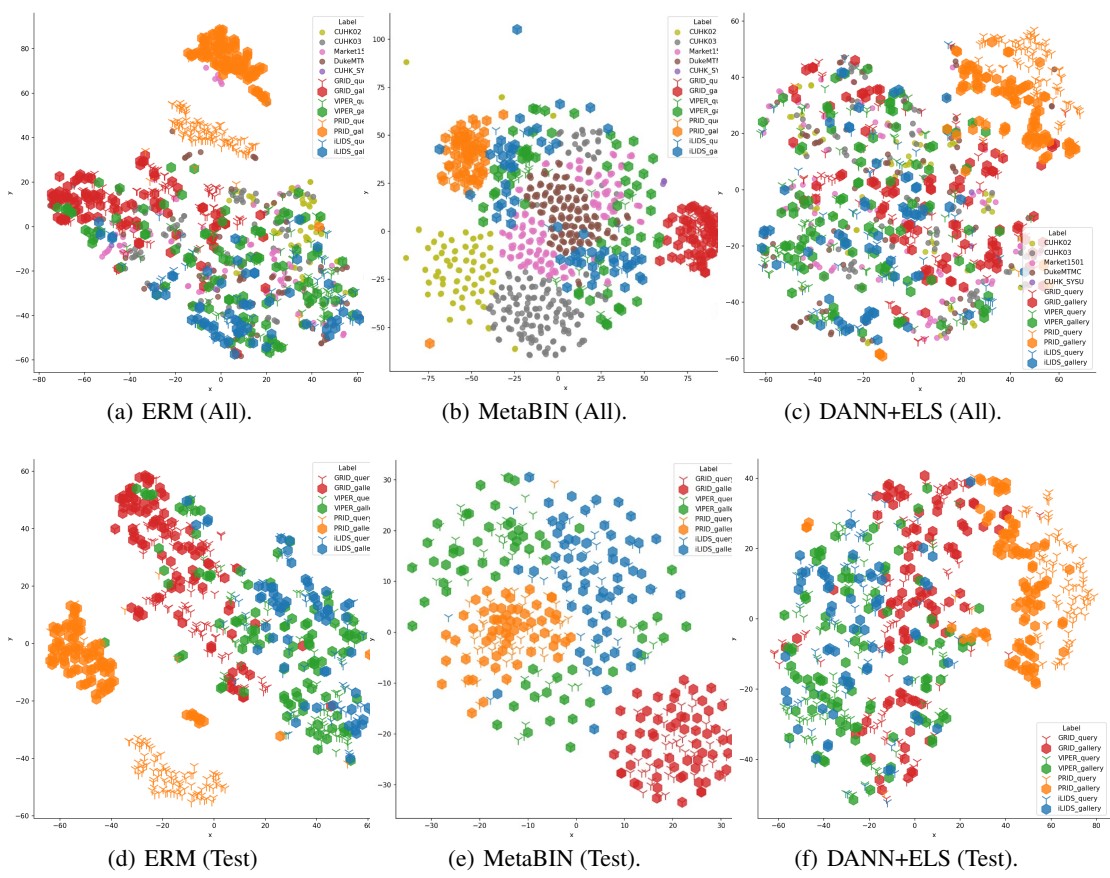

Figure 7: **Visualization of the embeddings on training and test datasets.** Query and gallery samples of these unseen datasets are shown using different types of mark. Best viewed in color.

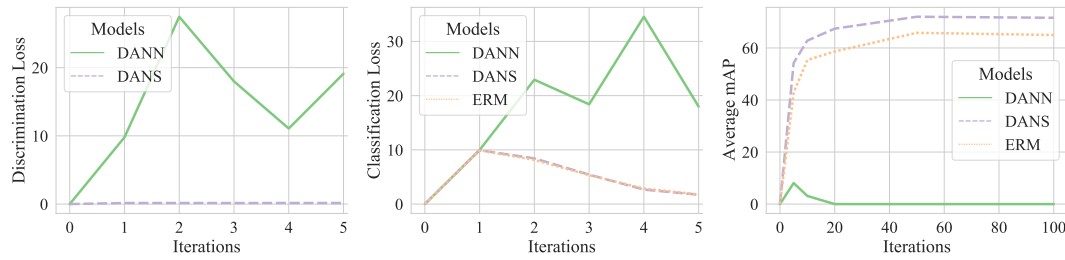

| (a) Domain discrimination loss. | (b) Identity classification loss. | (c) Average $m$AP on the test set. |
|---|---|---|

Figure 8: **Training statistics on ReID datasets**.

Table 9: Hyper-parameters for different benchmarks. $Lr_g, Decay_g$: learning rate and weight decay for the encoder and classifier; $Lr_d, Decay_d$: learning rate and weight decay for the domain discriminator; $bsz$: batch size during training; $d_{steps}$: the discriminator is trained $d_{steps}$ times once the encoder and classifier are trained; $W_{reg}$: tradeoff weight for the gradient penalty; $\lambda$: tradeoff weight for the adversarial loss. The default $\beta_2$ for Adam and AdamW optimizer is 0.99 and the momentum for SGD optimizer is 0.9. **/** means domain discriminators trained on the dataset use GRL but not alternating gradient descent.

| Task | Datsets | $Lr_g$ | $Lr_d$ | $\beta_1$ | $Decay_g$ | $Decay_d$ | **bsz** | $d_{steps}$ | $W_{reg}$ | $\lambda$ |
|---|---|---|---|---|---|---|---|---|---|---|
| | Rotated MNIST | 1E-03 | 1E-03 | 0.5 | 0E+00 | 0.0 | 64 | 1 | 1 | 0.5 |
| | PACS | 5E-05 | 5E-05 | 0.5 | 0E+00 | 0.0 | 32 | 5 | 1 | 0.5 |
| Images | VLCS | 5E-05 | 5E-05 | 0.5 | 0E+00 | 0.0 | 32 | 5 | 1 | 0.5 |
| Classification | Office-31(ResNet50) | 1E-02 | 1E-02 | SGD | 1E-3 | 1E-3 | 32 | / | 0 | 1.00 |
| | Office-Home (ResNet50) | 1E-02 | 1E-02 | SGD | 1E-3 | 1E-3 | 32 | / | 0 | 1.00 |
| | Office-31 (ViT) | 2E-03 | 2E-03 | SGD | 1E-3 | 1E-3 | 24 | / | 0 | 1.00 |
| | Office-Home (ViT) | 2E-03 | 2E-03 | SGD | 1E-3 | 1E-3 | 24 | / | 0 | 1.00 |
| | Rotating MNIST | 2E-04 | 2E-04 | 0.9 | 5E-04 | 5E-04 | 100 | 1 | 0 | 2.00 |
| Image Retrieval | MS | 1E-02 | 1E-02 | SGD | 5E-04 | 5E-04 | 80 | 1 | 0 | 1.00 |
| Neural Language | CivilComments | 1E-05 | 1E-05 | SGD | 1E-02 | 0 | 16 | 1 | 0 | 1.00 |
| Processing | Amazon | 1E-04 | 2E-04 | AdamW | 1E-02 | 0 | 8 | 1 | 0 | 0.11 |
| Genomics | RxRx1 | 1E-04 | 2E-04 | 0.9 | 1E-05 | 0 | 72 | 1 | 0 | 0.11 |
| and Graph | OGB-MolPCBA | 8E-04 | 8E-04 | 0.9 | 1E-05 | 0 | 32 | 1 | 0 | 0.11 |
| Sequential | Spurious-Fourier | 4E-04 | 4E-04 | 0 | 1E-03 | 0 | 78 | 3 | 1.25 | 1.56 |
| Prediction | HHAR | 3E-03 | 1E-03 | 0.5 | 0E+00 | 0 | 13 | 4 | 3.5 | 12 |

Table 10: Training Datasets Statistics.

| Dataset | IDs | Images |
|---|---|---|
| CUHK02 | 1,816 | 7,264 |
| CUHK03 | 1,467 | 14,097 |
| DukeMTMC-Re-Id | 1,812 | 36,411 |
| Market-1501 | 1,501 | 29,419 |
| CUHK-SYSU | 11,934 | 34,547 |

Table 11: Testing Datasets statistics.

| Dataset | Probe | | Gallery | |
|---|---|---|---|---|
| | Pr. IDs | Pr. Imgs | Ga. IDs | Ga. imgs |
| PRID | 100 | 100 | 649 | 649 |
| GRID | 125 | 125 | 1025 | 1,025 |
| VIPeR | 316 | 316 | 316 | 316 |
| i-LIDS | 60 | 60 | 60 | 60 |

Table 12: Details of our MNIST ConvNet architecture. All convolutions use 3×3 kernels and "same" padding

| # | Layer |
|---|---|
| 1 | Conv2D (in=d, out=64) |
| 2 | ReLU |
| 3 | GroupNorm (groups=8) |
| 4 | Conv2D (in=64, out=128, stride=2) |
| 5 | ReLU |
| 6 | GroupNorm (8 groups) |
| 7 | Conv2D (in=128, out=128) |
| 8 | ReLU |
| 9 | GroupNorm (8 groups) |
| 10 | Conv2D (in=128, out=128) |
| 11 | ReLU |
| 12 | GroupNorm (8 groups) |
| 13 | Global average-pooling |

Table 13: The domain generalization/adaptation accuracy on Rotated MNIST.

| Algorithm | Rotated MNIST 0 | 15 | 30 | 45 | 60 | 75 | Avg |
|---|---|---|---|---|---|---|---|
| ERM (Vapnik, 1999) | 95.3 ± 0.2 | 98.7 ± 0.1 | 98.9 ± 0.1 | 98.7 ± 0.2 | 98.9 ± 0.0 | 96.2 ± 0.2 | 97.8 |
| IRM (Arjovsky et al., 2019) | 94.9 ± 0.6 | 98.7 ± 0.2 | 98.6 ± 0.1 | 98.6 ± 0.2 | 98.7 ± 0.1 | 95.2 ± 0.3 | 97.5 |
| DANN (Ganin et al., 2016) | 95.9 ± 0.1 | 98.6 ± 0.1 | 98.7 ± 0.2 | 99.0 ± 0.1 | 98.7 ± 0.0 | 96.5 ± 0.3 | 97.9 |
| ARM (Zhang et al., 2021b) | 95.9 ± 0.4 | 99.0 ± 0.1 | 98.8 ± 0.1 | 98.9 ± 0.1 | 99.1 ± 0.1 | 96.7 ± 0.2 | 98.1 |
| DANN+ELS | 96.3 ± 0.1 | 98.7 ± 0.1 | 98.9 ± 0.3 | **99.1 ± 0.1** | 98.7 ± 0.0 | 96.9 ± 0.5 | 98.1 |
| ↑ | 0.4 | 0.1 | 0.2 | 0.1 | 0.0 | 0.4 | 0.2 |

Table 16: Comparison with recent state-of-the-art DG-ReID methods. —— denotes DANN cannot converge and attains infinite loss.

| Methods | Average R-1 | mAP | VIPeR R-1 | R-5 | R-10 | mAP | PRID R-1 | R-5 | R-10 | mAP | GRID R-1 | R-5 | R-10 | mAP | i-LIDS R-1 | R-5 | R-10 | mAP |
|---|---|---|---|---|---|---|---|---|---|---|---|---|---|---|---|---|---|---|
| DIMN (Song et al., 2019) | 47.5 | 57.9 | 51.2 | 70.2 | 76.0 | 60.1 | 39.2 | 67.0 | 76.7 | 52.0 | 29.3 | 53.3 | 65.8 | 41.1 | 70.2 | 89.7 | 94.5 | 78.4 |
| DualNorm (Jia et al., 2019) | 57.6 | 61.8 | 53.9 | 62.5 | 75.3 | 58.0 | 60.4 | 73.6 | 84.8 | 64.9 | 41.4 | 47.4 | 64.7 | 45.7 | 74.8 | 82.0 | 91.5 | 78.5 |
| DDAN (Chen et al., 2021) | 59.0 | 63.1 | 52.3 | 60.6 | 71.8 | 56.4 | 54.5 | 62.7 | 74.9 | 58.9 | **50.6** | 62.1 | 73.8 | 55.7 | 78.5 | 85.3 | 92.5 | 81.5 |
| DIR-ReID (Zhang et al., 2021c) | 63.8 | 71.2 | 58.5 | 76.9 | **83.3** | 67.0 | 69.7 | 85.8 | 91.0 | 77.1 | 48.2 | 67.1 | 76.3 | **57.6** | 79.0 | **94.8** | 97.2 | 83.4 |
| MetaBIN (Choi et al., 2021) | 64.2 | 71.9 | 59.3 | 76.8 | 81.9 | 67.6 | 70.6 | 86.5 | 91.5 | 78.2 | 47.3 | 66.0 | 74.0 | 56.4 | 79.5 | 93.0 | **97.5** | 85.5 |
| Group DRO (Sagawa et al., 2019) | 57.1 | 65.9 | 48.5 | 68.4 | 77.2 | 57.8 | 66.1 | 86.5 | 90.6 | 74.8 | 38.7 | 58.8 | 66.6 | 48.6 | 74.8 | 90.8 | 96.8 | 81.9 |
| Unit-DRO (Zhang et al., 2022b) | 65.4 | 72.8 | **60.0** | **78.2** | 82.8 | **68.4** | **73.5** | 85.3 | 91.7 | **79.4** | 47.5 | **69.3** | 77.4 | 57.2 | **80.7** | 94.0 | 97.0 | **86.2** |
| DANN (Ganin et al., 2016) | —— | —— | —— | | | | —— | | | | —— | | | | —— | | | |
| DANN+ELS | 64.2 | 72.1 | 59.3 | 76.4 | 82.7 | 67.4 | 69.6 | **87.7** | **91.7** | 77.7 | 48.1 | 67.5 | **77.8** | 57.2 | 79.8 | 94.7 | 97.2 | 86.1 |

Table 17: Generalization performance on multiple unseen target domains. ↑ denotes improvement of DANN+ELS compared to DANN, and $\gamma$ is the hyper-parameter for environment label smoothing.

| | Rotated MNIST Target domains $\{0°, 30°, 60°\}$ | | | Target domains $\{15°, 45°, 75°\}$ | | | |
|---|---|---|---|---|---|---|---|
| Method | 0° | 30° | 60° | 15° | 45° | 75° | Avg |
| ERM (Vapnik, 1999) | 96.0 ± 0.3 | 98.8 ± 0.4 | 98.7 ± 0.1 | 98.8 ± 0.3 | 99.1 ± 0.1 | 96.7 ± 0.3 | 98.0 |
| IRM (Arjovsky et al., 2019) | 80.9 ± 3.2 | 94.7 ± 0.9 | 94.3 ± 1.3 | 94.3 ± 0.8 | 95.5 ± 0.5 | 91.1 ± 3.1 | 91.8 |
| DANN (Ganin et al., 2016) | 96.6 ± 0.2 | 98.8 ± 0.3 | 98.7 ± 0.1 | 98.6 ± 0.4 | 98.8 ± 0.2 | 96.9 ± 0.1 | 98.1 |
| DANN+ELS | 96.7 ± 0.4 | 98.9 ± 0.2 | 98.8 ± 0.1 | 98.8 ± 0.1 | 99.0 ± 0.2 | 97.0 ± 0.4 | 98.2 |
| ↑ | 0.1 | 0.1 | 0.1 | 0.2 | 0.2 | 0.1 | 0.1 |

Table 18: Generalization performance on sequential benchmarks. ↑ denotes improvement of DANN+ELS compared to DANN.

| | HHAR | | | | | |
|---|---|---|---|---|---|---|
| *Train-domain validation* | | | | | | |
| **Algorithm** | **Nexus 4** | **Galazy S3** | **Galaxy S3 Mini** | **LG watch** | **Sam. Gear** | **Average** |
| ID ERM | 98.91±0.24 | 98.44±0.15 | 98.68±0.15 | 90.08±0.28 | 80.63±1.33 | 93.35 |
| ERM | 97.64±0.15 | 97.64±0.09 | 92.51±0.46 | 71.69±0.14 | 61.94±1.04 | 84.28 |
| IRM | 96.02±0.17 | 95.75±0.22 | 89.46±0.50 | 66.49±0.94 | 57.66±0.37 | 81.08 |
| SD | 98.14±0.01 | 98.32±0.19 | 92.71±0.09 | 75.12±0.18 | 63.85±0.28 | 85.63 |
| VREx | 95.81±0.50 | 95.92±0.23 | 90.72±0.10 | 69.04±0.23 | 56.42±1.57 | 81.58 |
| DANN | 94.45 ± 0.44 | 95.05 ± 0.10 | 88.70 ± 0.56 | 68.33 ± 0.49 | 58.45 ± 1.24 | 80.99 |
| DANN+ELS | 95.95 ± 0.39 | 95.65 ± 0.42 | 90.50 ± 0.39 | 69.55 ± 0.36 | 58.45 ± 0.24 | 82.02 |
| ↑ | 1.5 | 0.6 | 1.8 | 1.22 | 0.0 | 1.03 |
| *Oracle train-domain validation* | | | | | | |
| **Algorithm** | **Nexus 4** | **Galazy S3** | **Galaxy S3 Mini** | **LG watch** | **Sam. Gear** | **Average** |
| ID ERM | 98.91±0.24 | 98.44±0.15 | 98.68±0.15 | 90.08±0.28 | 80.63±1.33 | 93.35 |
| ERM | 97.98±0.02 | 97.92±0.05 | 93.09±0.15 | 71.96±0.04 | 64.08±0.66 | 85.01 |
| IRM | 96.02±0.17 | 95.75±0.22 | 89.91±0.25 | 68.00±0.34 | 57.77±0.42 | 81.49 |
| SD | 98.48±0.01 | 98.67±0.11 | 94.36±0.24 | 75.12±0.18 | 64.86±0.28 | 86.3 |
| VREx | 96.65±0.18 | 96.30±0.05 | 90.98±0.16 | 69.39±0.27 | 59.12±0.80 | 82.49 |
| DANN | 95.95 ± 0.21 | 96.20 ± 0.07 | 89.91 ± 0.73 | 72.70 ± 0.63 | 58.45 ± 1.77 | 82.64 |
| DANN+ELS | 96.79 ± 0.13 | 96.94 ± 0.13 | 91.57 ± 0.22 | 72.70 ± 0.63 | 59.80 ± 0.84 | 83.56 |
| ↑ | 0.84 | 0.74 | 1.66 | 0.0 | 1.35 | 0.92 |

