# OpenReview forum: "Free Lunch for Domain Adversarial Training: Environment Label Smoothing"
_ICLR.cc/2023/Conference — ICLR 2023 poster_

### Official Review · Reviewer_kFo4 · 2022-10-21

**Confidence:** 3
**Correctness:** 4
**Technical Novelty And Significance:** 2
**Empirical Novelty And Significance:** 3
**Recommendation:** 6

**Clarity, Quality, Novelty And Reproducibility:**

This paper is great in clarity and quality. The proposed ELS method is simple but the technical contribution is limited. Implementation details and codes are provided to ensure reproducibility.

**Details Of Ethics Concerns:**

No ethic concerns appear.

**Strength And Weaknesses:**

Strength:
- This paper is well-organized and easy to follow. The general framework is pretty clear and follows rigorous logic.
- The theoretical analysis shows many helpful properties of ELS, which are further validated by effective experimental results. To my opinion, this is a very solid paper.
- The ELS method is very easy to implement. The details as well as codes are provided to ensure reproducibility.

Weakness:
- Although this paper is quite solid, I am still concerned about the motivation for introducing label smoothing. Indeed, the domains from the VLCS dataset are similar to each other. However, in other domain adaptation datasets containing significantly different domains (such as the Office-Home dataset), using smoothed domain labels might not improve the performance any further. As shown in the continuously indexed domain adaptation in Table 5, the improvement is 10%. Compared to tables 2 and 4, the improvement of ELS is less than 1%, especially in Table 4, ELS only surpasses the backbone method by 0.3%. So, is it possible that the proposed method is only effective on a dataset that has a small domain gap, and as the domain gap becomes smaller, the improvement would decrease?
- When the encoder gets better, the generated features from different domains are more similar. However, a good encoder is also based on the quality of the discriminator. By using label smoothing, the discriminator can not be as discriminative as vanilla DAT which leverage hard labels, thus the performance of the encoder could doubtful. Is there any analysis of the learned discriminator?
- Moreover, the improvements on the dataset Office-Home can be achieved by tuning the Hyper-parameter $\gamma$, whose optimal value is not provided in the parameter analysis in Figure 4 (b). In my opinion, by tuning $\gamma$, there is always a better classifier than the vanilla DAT.
- In the introduction, the authors are motivated by the observation that VLCS has similar domains, thus label smoothing (smaller $\gamma$) should be introduced, instead of the vanilla DAT ($\gamma$=1). However, in Section 4.2, “PACS can be easily discriminated”, thus the optimal $\gamma$ is smaller than VLCS. Is this a contradiction? Please give an explanation.
- Additionally, the selection of $\gamma$ is unclear to me. Do the authors conduct a grid search in the experiments? How to obtain the mean and std of $\gamma$ to produce Figure 4 (b)?

**Summary Of The Paper:**

This paper studies the domain adversarial training problem and propose to use label smoothing for domain discrimination. Specifically, motivated by the observation that different domains from VLCS dataset show small difference, thus a soft domain label should be applied to domain adversarial training instead of hard label. By simply rewriting the cross-entropy loss, the proposed ELS can be easily implemented. Extensive theoretical and empirical analyses are provided to support the proposed method.

**Summary Of The Review:**

I have carefully read the whole paper. This paper is quite solid and has excellent completeness, however, there are still some concerns (see weaknesses). If the authors can address my concerns, I will consider raising my score.

---

> ### Author Response · Authors · 2022-11-17
> **Response to Reviewer  kFo4 [1/2]**
>
> We thank the reviewer for the careful reading and valuable feedback on our submission. We appreciate the agreement to our motivation based on good observations. We hope our response could solve your concerns.
>
> ----
>
> **[Question 1] Effectiveness against datasets with different domain gaps.**
>
> In the introduction section, we motivate this work via the domain gap issue in VLCS datasets. In this dataset, the domains are hard to distinguish and then environment label noise is more likely to exist. For significantly different domains, the discriminators will be easy to train and be overconfident, which leads to highly oscillatory gradients and is harmful to training stability, which is verified in Figure .5.
>
> We next conduct the experiments on the Office-Home dataset. As conjectured by the reviewer, using smoothed domain labels might not improve the performance any further.  We run domain generalization experiments on the Office-Home with the resnet-18 backbone for time limitations. The results are shown in the following table. We observe a not marginal improvement.
>
> | **Method**    | **A** | **C** | **P** | **R** | **Avg** |
> |---------------|-------|-------|-------|-------|---------|
> | DANN          |  50.1 |  44.6 |  67.8 |  67.5 |    57.5 |
> | DANN+ELS      |  50.9 |  46.6 |  68.6 |  69.9 |      59 |
> | Best $\gamma$ |   0.7 |   0.7 |   0.8 |   0.6 |         |
>
>
> The reason why the performance gains in Table.4 with the ViT backbone is just 0.3% may be from two aspects: (1) with a stronger backbone, the performance is harder to improve. For example, compared to CDAN, its variant SDAT can improve the performance by about 10% with the ResNet-50 backbone while only 5% with the ViT backbone. (2) For domain adaptation settings, only two domains are given and the influence of ELS will be weaker than that in Domain generalization. As shown in the DG results on the Office-home, VLCS and PACS dataset (table.3 in the main manuscript), It seems that **with a larger domain divergence, the performance gains brought by ELS will be also improved**. The argument is quite interesting and we will study the idea in future work. Thanks for your kind suggestion!
>
> ----
>
> **[Question 2] The analysis of the discriminator.**
>
> We agree with the point ``a good encoder is also based on the quality of the discriminator’’. In an adversarial training setting, however, a  highly discriminative discriminator may not be a good choice. In actuality, an optimal discriminator always leads to gradient vanishing [1] and is the main reason for training instability [2]. To verify this point for domain generalization, we conduct the following experiments on the PACS dataset.
>
> Specifically, after each encoder training step, we train the discriminator multiple times until its classification accuracy is greater than a margin $\epsilon$ or the accuracy will not be improved. In this case, we can study the impact of discriminator classification accuracy on the DAT algorithm.
>
>
> | **Acc of Disc** |   0.1 |   0.2 |   **0.3** |   0.4 |   0.5 |   0.6 |   0.7 |   0.8 |   0.9 |
> |-----------------|------:|------:|----------:|------:|------:|------:|------:|------:|------:|
> | **Performance** | 67.72 | 67.48 | **83.37** | 80.68 | 74.32 | 77.24 | 73.64 | 75.86 | 75.77 |
>
>
> Because adversarial training is a three-player game between classifier, encoder, and domain discriminator [3]. The local Nash equilibria exist, however, it is not necessarily obtained when the discriminator is optimal. Namely, a powerful discriminator trained by hard labels as you said is not the best choice.  In fact, a powerful or even optimal discriminator usually has a negative effect on the optimization of adversarial training [2], as shown in our table. Utilizing label smoothing can reduce discrimination confidence and brings more chances to attain the equilibria point.
>
>
> [1] Arjovsky, Martin, and Léon Bottou. "Towards principled methods for training generative adversarial networks." arXiv preprint arXiv:1701.04862 (2017).
>
> [2] Mescheder, Lars, Andreas Geiger, and Sebastian Nowozin. "Which training methods for GANs do actually converge?." International conference on machine learning. PMLR, 2018.
>
> [3] Acuna, David, et al. "Domain Adversarial Training: A Game Perspective." International Conference on Learning Representations. 2021.

---

> > ### Author Response · Authors · 2022-11-17
> > **Response to Reviewer kFo4 [2/2]**
> >
> >
> > **[Question 3] By tuning γ, there is always a better classifier than the vanilla DAT..**
> >
> > The best $\gamma$ for the office-home dataset is shown in the following table. We can see that most $\gamma$ is not one, namely **smoothing the environment labels is always helpful for the generalization performance**, which agrees with your idea that tuning γ, there is always a better classifier than the vanilla DAT.
> >
> > | **Best $\gamma$** | **A-C** | **A-P** | **A-R** | **C-A** | **C-P** | **C-R** | **P-A** | **P-C** | **P-R** | **R-A** | **R-C** | **R-P** |
> > |-------------------|:-------:|:-------:|:-------:|:-------:|:-------:|:-------:|:-------:|:-------:|:-------:|:-------:|:-------:|:-------:|
> > | ResNet-50         |   0.80  |   0.50  |   0.55  |   0.80  |   0.75  |   0.55  |   0.90  |   0.95  |   0.95  |   0.95  |   0.90  |   0.75  |
> > | ViT               |   0.80  |   0.90  |   0.75  |   0.90  |   0.95  |   1.00  |   1.00  |   0.95  |   0.95  |   0.95  |   0.90  |   0.95  |
> >
> > ----
> >
> > **[Question 4] The interpretation of $\gamma$ in PACS and VLCS dataset.**
> >
> > Thanks a lot for the valuable question and careful reading. Our statement is not rigorous enough and leads to misunderstanding, here we make the following elaboration and revision in the manuscript.
> >
> > We propose two motivations in the abstract, where the first is the label noise in the VLCS, and the second is the over-confident discriminator. Your statement that ``PACS is easy to discriminate” is exactly right, namely, the environment label noise is scarce in the PACS. However, as shown in the following table, the domain discriminator for the PACS dataset is easy to over-confident. With only $0.1$% images, a train-from-scratch domain discriminator will attain more than $97$% domain discrimination accuracy in 1000 training steps. As we discussed in the introduction and theoretical results, an over-confident discriminator is harmful to the optimization process and we then need a small $\gamma$ to alleviate the problem.
> >
> > | **Dataset** | **Iterations** | **Label Ratio** | **Performance** |
> > |-------------|----------------|-----------------|-----------------|
> > | PACS        |            100 |           0.001 |    0.7318548387 |
> > | PACS        |           1000 |            0.01 |     0.984231943 |
> > | VLCS        |            100 |           0.001 |    0.3628156905 |
> > | VLCS        |           1000 |            0.01 |    0.7239995662 |
> >
> >
> > In fact, when the environment label noise is large, we really need a smaller $\gamma$ and the performance gains brought by ELS will be larger (Figure .4), so the two points do not conflict, but it does cause some misunderstandings. We will explain this problem in the revised version. Thanks again.
> >
> > ----
> >
> >
> > **[Question 5] The selection of the best $\gamma$.**
> >
> >
> > Your understanding is exactly right and we indeed conduct a grid search in the experiments. Considering Figure 4 (b), because both PACS and VLCS have four domains, where each domain will be used as the target domain and the best $\gamma$s are as follows.
> > | Datasets       | PACS |     |     |     | VLCS |     |     |     |
> > |----------------|:----:|:---:|:---:|:---:|:----:|:---:|:---:|:---:|
> > | Domains        |   A  |  C  |  P  |  S  |   C  |  L  |  S  |  V  |
> > | Best $\gamma$s |  0.7 | 0.5 | 0.8 | 0.7 |  0.9 | 0.9 | 0.7 | 0.8 |
> >
> > The mean and std are calculated from $\gamma$ for all target domains. For other datasets, some specific domains are used for evaluation and no std can be calculated.

---

> > > ### Comment · Reviewer_kFo4 · 2022-11-19
> > > **Further Response**
> > >
> > > Dear Authors,
> > >
> > > Thanks for addressing my concerns. Although this work is rather technically simple, it is strongly supported by extensive theoretical and empirical analysis. In my opinion, "domain" is really a human-defined concept, which is the same as "class". So, there always be improved modeling through continuous label smoothing.
> > > This work has done some solid contributions to the domain generalization community. Hence, I have raised my score to 6 and vote for acceptance.
> > >
> > > Best,
> > > Reviewer.

---

### Official Review · Reviewer_hUFN · 2022-10-24

**Confidence:** 4
**Correctness:** 3
**Technical Novelty And Significance:** 3
**Empirical Novelty And Significance:** 2
**Recommendation:** 8

**Clarity, Quality, Novelty And Reproducibility:**

The paper is of high clarity and quality. Novelty is slightly questionable since the building blocks are not new.

**Strength And Weaknesses:**

Strengths
1. In general, I am in favor of simple-but-effective improvements.
2. The authors demonstrated via theoretical validations three advantages of the proposed method: (1) training stability, (2) robustness to label noise, and (3) non-asymptotic convergence speed. Further, the authors used empirical studies to complement the theoretical results. (1) is supported by Figure 2 and 5; (2) by Figure 4(a); and (3) by Figure 5.
3. I particularly like section 3.4, where the authors explicitly faced the unrealistic assumptions in their theoretical validation and analyzed the “empirical gap and parametrization gap”.
4. I am also glad about the fact that the authors intentionally organize the proofs and non-central results to the appendix, leaving the logic flow in the main text fluent and engaging.

Weaknesses
1. On the flip side of the first strong point I listed, one potential concern may be the lack of the so-called “novelty”. I fully acknowledge the thoroughness and comprehensiveness of this work, yet the technical contribution (apart from the analyses and insights) is among the least “innovative” I have seen. Afterall, label smoothing has been around for such a long time.

Minor Things
1. Would it be slightly more intuitive to write $D_S’ = D_T + \lambda(D_S - D_T)$ as $D_S’ = \lambda D_S + (1 - \lambda) D_T$ instead?
2. Typo. Page 2. “phenomena” instead of “phenomenons”.
3. Minor grammar thing. Page 3, under Proposition 2. “One **may** argue that adjusting the tradeoff weight λ can also balance AT and ERM**.** **H**owever, λ can only adjust the gradient contribution of AT part, …”

**Summary Of The Paper:**

Poor generalization on out-of-distribution data is a common problem among machine learning models. In classification tasks, one recently proposed solution is Domain Adversarial Training, which asks the model to predict the domain of the data sample in addition to its class, and encourages the model to learn domain-agnostic class-relevant features through an adversarial objective. However, this method is known to suffer from training instability. To that end, the authors proposed an extremely simple modification — label smoothing on the environment (i.e., domain) label — that improves not only the training stability but also robustness to label noise, convergence speed, and performance.

**Summary Of The Review:**

This paper introduced a seemingly trivial modification, label smoothing on environment label, to a popular approach for out-of-distribution generalization. Notably, the authors showcased the advantages of the label smoothing in three different aspects through a combination of theorical and empirical analyses. Overall, I find this a very solid piece of work and would recommend for acceptance of this submission.

---

> ### Author Response · Authors · 2022-11-17
> **Response to Reviewer hUFN**
>
> Thanks for your sincere comments. We appreciate your huge efforts for the valuable suggestions. We appreciate the agreement to the simple-but-effective method and our theoretical and empirical results. We modified the main paper based on your suggestion and fixed the typos. We hope our response could solve your concerns.
>
> ----
>
> **[Weaknesses] The novelty of this work. Label smoothing has been around for such a long time**
>
> We thank the reviewer for bringing this problem to our attention. This work tries to exploit the simple label smoothing to a more general setting, namely not for class labels but for environment labels.  At the same time, another major contribution is to explain why smoothing environment labels benefit domain generalization/adaptation and GANs.
>
>  We believe our proposed ELS is different from the existing literature in several perspectives listed as follow:
>
>
> 1. **The problem setting is different and the philosophy of using ELS  is beyond improving the accuracy or model calibration [1]**. In particular, we apply label smoothing to environment labels instead of class labels in conventional label smoothing settings. Our primary goal of using ELS is to address the discriminator overconfidence and environment label noise issues that arise in domain adversarial training, which is shown able to stabilize the training procedure.
>
> 2. **The proposed method is simple yet effective**. In our numerical experiments, the proposed ELS-based methods achieve superior generalization performance. With such a simple idea, ELS can boost generalization performance by a large margin in **5 tasks**, including **13 datasets** and **10 backbones**.
>
> 3. **We conduct rigorous theoretical analysis to show the nationality of using ELS**. We analyze the divergence minimization interpretation, the analysis in training stability, asymptotic convergence analysis, the empirical and parameterization gap, and the benefit of ELS under noisy labels.  Different from the standard GANs-related analysis, we innovate to (1) introduce a KL-divergence minimization interpretation for multiple source domains adversarial training; (2) utilize the Taylor series technique in gradient analysis of multi-domain adversarial training, indicating that ELS serves as a data-driven regularization and stabilizes the oscillatory gradients; (3) adopt the symmetric noise model to analyze the effect of ELS to noise labels; (4) analyze the empirical and parameterization gap of multi-domain adversarial training; (5) introduce the Hurwitz condition to analyze the asymptotic convergence analysis behavior of our adversarial training system when ELS is applied, indicating that with ELS, the adversarial training system could converge faster. To our best knowledge, **we are the first ones to combine those novel techniques with GANs analysis and derive meaningful results in DAT**.
>
>  ----
>
> **[Other concerns] The definition of mixed distribution.**
>
> This is a very valuable suggestion, thanks a lot, and we will follow your suggestion to define $D_{S'}=\gamma D_S+(1-\gamma)D_T$ and $D_{T'}=\gamma D_T+(1-\gamma)D_S$. We have formally added it to Proposition.1 and Proposition.2 in the manuscript.
>
> ----
>
> If you have any questions, please let us know. Thanks again.

---

> > ### Comment · Reviewer_hUFN · 2022-12-05
> > **Response to the authors' comment**
> >
> > The authors have nicely cleared my concerns. I have increased my rating.

---

> ### Author Response · Authors · 2022-11-28
> **A Gentle Reminder of Feedbacks**
>
> Dear Reviewer hUFN,
>
> Thanks again for your careful reading and valuable comments to improve our submission. We want to leave a gentle reminder due to the closing end time of the discussion period. We have tried our best to address your concerns point by point with detailed explanations. We would really appreciate feedback to make sure the responses and revisions have addressed all your concerns, or whether there is a leftover concern we can address.
>
> Sincerely
>
> Authors of Paper3679

---

### Official Review · Reviewer_LfGU · 2022-10-31

**Confidence:** 4
**Correctness:** 3
**Technical Novelty And Significance:** 2
**Empirical Novelty And Significance:** Not applicable
**Recommendation:** 5

**Clarity, Quality, Novelty And Reproducibility:**

The paper shared the code and seems like the results would be reproducible though I haven't checked all of them.

**Strength And Weaknesses:**

Strength:
- The paper is well written.
- The empirical results are backed by theoretical evidences
- Simple solution

Weakness
- The paper used an already established idea with minor modifications and that is the major drawback of the work
- The paper showed label smoothing approach but the question is wouldn't it attribute to the catastrophic forgetting of the discriminator.
Questions:
1) The contribution seems marginal as most of the proofs (Theorem 1 and 2) and  can be directly inferred from previous GAN based works.
2) Proposition 1, do we need both the $\gamma(D_{S}-D_{T})$ and $\gamma(D_{T}-D_{S})$
3) Proposition 2, would $p_{s^{'}}$ be $p_{t}$? If proposition 2 have issues for $p_{s}\rightarrow 0$ then that would be similar for proposition 1, right?
4) Proposition 3, for multilabel scenario would the $D_{Mix}$ provide any meaningful information? Though theoretically we could show the use of it but I am curious whether there would be any practical implication of that.
5) Sec A.5, would the mixture provide any meaningful information or it would do only catastrophic forgetting to the discriminator?

**Summary Of The Paper:**

The paper proposes a DAT label smoothing approach. The results are backed by the theoretical evidences. However, the main novelty is minimalistic.

**Summary Of The Review:**

The papers proposes a label smoothing idea for DAT. The novelty is limited as the idea of label smoothing is already established in so many previous papers.

---

> ### Author Response · Authors · 2022-11-17
> **Response to Reviewer LfGU [1/2]**
>
> We thank the reviewer for the careful reading and valuable feedback on our submission and hope our response can address your concerns.
>
> ----
>
> **[Weakness 1] The paper used an already established idea with minor modifications and that is the major drawback of the work.**
>
>
> We thank the reviewer for bringing it to our attention. We believe our proposed ELS is different from the existing literature in several perspectives listed as follows:
>
> 1. **The problem setting is different and the philosophy of using ELS  is beyond improving the accuracy or model calibration [1]**. In particular, we apply label smoothing to environment labels instead of class labels in conventional label smoothing settings. Our primary goal of using ELS is to address the discriminator overconfidence and environment label noise issues that arise in domain adversarial training, which is shown able to stabilize the training procedure.
>
> 2. **The proposed method is simple yet effective**. In our numerical experiments, the proposed ELS-based methods achieve superior generalization performance. With such a simple idea, ELS can boost generalization performance by a large margin in **5 tasks**, including **13 datasets** and **10 backbones**.
>
> 3. **We conduct rigorous theoretical analysis to show the nationality of using ELS**. We propose the divergence minimization interpretation, the analysis in training stability, asymptotic convergence, empirical and parameterization gap, and the benefit of ELS under noisy labels.  Since domain adversarial training is similar to GAN, it enables the use of similar theoretical foundations to conduct the analysis. We want to highlight that **our analysis contains non-trivial extensions of the standard GANs-related analysis**, we defer the detailed discussion of the response to “Technique innovations in Theorem 1 and 2. ”
>
> [1] Müller, Rafael, Simon Kornblith, and Geoffrey E. Hinton. "When does label smoothing help?." Advances in neural information processing systems 32 (2019).
>
> ----
>
> **[Weakness 2] The paper showed a label smoothing approach but the question is wouldn't it attribute to the catastrophic forgetting of the discriminator.**
>
>
> It is a good question. In classic GAN literature (e.g., [1]), during the training stage,  a discriminator with high complexity/capacity may suffer from the gradient of the generator vanishing, which makes the GAN training extremely hard. Various methods are proposed to prevent the discriminator from being over-confident. In Sec A.4 and A.5, we show that by using ELS, the gradient of the optimal discriminator will not be zero, which implies injecting noise to gradients can keep the optimization procedures from getting struck by a near-optimal discriminator. Namely, label smoothing prevents over-confident discriminators from forgetting some knowledge.
>
> [1] Arjovsky, Martin, and Léon Bottou. "Towards principled methods for training generative adversarial networks." arXiv preprint arXiv:1701.04862 (2017).
>
> ----
>
> **[Question 1] Technique innovations in Theorem 1 and 2**.
>
>  We summarize our innovation points as follows. (1) introduce a KL-divergence minimization interpretation for multiple source domains adversarial training; (2) utilize the Taylor series technique in gradient analysis of multi-domain adversarial training, indicating that ELS serves as a data-driven regularization and stabilizes the oscillatory gradients; (3) adopt the symmetric noise model to analyze the effect of ELS to noise labels; (4) analyze the empirical and parameterization gap of multi-domain adversarial training; (5) introduce the Hurwitz condition to analyze the asymptotic convergence analysis behavior of our adversarial training system when ELS is applied, indicating that with ELS, the adversarial training system could converge faster. To our best knowledge, we are the first one to combine those novel techniques with GANs analysis and derive meaningful results in DAT.
>
> ----
>
>
>
> **[Question 2] Do we need both $\gamma(D_S-D_T)$ and $\gamma(D_T-D_S)$?**.
>
> We don’t quite catch the key point to this question. It would be very appreciated if the reviewer could elaborate more on it.  Below, we give the response based on our current understanding and hope it could address your concerns.
>
> We guess your question may be due to the definition being unclear, so here we give a more intuitive definition of mixed distributions as follows.
>
> $D_{S'}=\gamma D_S+(1-\gamma)D_T$ and $D_{T'}=\gamma D_T+(1-\gamma)D_S$, namely $D_{S'}$ is a distribution that mixed of $D_S$ and $D_T$ with a parameter $\gamma$.
>
> Along this line,  $\gamma(D_S-D_T)$ and $\gamma(D_T-D_S)$ have different physical meanings and we need both in the theorem statement. If the reviewer suggests incorporating the definition above into the main paper, we are more than happy to do that. Please advice.

---

> > ### Author Response · Authors · 2022-11-17
> > **Response to Reviewer LfGU [2/2]**
> >
> >
> > **[Question 3] Proposition 2, would $p_{s′}$ be $p_t$? If proposition 2 has issues for  $p_s→0$  then that would be similar to proposition 1, right?**
> >
> > First, we want to mention that one-side label smoothing is different from classic label smoothing because the former only smooths the label of real data ($D_S$ in proposition 2). Therefore, $p_{s’}$ will not be $p_t$ in proposition 2.
> >
> > As we discussed in Appendix A.2, for the classic label smoothing, if the density of real data in a region is near zero (i.e., $p_s(x)\rightarrow 0$), the traditional environment label smoothing will lead to $p_{s'}=p_t+\gamma(p_s-p_t)\approx (1-\gamma) p_t$. Namely, the discriminator will not align the distribution between generated samples and real samples but enforce the generator to produce samples that follow the fake mode $D_T$. In contrast, one-sided label smoothing reserves the real distribution density as far as possible, that is, $p_{s'}=\gamma p_s, p_{t'}\approx \gamma p_t$, which avoids divergence minimization between fake mode to fake mode and relieves model collapse. Therefore, when $p_s→0$, one-side label smoothing has a better impact than the traditional LS (discussed in proposition 1) on the generator models.
> >
> > ----
> >
> > **[Question 4] Proposition 3, for multilabel scenarios would the $D_{Mix}$  provide any meaningful information?**
> >
> > Good question. Since this paper focuses on domain generalization and domain adaptation, the common problem setting may involve multiple domains. Exploring the influence of multi-domains compared to only two domains (a source and a target) from the theoretical perspective is very important to the out-of-distribution generalization fields.
> >
> > In our analysis, the term $D_{Mix}$  represents the mixed domain of all source domains. In domain generalization [1,2] or multi-source domain adaptation [3], people usually consider training the model with data from various domains. Via the $D_{Mix}$, we can easily extend the existing two-domain analysis to the complex multi domains settings.
> >
> >
> > [1] Zhang, Hanlin, et al. "Towards principled disentanglement for domain generalization." Proceedings of the IEEE/CVF Conference on Computer Vision and Pattern Recognition. 2022.
> >
> > [2] Albuquerque, Isabela, et al. "Adversarial target-invariant representation learning for domain generalization." (2020).
> >
> > [3] Peng X, Bai Q, Xia X, et al. Moment matching for multi-source domain adaptation[C]//Proceedings of the IEEE/CVF international conference on computer vision. 2019: 1406-1415.
> >
> > ----
> >
> > **[Question 5] Sec A.5, would the mixture provide any meaningful information or it would do only catastrophic forgetting to the discriminator??**
> >
> > See the response to weakness 2.

---

> ### Author Response · Authors · 2022-11-28
> **A Gentle Reminder of Feedbacks**
>
> Dear Reviewer LfGU,
>
> Thanks again for your careful reading and valuable comments to improve our submission. We want to leave a gentle reminder due to the closing end time of the discussion period. We have tried our best to address your concerns point by point with detailed explanations. We would really appreciate feedback to make sure the responses and revisions have addressed all your concerns, or whether there is a leftover concern we can address.
>
> Sincerely
>
> Authors of Paper3679

---

### Decision · Program_Chairs · 2023-01-20

**Decision:**

Accept: poster

**Justification For Why Not Higher Score:**

As described in detail below, although the AC believes the paper deserves acceptance, its message to the audience is relatively simple: domain label smoothing works for DMT. It is useful for the domain adaptation/generalization community but may be not for the broader audience, so the AC supposes that poster is a suitable venue.

**Justification For Why Not Lower Score:**

This paper received the initial scores of 5,5,6. While the all reviewers acknowledged the simplicity and effectiveness of the method as well as its theoretical analysis, the main concern was its novelty since it is seemingly a trivial import of the popular label smoothing technique to this task. They also raised concerns regarding some details of the method, analysis and experiments.
During the discussion period, the authors have presented additional results as well as the clarifications of the method, experiments, and theoretical contributions. The reviewers are mostly convinced by the new explanations and results, and have raised their scores to 6,6,8. (one update (5 to 6) is still not visible in OpenReview).
Overall, the idea of the proposed method might be seemingly straightforward, but this paper is quite solid and well beyond a ‘just tried and it worked’ paper. It is deeply analyzed in terms of different interesting viewpoints supported by well-founded theories and convincing experiments. The results and findings of the paper will stimulate the domain generalization community that such a simple idea is indeed effective. The reviewers are unanimous for acceptance, and the AC also believes that the pros of the paper sufficiently overweigh the cons, thus recommends acceptance.


**Metareview: Summary, Strengths And Weaknesses:**

Summary:
For the problem of out-of-distribution generalization, this paper presents a conceptually very simple yet effective method for domain adversarial training (DAT). Motivated by the intuition that some different domains show small difference, the proposed method utilizes soft domain labels for domain discrimination instead of the standard hard labels, in the form label smoothing. Extensive theoretical and empirical analyses are provided to support the proposed method. The proposed method is shown to boost generalization performance in 5 tasks, including 13 datasets and 10 backbones. Also notably, the authors showcased the advantages of the label smoothing in three different aspects through a combination of theorical and empirical analyses.

Strengths:
1. The proposed method is surprisingly simple but effective. It can be implemented easily, and codes are provided to ensure reproducibility.
2. Theoretical foundations are given addressing three advantages of the proposed method: (1) training stability, (2) robustness to label noise, and (3) non-asymptotic convergence speed.
3. Extensive empirical evidences support those theories.
4. The paper is written and organized well.

Weaknesses:
1. At a conceptual level, the method appears to be just an adaptation of the well-known label smoothing technique for DAT, posing the question of its novelty.


**Note From Pc:**

if the above contains the word "oral" or "spotlight" please see: "oral" presentation means -> notable-top-5% and "spotlight" means -> notable-top-25%. As stated in our emails, we are disassociating presentation type from AC recommendations